# Efficient and Sharp Off-Policy Evaluation in Robust Markov Decision Processes

**Andrew Bennett**[*]
Morgan Stanley
andrew.bennett@morganstanley.com

**Nathan Kallus**[*]
Cornell University
kallus@cornell.edu

**Miruna Oprescu**[*]
Cornell University
amo78@cornell.edu

**Wen Sun**[*]
Cornell University
ws455@cornell.edu

**Kaiwen Wang**[*]
Cornell University
kw437@cornell.edu

## Abstract

We study the evaluation of a policy under best- and worst-case perturbations to a Markov decision process (MDP), using transition observations from the original MDP, whether they are generated under the same or a different policy. This is an important problem when there is the possibility of a shift between historical and future environments, *e.g.* due to unmeasured confounding, distributional shift, or an adversarial environment. We propose a perturbation model that allows changes in the transition kernel densities up to a given multiplicative factor or its reciprocal, extending the classic marginal sensitivity model (MSM) for single time-step decision-making to infinite-horizon RL. We characterize the sharp bounds on policy value under this model – *i.e.*, the tightest possible bounds based on transition observations from the original MDP – and we study the estimation of these bounds from such transition observations. We develop an estimator with several important guarantees: it is semiparametrically efficient, and remains so even when certain necessary nuisance functions, such as worst-case Q-functions, are estimated at slow, nonparametric rates. Our estimator is also asymptotically normal, enabling straightforward statistical inference using Wald confidence intervals. Moreover, when certain nuisances are estimated inconsistently, the estimator still provides valid, albeit possibly not sharp, bounds on the policy value. We validate these properties in numerical simulations. The combination of accounting for environment shifts from train to test (robustness), being insensitive to nuisance-function estimation (orthogonality), and addressing the challenge of learning from finite samples (inference) together leads to credible and reliable policy evaluation.

## 1 Introduction

Offline policy evaluation (OPE) from historical data is crucial in domains where active, on-policy experimentation is costly, risky, unethical, or otherwise operationally infeasible. Relevant domains range from medicine to finance and recommendation systems. However, whenever historical data is used to study future behavior, there is a concern of non-stationarity – shift between the environment generating the data (training environment) and the environment in which a policy will be deployed (test environment). This may occur, *e.g.*, due to general distributional shifts in the environment over time, unobserved confounding in the observed historical data, or adversarial elements of the environment (such as other agents) that may react when the agent is deployed. While standard OPE in offline reinforcement learning (ORL) accounts for the change between the logging and evaluation policies, it may overlook the fact that the Markov decision process (MDP) too has changed. While

---

[*]Alphabetical order.

38th Conference on Neural Information Processing Systems (NeurIPS 2024).

this issue is particularly critical in high-stakes domains, it is broadly appealing to understand how value shifts across different environments in any application domain.

Robust MDPs [34, 56] model unknown environments by allowing an adversary to choose from any one environment in a set. Therefore, they offer a natural model for unknown environment shifts by simply considering all environments to which we could possibly shift. A variety of work addresses questions such as planning in a known robust MDP [30, 51, 80] as well as online learning [6, 79]. Here we focus on a purely statistical estimation question: given observations of transitions from some unknown transition kernel, we wish to estimate the worst-case (or best-case) value of a given evaluation policy in a robust MDP, defined by a set of MDPs whose transition functions are centered around the observed transition kernel.

This setting captures the previously studied unconfounded robust OPE problem [73], where the observed transition kernel corresponds to an MDP, and the observed transitions are the result of applying some logging policy within this MDP. In such cases, the goal is to estimate policy values that are robust to future changes in the MDP dynamics. However, our setting is more general in that it also captures problems where the observed transitions are confounded by some unobserved variables, in which case they do *not* correspond to observations from the transition kernel of an MDP. In this case, the robust MDP and the robust policy value estimates are designed to account for worst-case (or best-case) impact of this confounding bias. In either case, as in ORL, we emphasize that we do *not* know the observational MDP, and can only access it via a sample of transitions. Furthermore, even in the simple case with no unmeasured confounding, in a notable departure from standard ORL, the problem can be difficult even if the logging and evaluation policies are the same (the usually easy on-policy setting), since the policy can induce very different visitation distributions in the original and perturbed MDPs.

Such robust offline evaluation from transition data was considered in recent work [12, 59]. We build on this recent work by focusing the question of statistically *efficient* and *robust* estimation of the *sharp* bounds (*i.e.*, the tightest possible given the data). Previous work focused on evaluation using only the Q-function under the worst-case environment (in some cases under a relaxation of the adversary, leading to loose bounds). Thus, any error in its estimation translates directly to error in evaluation. In other words, flexible nonparametric modeling of this function can mean slow rates for estimated bounds and a lack semiparametric efficiency. Moreover, without a clear understanding of the noise in the estimates, we cannot add confidence bands to the bounds, leading to bounds that are too tight.

We address these challenges by developing an orthogonalized estimation method that combines several nuisance functions: the worst-case $Q$-function, the state-visitation frequency in the worst-case environment, and a threshold function characterizing the worst-case transition kernel. Our first key result is that, to first order, our estimator behaves as a sample average using the true values of these functions without having to estimate them at all, provided we just estimate them at certain slow nonparametric rates. This ensures we not only have a $\sqrt{n}$-rate of estimation even when nuisances are estimated more slowly, but also that our estimator is asymptotically normal. This allows for the construction of confidence bands on the bounds, providing assurance that the true bound is captured. We further show that our asymptotic variance is in fact the minimum variance among all regular and asymptotically linear (RAL) estimators, ensuring semiparametric efficiency. Our second key result is that even if we do not estimate some of the nuisance functions correctly, we are still consistent to sharp or valid bounds. That is, even when we are biased due to misestimation of nuisances, our bias (if any) only enlarges our bounds, so they remain valid. We illustrate these guarantees numerically. Collectively, these guarantees lend substantial credibility to the bounds generated by our method.

Our contributions are summarized as follows:

1. We provide novel algorithms and analysis for learning robust $Q$-functions (Section 3) and robust visitation density ratios (Section 4) under the function approximation setting.
2. We derive the sharp and efficient estimator for the robust policy value, which is optimal in the local-minimax sense and is the gold standard in semiparametric estimation (Section 5).
3. We empirically validate the efficiency and sharpness of our approach (Section 6).

## 1.1 Related Works

**Unobserved Confounding in Sequential Decision-Making.** OPE in robust MDPs is related to OPE bounds in confounded MDPs, where the behavior policy and the transition kernel are influenced by

unobserved confounders. The constraint Eq. (1) that defines our target robust MDP aligns with the Marginal Sensitivity Model (MSM) [66] employed in sensitivity analysis for causal inference. Yet, unlike the MSM, which limits the ratio of policy densities, our approach directly constrains the ratio of the transition kernels. Our formulation can be viewed as a generalization of the MSM from traditional two-action no-horizon causal effects (where the constrains coincide) to multi-action infinite-horizon discounted MDPs, where the next state is the "potential outcome". In that sense, our model essentially serves as an outcome-based sensitivity model [10]. This distinction is crucial as it enables our model to subsume the policy-based MSM in cases where the policy is confounded. Nonetheless, the reverse does not hold, and the policy-based MSM does not imply a transition kernel-based MSM for $A > 2$. This point is further corroborated by [12], who explore the policy-based MSM within confounded MDPs and obtain *non-sharp* identification bounds when $A > 2$. In contrast, our approach yields *sharp* identification in general, regardless of the number of actions and without placing assumptions on the behavior policy, which may or may not be confounded.

[13] also considered an MSM-like model in the transition kernel but their formulation assumes $A = 2$. [40] operates under the setting of [12] and required tabular states. We note that all these works including ours considers *i.i.d.* confounders at each step, which translates to a robust MDP with $(s, a)$-rectangularity and ensures that the worst-case problem is still an MDP rather than a POMDP. The importance of this assumption was verified by [55], who showed that without it, the non-memoryless confounder can create exponential-in-horizon changes in value.

**Neyman Orthogonality and Semiparametric Efficient Estimation.** We leverage a body of research focusing on learning with nuisances functions (e.g., Q-functions) that we need to estimate from data but are not the primary target (e.g., policy value). Much of this research [7, 16, 17, 29, 64, 70, among others] aims to identify Neyman-orthogonal estimators, which are first order orthogonal (insensitive) to nuisance errors. This literature is tightly linked to the semiparametric efficient estimation literature since Neyman-orthogonal scores can arise naturally from efficient influence functions [33, 62]. Going beyond the no-horizon causal inference setting, some explore such estimators in off-policy sequential-decisions contexts [19, 38, 42, 48, 50]. Notably, [39] derive efficient influence functions and orthogonal estimation in standard, non-robust OPE in infinite-horizon RL, which coincides with our unconfounded no-uncertainty case ($\Lambda = 1$).

Moving beyond point-identified settings, some works explore orthogonality and efficiency for partial identification and sensitivity analysis. In the causal inference literature, efficient/orthogonal estimation in the no-horizon setting has been studied extensively under several sensitivity models [10, 18, 24, 58]. Closest to our work is [24] who provide an orthogonal estimator and convergence rates under the MSM [66], which coincides with our setting under $\gamma = 1$. In the sequential setting, [55] considers confounding at a single time step under the MSM, but their estimator is not orthogonal when the quantile function is unknown. [12] provide a fitted-Q-iteration learner with an orthogonalized loss function, but not orthogonal/efficient estimates of worst-case policy value.

## 2 Preliminaries

We consider an MDP with state space $\mathcal{S}$, action space $\mathcal{A}$, transition kernel $P(s' \mid s, a)$, reward function $r(s, a) \in [0, 1]$ and initial state distribution $d_1 \in \Delta(\mathcal{S})$. We do not require $\mathcal{S}$ or $\mathcal{A}$ to be finite. We assume $r$ and $d_1$ are known for simplicity, and it is standard to extend our analysis to when they are unknown. We are given a dataset $\mathcal{D}$ of $n$ *i.i.d.* tuples $(s_i, a_i, r_i, s'_i)$ such that $(s_i, a_i) \sim \nu$, $s'_i \sim P(\cdot \mid s, a)$ and $r_i = r(s_i, a_i)$, where $\nu$ is an arbitrary data-generating distribution. For discount factor $\gamma \in [0, 1)$, let the $Q$ function be the discounted cumulative rewards under a policy $\pi : \mathcal{S} \to \mathcal{A}$, $Q_{\pi,P}(s, a) = \mathbb{E}_{\pi,P}[\sum_{t=0}^{\infty} \gamma^t r_t(s_t, a_t) \mid s_1 = s, a_1 = a]$. Similarly, define the value function as $V_{\pi,P}(s) = Q_{\pi,P}(s, \pi)$, where we use the notation $f(s, \pi) := \mathbb{E}_{a \sim \pi(s)}[f(s, a)]$ for any function $f : \mathcal{S} \times \mathcal{A} \to \mathbb{R}$.

We are interested in estimating the value of a fixed target policy $\pi_t$ (a.k.a. evaluation policy) in an unobserved MDP with a feasible perturbed transition kernel $U$. We say $U$ is a feasible perturbation of the observed, nominal kernel $P$ if for all $s, a, s'$: we have

$$\Lambda^{-1}(s, a) \leq \frac{\mathrm{d}U(s'|s,a)}{\mathrm{d}P(s'|s,a)} \leq \Lambda(s, a) \tag{1}$$

where $\Lambda(s, a) \in [1, \infty)$ is a sensitivity parameter chosen by the practitioner. On the extremes, $\Lambda = 1$ corresponds to no-confounding (*i.e.*, classic OPE setting) and $\Lambda = \infty$ corresponds to maximal-confounding (*i.e.*, worst or best outcome). We denote the set of all feasible perturbations of $P$ by

$\mathcal{U}(P)$, which is an $s, a$-rectangular set [51]. We define the best- and worst-case $Q$ functions of $\pi_t$ as

$$Q^+(s,a) := \sup_{U \in \mathcal{U}(P)} Q_{\pi_t,U}(s,a); \qquad Q^-(s,a) := \inf_{U \in \mathcal{U}(P)} Q_{\pi_t,U}(s,a). \tag{2}$$

Thus, the goal of this paper is to estimate the best- and worst-case value of $\pi_t$ at the initial state,

$$V_{d_1}^\pm := (1-\gamma)\mathbb{E}_{s_1 \sim d_1}[V^\pm(s_1)]. \tag{3}$$

where $V^\pm(s) = \mathbb{E}_{a \sim \pi_t(s)}[Q^\pm(s,a)]$ and the $\pm$ symbol signals that an equation should be read twice, once with $\pm = +$ and once with $\pm = -$. For clarity, we focus the discussion in the main text on estimating the worst-case policy value, $V_{d_1}^-$. We provide a similar analysis for policy values under best-case perturbations ($V_{d_1}^+$) in Appendix B.

Compared to standard OPE, robust OPE is more challenging since the best- and worst-case transition kernels $U^\pm$ are unobserved as our dataset $\mathcal{D}$ is generated under $P$. For example, standard OPE is easy in the on-policy case *i.e.*, if $\mathcal{D}$ were generated by $\pi_t$, but our problem is still "off-data" and non-trivial.

**Discounted Visitation Distributions.** For any transition kernel $U$, define the discounted visitation distribution of $\pi_t$ under $U$ as: $d_{d_1,U}^{\pi_t,\infty}(s) := (1-\gamma)\sum_{h=1}^\infty \gamma^{h-1} d_{d_1,U}^{\pi_t,h}(s)$, where $d_{d_1,U}^{\pi_t,h}(s)$ is the probability of reaching state $s$ in the Markov chain induced by $U$ and policy $\pi_t$ starting from $d_1(\cdot)$. We use $d^{-,\infty}$ as shorthand for $d_{d_1,U^-}^{\pi_t,\infty}$, where $U^-$ denotes the worst-case kernel in $\mathcal{U}(P)$.

**Bellman-type Operators.** For any function $f : \mathcal{S} \times \mathcal{A} \to \mathbb{R}$ and transition kernel $U$, recall the Bellman operator is defined as $\mathcal{T}_U f(s,a) := r(s,a) + \gamma \mathbb{E}_U[f(s',\pi_t) \mid s,a]$. For robust OPE, we define the following robust analog $\mathcal{T}_{rob}^+ f(s,a) := r(s,a) + \gamma \sup_{U \in \mathcal{U}(P)} \mathbb{E}_U[f(s',\pi_t) \mid s,a]$ and $\mathcal{T}_{rob}^- f(s,a) := r(s,a) + \gamma \inf_{U \in \mathcal{U}(P)} \mathbb{E}_U[f(s',\pi_t) \mid s,a]$. Moreover, we define $\mathcal{J}_U f(s,a) := \gamma \mathbb{E}_U[f(s',\pi_t) \mid s,a] - f(s,a)$. For any linear operator $\mathcal{T}$, also let $\mathcal{T}'$ denote its adjoint: that is, for all $f,g \in L_2(\nu)$, $\langle f, \mathcal{T}g \rangle = \langle \mathcal{T}'f, g \rangle$, where $\langle \cdot, \cdot \rangle$ is the inner product in $L_2(\nu)$.

**Conditional Value-at Risk (CVaR).** For a random variable $X$, its upper/lower CVaRs at level $\tau \in [0,1]$ is defined as the average outcome of the upper/lower $\tau$-fraction of cases, and are formally defined as follows [61]:

$$\text{CVaR}_\tau^+(X) := \min_{b \in \mathbb{R}}\{b + \tau^{-1}\mathbb{E}[(X-b)_+]\},$$

$$\text{CVaR}_\tau^-(X) := \max_{b \in \mathbb{R}}\{b + \tau^{-1}\mathbb{E}[(X-b)_-]\},$$

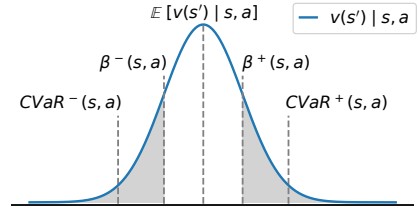

Figure 1: Lower and upper CVaRs and quantiles ($\beta$) of $v(s') \mid s,a$ distribution.

where $y_+ := \max(0,y)$ and $y_- := \min(0,y)$ for $y \in \mathbb{R}$. The optima are attained at the upper/lower $\tau$-th quantile of $X$ which we denote as $\beta_\tau^+(X)/\beta_\tau^-(X)$, *i.e.*,

$$\text{CVaR}_\tau^+(X) := \beta_\tau^+(X) + \tau^{-1}\mathbb{E}[(X-\beta_\tau^+(X))_+], \quad \text{CVaR}_\tau^-(X) := \beta_\tau^-(X) + \tau^{-1}\mathbb{E}[(X-\beta_\tau^-(X))_-].$$

If $X$ has a cumulative distribution function (CDF) which is differentiable at $\beta_\tau^\pm(X)$, its CVaRs simplify to $\text{CVaR}_\tau^+(X) = \mathbb{E}[X \mid X \geq \beta_\tau^+(X)]$ and $\text{CVaR}_\tau^-(X) = \mathbb{E}[X \mid X \leq \beta_\tau^-(X)]$. In the paper, $\tau$ will often be set to $(\Lambda+1)^{-1} \in [0, 0.5]$.

**Notations.** We use $x \lesssim y$ to mean that $x \leq Cy$ holds for some universal constant $C$. The indicator function $\mathbb{I}[p]$ takes value 1 if $p$ is true and 0 otherwise. For a measure $\mu$, we let $\|f\|_\mu := (\mathbb{E}_\mu|f(X)|^2)^{1/2}$ denote the $L_2$ norm of $f$, provided it exists. When $\mu$ is clear from context, we also use $\|f\|_p := (\mathbb{E}|f(X)|^p)^{1/p}$ to denote the $L_p$ norm of $f$ and $\|f\|_{p,n} := (\mathbb{E}_n|f(X)|^p)^{1/p}$ to denote the empirical analog. For a data sample of size $n$, we define the empirical mean as $\mathbb{E}_n[f(X)] = \frac{1}{n}\sum_{i=1}^n f(x_i)$. For a nuisance function $f$, we reserve $f^*$ as its true value and $\widehat{f}$ as the learned value from data. Moreover, we employ $+$ and $-$ to denote functions corresponding to best- and worst-case bounds, respectively. See Appendix A for a comprehensive notation table.

## 2.1 Background: Non-robust OPE

We provide a quick primer on the double RL (DRL) estimator for classic OPE in non-robust MDPs [38], which combines estimates of the $Q$-function and density ratio $w$ to achieve orthogonality, double robustness and semiparametric efficiency. This sets the stage for our orthogonal estimator

in Section 5, which generalizes DRL to robust MDPs by incorporating the robust $Q$-function and density ratio in the worst-case MDP, as described in Section 3 and Section 4 respectively.

The DRL estimator involves two nuisances: (1) $q$, for which the oracle (true value) is the $Q$-function of the target policy $Q^{\pi_t}$, and (2) $w$, for which the oracle is the density ratio of the target policy's visitation distribution and the data distribution $w^{\pi_t} = \mathrm{d}d_{d_1,P}^{\pi_t,\infty}/\mathrm{d}\nu$. In this section, let $\eta = (w, q)$ denote the DRL nuisances (outside this section, we use $\eta$ to denote our robust estimator's nuisances) and let $\eta^\star = (w^{\pi_t}, Q^{\pi_t})$ denote their true values, then the recentered efficient influence function (EIF) of $V_{d_1}^{\pi_t}$ in non-robust MDPs is given by:
$$\psi^{\mathsf{DRL}}(s, a, s'; w, q) = V_{d_1}^{\pi_t} + w(s, a) \cdot (r(s, a) + \gamma q(s', \pi_t) - q(s, a)).$$

The DRL estimator uses cross-fitting to learn nuisances $\widehat{\eta}^{[k]}$ on all data excluding the $k$-th fold $\mathcal{D}^k$, for $k = 1, 2, \ldots, K$ and estimates the OPE value via:
$$\widehat{V}_{d_1}^{\mathsf{DRL}} = \frac{1}{n} \sum_{k=1}^{K} \sum_{(s,a,s') \in \mathcal{D}^k} \psi^{\mathsf{DRL}}(s, a, s'; \widehat{\eta}^{[k]}).$$

As we will see, this paves the way for the EIF of the robust value (Theorem 5.1) and our orthogonal estimator (Algorithm 3). There are two main guarantees for DRL: double robustness and semiparametric efficiency. Let $r_n^w$ and $r_n^q$ be rate functions depending on $n = |\mathcal{D}|$ such that $\|\widehat{q}^{[k]} - Q^{\pi_t}\|_2 \le r_n^q$ and $\|\widehat{w}^{[k]} - w^{\pi_t}\|_2 \le r_n^w$. Then, DRL enjoys $|\widehat{V}_{d_1}^{\mathsf{DRL}} - V_{d_1}^{\pi_t}| \le O_p(n^{-1/2} + r_n^w r_n^q)$, which confers the algorithm double robustness properties. Moreover, if $\Sigma^{\mathsf{ope}}$ is the efficiency bound (*i.e.*, minimum achievable asymptotic variance among RAL estimators in nonparametric models for $(s, a, s')$), then $\sqrt{n}(\widehat{V}_{d_1}^{\mathsf{DRL}} - V_{d_1}^{\pi_t}) \xrightarrow{d} \mathcal{N}(0, \Sigma^{\mathsf{ope}})$. We seek similar guarantees for our orthogonal robust estimator.

# 3 Robust $Q$-Function Estimation with Fitted-$Q$ Evaluation

In this section, we identify the robust $Q$-function using the robust Bellman equation and then derive convergence rates for iteratively minimizing the robust Bellman error.

## 3.1 Identification of the worst-case $Q$-function

The robust worst-case $Q$-function of $\pi_t$, denoted as $Q^-$, satisfies the robust Bellman equation $Q^-(s, a) = \mathcal{T}_{\mathsf{rob}}^- Q^-(s, a), \forall s, a$ since the uncertainty set $\mathcal{U}(P)$ factorizes over $s, a$ [34]. While these equations may seem intractable due to the $\inf$ in the definition of $\mathcal{T}_{\mathsf{rob}}^-$, [12] showed that $\mathcal{T}_{\mathsf{rob}}^-$ has a closed form solution in terms of the CVaR under the *observed* kernel $P$.

**Lemma 3.1.** *Set* $\tau(s, a) = (\Lambda(s, a) + 1)^{-1}$. *Then, for any* $q : \mathcal{S} \times \mathcal{A} \to \mathbb{R}$,
$$\mathcal{T}_{\mathsf{rob}}^- q(s, a) = r(s, a) + \gamma \Lambda^{-1}(s, a) \mathbb{E}[v(s') \mid s, a] + \gamma(1 - \Lambda^{-1}(s, a)) \operatorname{CVaR}_{\tau(s,a)}^-[v(s') \mid s, a],$$
*where* $v(s') = \mathbb{E}_{a' \sim \pi_t(s')}[q(s', a')]$, *and* $\mathbb{E}, \operatorname{CVaR}_\tau$ *are under the observed kernel* $P(\cdot \mid s, a)$.

Lemma 3.1 implies that $Q^-$ is identified via the following equation of observable distributions:
$$Q^-(s, a) = r(s, a) + \gamma \Lambda^{-1}(s, a) \mathbb{E}[Q^-(s', \pi_t) \mid s, a] + \gamma(1 - \Lambda^{-1}(s, a)) \operatorname{CVaR}_{\tau(s,a)}^-[Q^-(s', \pi_t) \mid s, a].$$
Under no confounding ($\Lambda(s, a) = 1$), this recovers the classic Bellman equation.

## 3.2 Estimating the Robust $Q$-Function with Robust FQE

In this section, we estimate $Q^-$ via an iterative fitting algorithm based on fitted Q-evaluation (FQE) [54]. Our algorithm RobustFQE (Algorithm 1) proceeds for $M$ iterations with two main steps in each iteration $i$. First, in Line 5, we estimate the lower-quantile of $\widehat{v}_{i-1}(s') \mid s, a$. Here, we assume access to an oracle QR for quantile regression, which is a well-established problem, allowing for the use of various existing algorithms. Second, in Line 6, we solve the tractable robust Bellman equation in Lemma 3.1 with the CVaR term estimated by its orthogonal estimating equation with the learned quantiles [57]. By orthogonally estimating CVaR, we achieve second-order dependence on the quantile estimation errors from the first step. Next, we minimize the mean squared error using a general function class, $\mathcal{Q} \subset \mathcal{S} \times \mathcal{A} \mapsto [0, (1 - \gamma)^{-1}]$.

To enable convergence guarantees, we make two assumptions. First, we assume that the quantile regression oracle has a specific convergence rate, which can be guaranteed under certain smoothness conditions [9, 14, 27, 28, 52, 60, 65]. Distributional RL may also be modified to learn quantiles of the next state value and have shown benefits in practice [21, 22] and in theory [5, 75, 76, 78].

---

**Algorithm 1** RobustFQE: Iterative fitting for estimating $Q^-$ and $\beta_\tau^-$.

---

1: **Input:** Number of iterations $M$, Dataset $\mathcal{D}$ of size $n$, Q-function class $\mathcal{Q}$.
2: Initialize $\widehat{v}_0^-(s') = 0$.
3: **for** $i = 1, 2, \ldots, M$ **do**
4:     Set $\mathcal{D}_i = \mathcal{D}[ni/M : n(i+1)/M]$.
5:     On the first half of $\mathcal{D}_i$, estimate the $\tau(s,a)$ lower quantile of $\widehat{v}_{i-1}^-(s')$, $s' \sim P(\cdot \mid s,a)$.
    Let $\widehat{\beta}_i^-(s,a)$ denote the learned lower quantiles from the quantile regression oracle QR.
6:     Using the second half of $\mathcal{D}_i$, solve the empirical robust Bellman equation by minimizing squared prediction error for the pseudo-outcome:

$$\widehat{q_i^-} \leftarrow \arg\min_{q \in \mathcal{Q}} \frac{1}{|D_i|/2} \sum_{(s,a,s') \in \mathcal{D}_i[|\mathcal{D}_i|/2+1:]} [(y^-(s,a,s') - q(s,a))^2], \quad \text{where}$$

$$y^-(s,a,s') = r(s,a) + \gamma \Lambda^{-1}(s,a)\widehat{v}_{i-1}^-(s') + \gamma(1 - \Lambda^{-1}(s,a))$$
$$\times (\widehat{\beta}_i^-(s,a) + \tau^{-1}(s,a)(\mathbb{E}_{a' \sim \pi_t(s')}[(\widehat{q_i^-}(s',a') - \widehat{\beta}_i^-(s,a))_-])).$$

7: **Output:** $\widehat{q_M^-}, \widehat{\beta}_M^-$.

---

**Assumption 3.2** (QR Oracle). For any $v : \mathcal{S} \mapsto [0, (1-\gamma)^{-1}]$, let the true $\tau(s,a)$-quantile of $v(s'), s' \sim P(s,a)$ be denoted by $\beta_\tau^v(s,a)$. Given a dataset $\mathcal{D}_{\mathsf{QR}}$, we assume QR outputs estimates $\widehat{\beta}_v$ with bounded $\ell_\infty$ error: for any $\delta$, w.p. $1 - \delta$, $\|\widehat{\beta}_q - \beta_\tau^q\|_\infty < \mathsf{err}_{\mathsf{QR}}(|\mathcal{D}_{\mathsf{QR}}|, \delta)$.

The second assumption is completeness under the robust Bellman $\mathcal{T}_{\mathsf{rob}}^-$. Completeness is a standard assumption in algorithms based on temporal-difference learning and without it, fitted-Q can diverge or converge to suboptimal fixed points [45, 68].

**Assumption 3.3** (Completeness). For all $q \in \mathcal{Q}$, we have $\mathcal{T}_{\mathsf{rob}}^- q \in \mathcal{Q}$.

We note that the current proofs of [12, 59] require a stronger completeness: $\mathcal{T}_\beta q \in \mathcal{Q}$ for all $q \in \mathcal{Q}$ and feasible $\beta$. We circumvent the need for the stronger "all-$\beta$" completeness by bounding model misspecification of least squares regression with second order error in the quantile regression.

Finally, we express our bounds with the critical radius $\varepsilon_n^\mathcal{Q}$, a standard tool for deriving fast rates in statistics; see Appendix D.2 for a summary. Also, we denote the standard concentrability coefficient with $C_{d_1}^- := \big\| \mathrm{d}d_\mu^{-,\infty}/\mathrm{d}d_1 \big\|_\infty$, a standard and necessary quantity for OPE.

**Theorem 3.4.** Let $\varepsilon_n^\mathcal{Q}$ denote the critical radius of $\mathcal{Q}$. Under Assumptions 3.2 and 3.3, RobustFQE ensures that for any $\delta \in (0,1)$, w.p. $1 - \delta$,

$$\|\widehat{q_M^-} - Q^-\|_{d_1} \lesssim (1-\gamma)^{-2}\big(\sqrt{C_{d_1}^-} \cdot \varepsilon_n^\mathcal{Q} + \mathsf{err}_{\mathsf{QR}}^2(n/2M, \delta/2M)\big), \quad \text{and}$$

$$\big|(1-\gamma)\mathbb{E}_{d_1}[\widehat{v}_M^-(s_1)] - V_{d_1}^-\big| \lesssim \gamma^M + (1-\gamma)^{-1}\big(\sqrt{C_{d_1}^-} \cdot \varepsilon_n^\mathcal{Q} + \mathsf{err}_{\mathsf{QR}}^2(n/2M, \delta/2M)\big).$$

For parametric classes (*e.g.*, finite or linear), the critical radius converges at the standard $\widetilde{\mathcal{O}}(n^{-1/2})$ rate. Due to the orthogonal estimation of CVaR, we benefit from a favorable second-order dependence on $\mathsf{err}_{\mathsf{QR}}$ which allows for quantile regression to converge at slower $\widetilde{\mathcal{O}}(n^{-1/4})$ rates. The main disadvantage of this direct approach is that it converges at a slow sub-$\sqrt{n}$ rate if $\varepsilon_n^\mathcal{Q}$ converges at a sub-$\sqrt{n}$, *e.g.*, $\varepsilon_n^\mathcal{Q}$ converges at a $\widetilde{\mathcal{O}}(n^{-1/4})$ rate if $\mathcal{Q}$ is nonparametric with metric entropy at most $1/t^2$ [71]. In Section 5, we present an orthogonal estimator that is both robust to slower rates of $Q$ and achieves semiparametric efficiency.

## 4    Robust $w$-Function Estimation with Minimax Learning

Before we present our orthogonal estimator, we study another essential nuisance function: the robust visitation density ratio, *i.e.*, the robust $w$-function [2, 39]. In this section, we first identify the worst-case transition kernel $U^-$ in our uncertainty set $\mathcal{U}(P)$. Then, we propose a minimax estimator [69] for the robust $w$-function, an important nuisance function for our orthogonal estimator in Section 5.

**Identification of $U^-$.**    The robust transition kernel $U^-$ is defined as the feasible perturbed kernel that achieves the inf in the robust Bellman equation $Q^-(s,a) = \mathcal{T}_{\mathsf{rob}}^- Q^-(s,a)$. Let $F^-(y \mid s,a) =$

---

**Algorithm 2** RobustMIL: Minimax Estimation of $w^{\pm}$ with a Stabilizer

---

1: **Input:** Dataset $\mathcal{D}$, prior stage estimate $\widetilde{\zeta}$, function classes $\mathcal{W}, \mathcal{F}$, stabilizer weight $\lambda > 0$.
2: Define weights $\xi^-(s,a,s') := \Lambda^{-1}(s,a) + (1 - \Lambda^{-1}(s,a))\tau^{-1}(s,a)\mathbb{I}[\widetilde{\zeta}(s,a,s') \leq 0]$.
3: **Output:**

$$\widehat{w}^- = \underset{w \in \mathcal{W}}{\arg\min} \max_{f \in \mathcal{F}} \mathbb{E}_n\big[w(s,a)(\gamma\xi^-(s,a,s')f(s',\pi_t) - f(s,a)) + (1-\gamma)\mathbb{E}_{d_1}f(s_1,\pi_t)\big]$$
$$- \lambda\|\gamma\xi^-(s,a,s';\widetilde{\zeta})f(s',\pi_t) - f(s,a)\|_{2,n}^2 \quad (6)$$

---

$P(V^-(s') \leq y \mid s,a)$ be the next-state pushforward measure of the robust value function $V^-$. Then, $U^-$ is a convex combination of the nominal kernel $P$ and a reweighting of $P$ by an indicator function.

**Lemma 4.1.** *Suppose* $F^-(\beta_\tau^-(s,a) \mid s,a) = \tau$, *where* $\beta_\tau^-(s,a)$ *is the lower* $\tau$-*th quantile of* $F^-(\cdot \mid s,a)$. *Then,*

$$U^-(s' \mid s,a)/P(s' \mid s,a) = \Lambda^{-1}(s,a) + (1 - \Lambda^{-1})\tau(s,a)^{-1}\mathbb{I}[(V^-(s') - \beta_\tau^-(s,a)) \leq 0]. \quad (4)$$

The proof strategy decomposes $U^-$ into its nominal and perturbed components, leveraging the primal solution of $\text{CVaR}_\tau$ [3]; we formalize this in Appendix E.2.

**Identification of** $w^-$. Using the identification of $U^-$ in Lemma 4.1, we can now identify the robust $w$-function based on the Bellman flow equations in the worst-case MDP. The Bellman flow in the robust MDP is given by $d^{-,\infty}(s) = (1-\gamma)d_1(s) + \gamma\mathbb{E}_{\widetilde{s} \sim d^{-,\infty}, \widetilde{a} \sim \pi_t(\widetilde{s})}U^-(s \mid \widetilde{s}, \widetilde{a})$. where $d^{-,\infty}(s)$ was defined in Section 2. Thus, the robust visitation density, defined as $w^-(s) := {}^{\mathrm{d}d^{-,\infty}(s)}/_{\mathrm{d}\nu(s)}$, satisfies the following moment condition for all $f : \mathcal{S} \mapsto \mathbb{R}$:

$$\mathbb{E}[w^-(s)f(s)] = (1-\gamma)\mathbb{E}_{d_1}[f(s_1)] + \gamma\mathbb{E}[w^-(s,a)\mathbb{E}_{s' \sim U^-(s,a)}[f(s')]], \quad (5)$$

where we relaxed notation and defined $w^-(s,a) := w(s) \cdot \pi_t(a \mid s)/\nu(a \mid s)$. As before, in the unconfounded base ($\Lambda = 1$), this result recovers the classic Bellman flow.

## 4.1 Estimating $w^-$ with Robust Minimax Indirect Learning

We now propose a penalized minimax estimator for $w^-$ that generalizes the Minimax Indirect Learning (MIL) of [69] to our robust MDP setting. Our estimator, RobustMIL (Algorithm 2), leverages a general function class $\mathcal{W} \subset \mathcal{S} \times \mathcal{A} \mapsto \mathbb{R}_+$ to approximately solve the moment equation in Eq. (5). It does so by minimizing the difference between the left- and right-hand sides of the equation across a sufficiently large set of adversaries $f$ in a discriminator class $\mathcal{F} \subset \mathcal{S} \times \mathcal{A} \mapsto \mathbb{R}$. Since $U^-$ is unknown, we approximate it via Eq. (4) by plugging in a threshold $\widetilde{\zeta}(s,a,s')$ in the indicator function to approximate the true threshold $\zeta^-(s,a,s') := V^-(s') - \beta_{\tau(s,a)}^-(s,a)$. This yields the minimax objective in Eq. (6), where we also allow for an optional regularization of the adversary's norm which can be useful for obtaining fast convergence rates.

We make the following assumptions for MIL [69]. The first is a regularity condition that (i) our function class has bounded outputs and (ii) $\zeta$ is continuously distributed around the threshold.

**Assumption 4.2** (Regularity). (i) $\sup_{w \in \mathcal{W} \cup \{w^-\}} \|w\|_\infty < \infty$; (ii) the marginal CDF of $V^-(s') - \beta^-(s,a)$, *i.e.*, $F(y) = P(V^-(s') - \beta_{\tau(s,a)}^-(s,a) \leq y)$, is boundedly differentiable around 0.

If next-value distribution is discrete, we can use the discrete form of CVaR and (ii) can be removed.

The second is that the adversary class is rich enough to capture all projected errors under the adjoint of the operator $\mathcal{J}_{U^-}f(s,a) := \gamma\mathbb{E}_{U^-}[f(s',\pi_t) \mid s,a] - f(s,a)$.

**Assumption 4.3** ($w^-$-realizability and completeness). $w^- \in \mathcal{W}$ and $\mathcal{J}'_{U^-}(\mathcal{W} - w^-) \subset \mathcal{F}$.

We note that Assumption 4.3 is monotone in the function class size and can be satisfied by making the function class more expressive, *e.g.*, increasing size of the neural net. Our algorithms are also robust to violations in Assumption 4.3, which we show in Appendix G.

We are now ready to state the main estimation result for $w^-$ in terms of the critical radius (Appendix D.2) of the function class.

---

**Algorithm 3** Orthogonal Estimator for $V_{d_1}^-$

---

1: **Input:** Dataset $\mathcal{D}$, number of splits $K$.
2: **for** $k = 1, 2, \ldots, K$ **do**
3:     Use data $\mathcal{D} \setminus \mathcal{D}_k$ to learn $(q^{-,[k]}, \beta^{-,[k]})$ with Algorithm 1 and $w^{-,[k]}$ with Algorithm 2
4:     **for** $i = \lfloor (k-1)n/K \rfloor, \ldots, \lfloor kn/K \rfloor - 1$ **do** $\psi_i^- = \psi(s_i, a_i, s_i', \widehat{\eta}^-)$
5: **Output:** $\widehat{V}_{d_1}^- = \frac{1}{n} \sum_{i=1}^n \psi_i^-$.

---

**Theorem 4.4.** *Let $\varepsilon_n^{\mathcal{W}}$ denote the maximum critical radii of the following classes:*

$$\mathcal{G}_1 = \{(s, a, s') \mapsto (f(s,a) - \gamma f(s', \pi_t)), f \in \mathcal{F}\},$$

$$\mathcal{G}_2 = \{(s, a, s') \mapsto (w(s,a) - w^-(s,a))(\gamma f(s', \pi_t) - f(s,a)), f \in \mathcal{F}, w \in \mathcal{W}\}.$$

*Under Assumptions 4.2 and 4.3,* RobustMIL *ensures that for any $\delta$, w.p. $1 - \delta$,*

$$\left\| \mathcal{J}_{U^-}'(\widehat{w} - w^-) \right\|_2 \lesssim \varepsilon_n^{\mathcal{W}} + \|\widetilde{\zeta}^- - \zeta^-\|_\infty + \sqrt{\log(1/\delta)/n}.$$

As before, the critical radius $\varepsilon_n^{\mathcal{W}}$ converges at an $\widetilde{\mathcal{O}}(n^{-1/2})$ rate for parametric classes. Notably, our bounds degrade linearly w.r.t. the $\ell_\infty$ error in $\widetilde{\zeta}^-$ for estimating $\zeta^-$. For example, if $\widetilde{\zeta}(s, a, s') = \widehat{v}(s') - \widehat{\beta}(s,a)$ where $\widehat{v}, \widehat{\beta}$ are estimated with RobustFQE, then the $\zeta$-error can be bounded by $\mathcal{O}(\|\widehat{v} - v^-\|_\infty + \|\widehat{\beta} - \beta^-\|_\infty)$. We present the full proof in Appendix G, where we also present a more general result that is robust to misspecifications to realizability and completeness (Assumption 4.3).

## 5   Orthogonal and Efficient Estimator for Robust Policy Value

In this section, we propose an orthogonal estimator that is robust against errors in the nuisances (exhibiting only second-order sensitivity), achieves semiparametric efficiency, and enables inference. Our estimator is based on the efficient influence function (EIF) of $V_{d_1}^-$, which is the canonical gradient of a statistical estimand [67]. The adoption of EIFs for developing efficient estimators is a broadly employed technique in causal inference [16, 43] and reinforcement learning [35, 39].

We define the collection of nuisance parameters by $\eta^- = (w^-, q^-, \beta^-)$. The notation $\widehat{\eta}$ indicates that these functions are estimated from data, while the notation $\eta$ denotes their true values.

**Theorem 5.1** ((Recentered) Efficient Influence Function)**.** *The (R)EIF of $V_{d_1}^-$ is given by:*

$$\psi(s, a, s'; \eta^-) = V_{d_1}^- + w^-(s,a)\big(r(s,a) + \gamma \rho^-(s, a, s'; v^-, \beta^-) - q^-(s,a)\big), \quad \text{where}$$

$$\rho^-(s, a, s'; v^-, \beta^-) = \Lambda(s,a)^{-1} v^-(s') + (1 - \Lambda(s,a)^{-1})\big(\beta^-(s,a) + \tau^{-1}(v^-(s') - \beta^-(s,a))_-\big).$$

*Remark* 5.2. When $\Lambda = 1$, there is no shift in the target environment, and the weight on the CVaR term is zero. The (R)EIF then reduces to the (R)EIF in [39] for regular OPE with an infinite horizon. As $\Lambda \to \infty$, the CVaR term becomes predominant, with the quantile $\beta^-(s,a)$ taking extreme values. This yields the (novel) (R)EIF for the problem in [25], where the expected value term is replaced solely by a CVaR component in the Bellman equation.

The (R)EIF forms the basis of our orthogonal estimator. First, we note that $\mathbb{E}[\psi(s, a, s'; \eta^-)]$ is an unbiased estimator of $V_{d_1}^-$. Furthermore, the expression for $\psi(s, a, s'; \eta^-)$ depends only on quantities $w^-, q^-, \beta^-$ which can be estimated from data. Thus, we can cast the expression $\mathbb{E}[\psi(s, a, s'; \eta^-)]$ as a statistical estimand to be learned from the observed sample. This suggests a natural two-stage estimator that we summarize in Algorithm 3. In the first stage, we estimate the nuisance parameters $\widehat{\eta}$ from the data with $K$-fold cross-fitting; in the second stage, these estimates are incorporated into the (R)EIF expression and we calculate the empirical average using the observed data. We summarize our procedure in Algorithm 3.

The nuisance estimation is detailed in Sections 3.2 and 4.1. The reliance on the EIF confers our estimator desirable statistical properties including a second order bias due to the nuisances, meaning the bias has a product structure with respect to the nuisance errors. Thus, this special structure orthogonalizes away the dependency on $\widehat{Q}^-$ errors which now only appear in second order. Furthermore, our estimator is semiparametrically efficient in the sense that under mild consistency assumptions, it achieves minimum variance among all regular and asymptotically linear (RAL) estimators. We provide theoretical justifications for these properties in the next section.

## 5.1 Theoretical Guarantees of the Orthogonal Estimator

We now characterize the theoretical properties of our orthogonal estimator. We consider the $K$-fold cross-fitted estimator in Algorithm 3 given by

$$\widehat{V}_{d_1}^- = \tfrac{1}{n} \sum_{k=1}^K \sum_{(s,a,s') \in \mathcal{D}^k} \psi(s, a, s'; \widehat{\eta}^{[k]}),$$

where nuisances $\widehat{\eta}^{[k]}, k \in [K]$ are trained on all data excluding the $k^{\text{th}}$ fold $\mathcal{D}^k$. The following theorem outlines the theoretical guarantees of this estimator:

**Theorem 5.3** (Efficiency of $\widehat{V}_{d_1}^-$). *Let $r_{n,p}^w, r_{n,p}^q, r_{n,p}^\beta$ be functions of the same size $n = |\mathcal{D}|$ such that $\|\mathcal{J}'_{U^-}(\widehat{w}^{-,[k]} - w)\|_p \leq r_{n,p}^w$, $\|\widehat{q}^{-,[k]} - q\|_p \leq r_{n,p}^q$, and $\|\beta^{-,[k]} - \beta\|_p \leq r_{n,p}^\beta$ for any $k \in [K]$. Furthermore, assume that the regularity conditions in Assumption 4.2 hold. Then:*

$$|\widehat{V}_{d_1}^- - V_{d_1}^-| \lesssim O_p(n^{-1/2}) + O_p(r_{n,2}^w r_{n,2}^q + (r_{n,\infty}^q)^2 + (r_{n,\infty}^\beta)^2) \qquad \text{(Rates)}$$

*Furthermore, if $r_{n,2}^w \vee r_{n,2}^q = o_p(1)$, $r_{n,2}^w r_{n,2}^q = o_p(n^{-1/2})$, $r_{n,\infty}^q = o_p(n^{-1/4})$, and $r_{n,\infty}^\beta = o_p(n^{-1/4})$, then $\widehat{V}_{d_1}^-$ satisfies:*

$$\sqrt{n}(\widehat{V}_{d_1}^- - V_{d_1}^-) \xrightarrow{d} \mathcal{N}(0, \Sigma), \quad \Sigma = \mathrm{Var}(\psi(s, a, s'; \eta^-)). \qquad \text{(Normality \& Efficiency)}$$

*Moreover, $\Sigma$ is the minimum achievable asymptotic variance among RAL estimators in the nonparametric model for $(s, a, s')$ (the efficiency bound).*

We provide the intuition along with a detailed proof in Appendix H. The first part of Theorem 5.3 implies that as long as we estimate the nuisances at rates faster that $n^{-1/4}$, then we can learn $\widehat{V}_{d_1}^-$ at parametric rates. The second part of Theorem 5.3 states that under mild consistency assumptions, our estimator attains the efficiency bound and is asymptotically normal. That means, for example, we can construct asymptotically valid lower 95%-confidence bound on $\widehat{V}_{d_1}^-$ by simply subtracting 1.64 times $\widehat{\mathrm{se}} = \tfrac{1}{n}(\sum_{k=1}^K \sum_{(s,a,s') \in \mathcal{D}^k} (\psi(s, a, s'; \widehat{\eta}^{[k]}) - \widehat{V}_{d_1}^-)^2)^{1/2}$. Then, we can be sure to have a bound on the worst-case RL policy value, accounting *both* for potential environment shift and finite data. Finally, in Appendix J, we describe two settings when our orthogonal estimator remains valid even if some nuisances are *inconsistent*, which is a desirable guarantee for sensitivity analysis [23].

**Bringing it all together.** We can instantiate Theorem 5.3 with the nuisance estimators from the previous sections. First, use RobustFQE to estimate $\widehat{q}^-$ and $\widehat{\beta}^-$, ensuring $\|\widehat{q}^- - Q^-\|_2 \leq \mathcal{O}(\varepsilon_n^{\mathcal{Q}} + \mathrm{err}_{\mathsf{QR}}^2)$. Under smoothness conditions (Lemma D.2), the $L_2$ guarantee for $\widehat{q}^-$ implies an $L_\infty$ guarantee for $\widehat{q}^-$, which also ensures an $L_\infty$ guarantee for $\widehat{\beta}^-$. This ensures $\max(\|\widehat{q}^- - Q^-\|_\infty, \|\widehat{\beta}^- - \beta^-\|_\infty)$ is well-controlled. Then, we can set $\widetilde{\zeta}^-(s, a, s') = \widehat{q}^-(s', \pi_t) - \widehat{\beta}^-(s, a)$ and run RobustMIL for estimating $\widehat{w}^-$. By Theorem 4.4, its projected-$L_2$ error is $\mathcal{O}(\varepsilon_n^{\mathcal{W}} + \|\widehat{q}^- - Q^-\|_\infty + \|\widehat{\beta}^- - \beta^-\|_\infty)$. Therefore, the final rate via Theorem 5.3 is $\mathcal{O}((\varepsilon_n^{\mathcal{Q}} + \mathrm{err}_{\mathsf{QR}}^2) \cdot \varepsilon_n^{\mathcal{W}} + \|\widehat{q}^- - Q^-\|_\infty^2 + \|\widehat{\beta}^- - \beta^-\|_\infty^2)$.

## 6 Empirical Evaluation

We now provide a proof-of-concept empirical investigation to validate our theoretical findings. We experiment with our proposed methodology in a simple synthetic environment. First, we discuss our environment, followed by our approach for solving for the nuisances functions $\eta^-$. Then, we provide empirical results for our orthogonal estimator, and compare its performance to weighted or direct estimators using the $Q^-$ or $w^-$ nuisances only. The code for our experiments is open-sourced and available at https://github.com/CausalML/adversarial-ope/.

**Experimental Setup** We consider a synthetic MDP with a one-dimensional state and two actions, modeled after a simple control problem with non-deterministic dynamics. The task is to estimate the worst-case policy value $V_{d_1}^-$ of a fixed candidate policy $\pi_t$, across four different constant values of the sensitivity parameter: $\Lambda(s, a) \in \{1, 2, 4, 8\}$.

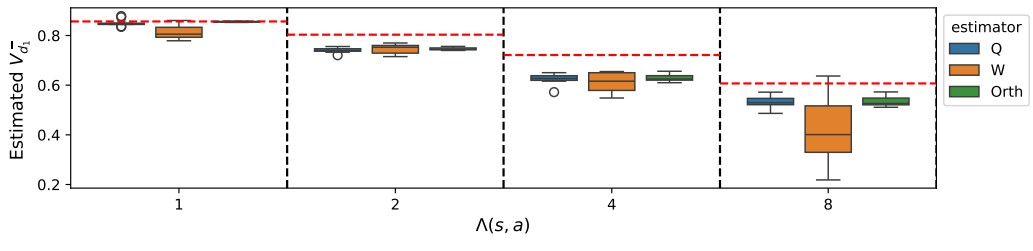

Mean squared error (MSE) to true worst-case policy value

| $\Lambda$ | Q | W | Orth |
|---|---|---|---|
| 1 | $.000240 \pm .000170$ | $.002722 \pm .002266$ | $\mathbf{.000005 \pm .000006}$ |
| 2 | $.004053 \pm .001329$ | $.003584 \pm .002311$ | $\mathbf{.003244 \pm .000600}$ |
| 4 | $.009799 \pm .005172$ | $.013862 \pm .009228$ | $\mathbf{.008721 \pm .002543}$ |
| 8 | $.006247 \pm .003980$ | $.052643 \pm .050839$ | $\mathbf{.005713 \pm .002730}$ |

Figure 2: Results of our synthetic data experiments. We show results for our three estimators on all four $\Lambda$ values, over our 10 experiment replications. **Above:** Box plot summarizing range of policy value estimates for each combination of estimator and $\Lambda$, with Horizontal red dashed lines showing the true worst-case policy values $V_{d_1}^-$. **Below:** Table summarizing the corresponding MSE of these estimators for the true worst-case policy value, along with one standard deviation errors.

We considered three methods for estimating the robust value $V_{d_1}^-$:

1. **Q** (RobustFQE): Direct method using the estimated robust quality function $\widehat{Q}^-$ only.
2. **W** (RobustMIL): Importance-sampling method using the estimated robust density ratio $\widehat{w}^-$ only.
3. **Orth**: Our orthogonal estimator which combines the former two, as described in Algorithm 3.

We performed 10 replications of our experimental procedure, where for each replication we: (1) sampled a dataset of 20,000 tuples using a different fixed logging policy $\pi_b$; (2) fit the nuisance functions $Q^-$, $\beta^-$, and $w^-$ following the method outlined in Algorithms 1 and 2 for each $\Lambda$; and (3) estimated the corresponding robust policy value $V_{d_1}^-$ for all estimators using the fitted nuisances.

**Results**  We summarize our results in Fig. 2. We note that all of our estimators are consistently valid for all values of $\Lambda$ in our experiment. Notably, **Orth** consistently has the lowest mean squared error for the true worst-case policy value. In particular, incorporating the robust importance-sampling weights improves the RobustFQE estimator **Q**, even though these importance-sampling weights by themselves (as in **W**) are much noisier estimators. This is consistent with our theory that the orthogonal estimator is semiparametrically efficient and insensitive to errors in the nuisance functions.

Full experimental details, including our MDP, target/logging policies, methodology for computing the true robust policy values $V_{d_1}^-$, and nuisance estimation, are provided in Appendix K. Finally, we also performed an empirical evaluation in the real-world medical problem of sepsis management using the MIMIC-III dataset [36]. We detail these results in Appendix L.

## 7  Conclusion

We consider the problem of infinite-horizon OPE in RL settings when there can be unknown, but bounded, shifts in the transition distribution compared to the transition distribution generating the data. This can arise due to unobserved confounding, where observed transitions do not reflect the true causal ones, non-stationarity in the environment, or adversarial environments. We propose a sensitivity model for such transition kernel shifts analogous to the classic MSM for static decision making, and provide theoretical guarantees for identifying and estimating the sharp (*i.e.*, tightest possible) bounds on the best/worst-case policy value, as well as the corresponding robust $Q$-function and state density ratio functions. Our estimator for the best/worst-case policy value is orthogonal (insensitive to how the nuisance functions are estimated) and achieves semiparametric efficiency (attaining the best possible asymptotic variance). Finally, our estimator also supports inference, ensuring we can derive reliable bounds for the robust policy value even with finite data.

**Acknowledgements**

We thank the anonymous reviewers for their valuable feedback and insightful suggestions. This material is based upon work supported by the National Science Foundation under Grant Numbers 1846210, IIS-2154711, CAREER 2339395, and by the U.S. Department of Energy, Office of Science, Office of Advanced Scientific Computing Research, under Award Number DE-SC0023112.

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

# Appendices

## A    Notations

Table 1: List of Notations

| | |
|---|---|
| $\mathcal{S}, \mathcal{A}$ | State and action spaces. |
| $\Delta(S)$ | The set of distributions supported by set $S$. |
| $d_1$ | The initial state distribution. |
| $\Lambda(s, a)$ | Tolerance parameter for kernel shift at $(s, a)$. Takes values $[1, \infty]$. |
| $\tau(s, a)$ | $\tau(s, a) = \frac{1}{1 + \Lambda(s,a)} \in [0, \frac{1}{2}]$. |
| $V^\pm, Q^\pm$ | Robust value and quality functions of the target policy $\pi_{\mathrm{t}}$. |
| $f(s, \pi)$ | $f(s, \pi) := \mathbb{E}_{a \sim \pi(s)}[f(s, a)]$. |
| $U^\pm(s' \mid s, a)$ | Robust transition kernel which attains the best- or worst-case value. |
| $\mathcal{T}_U, \mathcal{T}_{\mathsf{rob}}^\pm$ | Bellman operator under $U$ and the robust Bellman operators. |
| $\mathcal{J}_U$ | $\mathcal{J}_U f(s, a) := \gamma \mathbb{E}_U[f(s', \pi_{\mathrm{t}}) \mid s, a] - f(s, a)$ |
| $\beta_\tau^\pm(s, a)$ | The upper $\tau$-th quantile of $V^+(s')$ and lower $\tau$-th quantile of $V^-(s')$, $s' \sim P(s, a)$. |
| $d_{d_1, U}^{\pi_{\mathrm{t}}, \infty}$ | The $\gamma$-discounted average visitation of $\pi_{\mathrm{t}}$ under MDP with transition $U$ starting from $d_1$. |
| $d^{\pm, \infty}$ | $d^{\pm, \infty} = d_{d_1, U^\pm}^{\pi_{\mathrm{t}}, \infty}$. |
| $\nu(s), \nu(s, a)$ | Data generating distribution. $\nu(s)$ marginalizes over actions. |
| $w^\pm$ | $w^\pm = {\mathrm{d}d^{\pm, \infty}}/{\mathrm{d}\nu}$. This is valid both as a function of $s$ or $(s, a)$. |
| $\omega(s, a)$ | $\omega(s, a) = \frac{\pi_{\mathrm{t}}(a \mid s)}{\nu(a \mid s)}$. |
| $x_+, x_-$ | $\max(0, x), \min(0, x)$ respectively, for $x \in \mathbb{R}$. |
| $x \lesssim y$ | $x \leq C y$ for some constant $C$. |
| $\mathbb{E}_n$ | Empirical average over $n$ samples. |
| $\|f\|_p$ | $L^p$ norm, $(\mathbb{E}\|f(X)\|^p)^{1/p}$. |
| $f^\star$ | True (oracle) value of a parameter or function $f$. |
| $f, \bar{f}$ | Putative value of a parameter or function $f$. |
| $\widehat{f}$ | Estimated value of a parameter or function $f$. |

## B    Results for Policy Evaluation Under Best-Case Perturbations

In this section, we present analogous results for the best-case perturbation under the uncertainty set, corresponding to the supremum case of Eq. (2). We derive a similar orthogonal estimator with the properties outlined in Theorem 5.3, following the same reasoning presented in the main text.

$Q^+$ **Identification and Estimation.**    We present the results of Lemma 3.1 for $\mathcal{T}_{\mathsf{rob}}^+$:

$$\mathcal{T}_{\mathsf{rob}}^+ q(s, a) = r(s, a) + \gamma \Lambda^{-1}(s, a) \mathbb{E}[v(s') \mid s, a] + \gamma(1 - \Lambda^{-1}(s, a)) \operatorname{CVaR}_{\tau(s,a)}^+[v(s') \mid s, a].$$

Next, applying Assumption 3.2 and Assumption 3.3 to $\mathcal{T}_{\mathsf{rob}}^+$, we derive from Theorem 3.4 for $Q^-$ that:

$$\|\widehat{q}_M^+ - Q^+\|_{d_1} \lesssim (1 - \gamma)^{-2} (\sqrt{C_{d_1}^+} \cdot \varepsilon_n^{\mathcal{Q}} + \operatorname{err}_{\mathsf{QR}}^2(n/2M, \delta/2M)), \text{ and}$$

$$\left|(1 - \gamma)\mathbb{E}_{d_1}[\widehat{v}_M^+(s_1)] - V_{d_1}^+\right| \lesssim \gamma^M + (1 - \gamma)^{-1} (\sqrt{C_{d_1}^+} \cdot \varepsilon_n^{\mathcal{Q}} + \operatorname{err}_{\mathsf{QR}}^2(n/2M, \delta/2M)).$$

$w^+$ **Identification and Estimation.**    We first state the identification result for $U^-$ as in Lemma 4.1:

$$U^+(s' \mid s, a)/P(s' \mid s, a) = \Lambda^{-1}(s, a) + (1 - \Lambda^{-1})\tau(s, a)^{-1}\mathbb{I}[(V^+(s') - \beta_\tau^+(s, a)) \geq 0].$$

---

**Algorithm 4** Orthogonal Estimator for $V_{d_1}^+$

---

1: **Input:** Dataset $\mathcal{D}$, number of splits $K$.
2: **for** $k = 1, 2, \ldots, K$ **do**
3:     Use data $\mathcal{D} \setminus \mathcal{D}_k$ to learn $(q^{+,[k]}, \beta^{+,[k]})$ with Algorithm 1 and $w^{+,[k]}$ with Algorithm 2
4:     **for** $i = \lfloor (k-1)n/K \rfloor, \ldots, \lfloor kn/K \rfloor - 1$ **do** $\psi_i^+ = \psi(s_i, a_i, s_i', \widehat{\eta}^+)$
5: **Output:** $\widehat{V}_{d_1}^+ = \frac{1}{n} \sum_{i=1}^n \psi_i^+$.

---

Then, under Assumption 4.2 and Assumption 4.3 formulated for $U^+$, the minimax rates from Theorem 4.4 are given by:

$$\left\| \mathcal{J}'_{U^+}(\widehat{w} - w^+) \right\|_2 \lesssim \varepsilon_n^{\mathcal{W}} + \|\widetilde{\zeta}^+ - \zeta^+\|_\infty + \sqrt{\log(1/\delta)/n}.$$

**Orthogonal and Efficient Estimator for $V_{d_1}^+$.** Let the set of nuisance parameters be denoted by $\eta^+ = (w^+, q^+, \beta^+)$. Then, the (recentered) efficient influence function (R)EIF (see Theorem 5.1) for in $V_{d_1}^+$ is formulated as:

$$\psi(s, a, s'; \eta^+) = V_{d_1}^+ + w^+(s, a)\big(r(s, a) + \gamma\rho^+(s, a, s'; v^+, \beta^+) - q^+(s, a)\big), \quad \text{where}$$

$$\rho^+(s, a, s'; v^+, \beta^+) = \Lambda(s, a)^{-1} v^+(s') + (1 - \Lambda(s, a)^{-1})\big(\beta^+(s, a) + \tau^{-1}(v^+(s') - \beta^+(s, a))_+\big).$$

Using this (R)EIF, the orthogonal estimator for $V_{d_1}^+$ is presented in Algorithm 4. We now restate Theorem 5.3 for $\widehat{V}_{d_1}^+$:

**Theorem B.1** (Efficiency of $\widehat{V}_{d_1}^+$). *Let $r_{n,p}^w, r_{n,p}^q, r_{n,p}^\beta$ be functions of $n = |\mathcal{D}|$ such that $\|\mathcal{J}'_{U^+}(\widehat{w}^{+,[k]} - w^*)\|_p \leq r_{n,p}^w$, $\|\widehat{q}^{+,[k]} - q^*\|_p \leq r_{n,p}^q$, and $\|\beta^{+,[k]} - \beta^*\|_p \leq r_{n,p}^\beta$ for any $k \in [K]$. Furthermore, assume that the regularity conditions in Assumption 4.2 hold. Then:*

$$|\widehat{V}_{d_1}^+ - V_{d_1}| \lesssim O_p(n^{-1/2}) + O_p(r_{n,2}^w r_{n,2}^q + (r_{n,\infty}^q)^2 + (r_{n,\infty}^\beta)^2) \tag{Rates}$$

*Furthermore, if $r_{n,2}^w \vee r_{n,2}^q = o_p(1)$, $r_{n,2}^w r_{n,2}^q = o_p(n^{-1/2})$, $r_{n,\infty}^q = o_p(n^{-1/4})$, and $r_{n,\infty}^\beta = o_p(n^{-1/4})$, then $\widehat{V}_{d_1}^+$ satisfies:*

$$\sqrt{n}(\widehat{V}_{d_1}^+ - V_{d_1}) \xrightarrow{d} \mathcal{N}(0, \Sigma), \quad \Sigma = \mathrm{Var}(\psi(s, a, s'; \eta^+)). \tag{Normality \& Efficiency}$$

*Moreover, $\Sigma$ is the minimum achievable asymptotic variance among RAL estimators in the nonparametric model for $(s, a, s')$ (the efficiency bound).*

## C    Additional Related Works

**Robust MDPs.** There is a rich literature on Robust MDPs [30, 34, 51, 80] with $s, a$-rectangular uncertainty sets, but these foundational works assumed knowledge of the transition kernel. Recently, learning-based robust MDP algorithms have been proposed for uncertainty sets under the total variation [47, 59] and more generally $L_p$ balls [46]. These $L_p$ uncertainty sets are additive in nature, *i.e.*, the adversary adds or subtracts a vector in the $\ell_p$ ball to $P(\cdot \mid s, a)$, whereas our uncertainty set is multiplicative in nature, *i.e.*, the adversary can multiply or divide a bounded factor and is more commonly used in causal inference to model unobserved confounding. In the contextual bandit setting, [41] also derived efficiency bounds for robust OPE where both state distribution and reward distributions may shift – their work is however restricted to the one-step bandit setting while our full RL setting is more challenging.

**Risk-Sensitive RL.** Risk-sensitive RL is the problem of optimizing the risk measure of cumulative rewards [32] and is tightly related to robust MDPs [20]. For example, as we proved in Lemma 3.1, the MSM uncertainty set is indeed equivalent to risk-sensitive RL with the dynamic risk measure $\Lambda \mathbb{E} + (1 - \Lambda) \mathrm{CVaR}_\tau$. We note that efficient online RL algorithms have been proposed for similar measures [25, 81]. Static risk-sensitive RL also modifies the Bellman equations in an augmented MDP [74, 77]. Our focus is on deriving the optimal *off-policy evaluation* estimators for the problem, which involves a different set of challenges such as deriving the efficiency bound and ensuring sharpness guarantees even when nuisances are estimated slowly.

# D  Additional Technical Details

## D.1  Higher Order Norms via Smoothness

For any $x \in \mathbb{R}^+$, define $\lfloor x \rfloor$ as the greatest integer that is strictly less than $x$, and let $x$ and $\{x\} = x - \lfloor x \rfloor$ represent the fractional part. Thus, we obtain the distinct decomposition $x = \lfloor x \rfloor + \{x\}$, where $\lfloor x \rfloor \in \mathbb{N}$ and $\{x\} \in (0, 1]$.

**Definition D.1** ($\alpha$-smooth functions). Given $\alpha \in (0, \infty)$ and $\mathcal{X} \subseteq \mathbb{R}^m$, $f : \mathcal{X} \to \mathbb{R}$ is an $\alpha$-smooth function if (1) the mixed derivatives up to $\lfloor \alpha \rfloor$-order exist and are bounded; and (2) all $\lfloor \alpha \rfloor$-order derivatives are $\{\alpha\}$-Hölder continuous [49].

**Lemma D.2** ($L^\infty$ Bound for $\alpha$-Smooth Functions). *Let* $f : \mathcal{X} \to \mathbb{R}, \mathcal{X} \subseteq \mathbb{R}^m$ *be an $\alpha$-smooth function as in Definition D.1. Then, if $\mathcal{X}$ is $\mathbb{R}^m$, a half-space or a bounded Lipschitz domain in $\mathbb{R}^m$, there exists a constant $C$ such the following inequality holds:*

$$\|f\|_\infty \le C \|f\|_p^{\frac{p\alpha}{p\alpha + m}}.$$

*Proof.* This lemma is a direct application of the fractional Gagliardo-Nirenberg interpolation inequality (Theorem 1 in [11]) from the functional analysis literature. For a more comprehensive exposition on this result, see Appendix A.1 in [8]. $\qquad\square$

## D.2  Localized Rademacher Complexity and Critical Radius

Here, we recap the localized Rademacher complexity and critical radius which is a standard complexity measure for obtaining fast rates for squared loss [72]. Let $\mathcal{G}$ be a class of functions $g : \mathcal{Z} \to \mathbb{R}$. Given $n$ datapoints $z_1, z_2, \ldots, z_n$, the empirical localized Rademacher complexity is:

$$\mathcal{R}_n(\varepsilon, \mathcal{G}) := \mathbb{E}_\sigma \left[ \sup_{g \in \mathcal{G} : \|g\|_n \le \varepsilon} \frac{1}{n} \sum_{i=1}^n \epsilon_i g(z_i) \right],$$

where $\mathbb{E}_\sigma$ is expectation over $n$ independent Rademacher random variables $\sigma_1, \sigma_2, \ldots, \sigma_n$, *i.e.*, $\mathbb{E}_\sigma[\cdot] = \frac{1}{2^n} \sum_{\sigma \in \{-1,1\}^n}[\cdot]$. Note that when $\varepsilon = \infty$, there is no localization and $\mathcal{R}_n(\infty, \mathcal{G})$ reduces to the vanilla Rademacher complexity. Let $C := \sup_{g \in \mathcal{G}} \|g\|_\infty$ be the envelope of $\mathcal{G}$. Then, the critical radius of $\mathcal{G}$ with $n$, called $\varepsilon_n$, is the smallest $\varepsilon$ that satisfies $\mathcal{R}_n(\varepsilon, \mathcal{G}) \le \varepsilon^2/C$.

Unless otherwise stated, we will posit that $\mathcal{G}$ is star-shaped: there exists $g_0 \in \mathcal{G}$ such that for all $g \in \mathcal{G}$ and $\alpha \in [0, 1]$, we have $\alpha g_0 + (1 - \alpha)g \in \mathcal{G}$. If not, we can replace $\mathcal{G}$ by its star-hull, *i.e.*, the smallest star-shaped set containing $\mathcal{G}$. We will also posit that $\mathcal{G}$ is symmetric for simplicity.

The critical radius is a well-studied quantity in statistics [72] and also recently in RL [26, 69]. For example if $\mathcal{G}$ has $d$ VC-subgraph dimension, then w.p. $1 - \delta$, $\varepsilon_n \le \mathcal{O}(\sqrt{d \log n / n})$. For nonparametric models with metric entropy at most $1/t^\beta$, the critical radius can also be bounded by $\mathcal{O}(n^{-1/(\max(2+\beta, 2\beta))})$ [69], *e.g.*, is $\mathcal{O}(n^{-1/4})$ if $\beta = 2$.

# E  Proofs for Identification Results

## E.1  Identification of robust $Q$

**Lemma 3.1.** *Set* $\tau(s, a) = (\Lambda(s, a) + 1)^{-1}$. *Then, for any* $q : \mathcal{S} \times \mathcal{A} \to \mathbb{R}$,

$$\mathcal{T}_{\mathsf{rob}}^- q(s, a) = r(s, a) + \gamma \Lambda^{-1}(s, a) \mathbb{E}[v(s') \mid s, a] + \gamma (1 - \Lambda^{-1}(s, a)) \, \mathrm{CVaR}_{\tau(s,a)}^-[v(s') \mid s, a],$$

*where* $v(s') = \mathbb{E}_{a' \sim \pi_t(s')}[q(s', a')]$, *and* $\mathbb{E}, \mathrm{CVaR}_\tau$ *are under the observed kernel* $P(\cdot \mid s, a)$.

*Proof.* Consider the uncertainty set in $\mathcal{T}_{\text{rob}}$ where the constraint on $U$ (Eq. (1)) can be rewritten as:

$$0 \le \frac{U(s'|s,a) - \Lambda^{-1}(s,a)P(s'|s,a)}{P(s'|s,a)} \le \Lambda(s,a) - \Lambda^{-1}(s,a).$$

Therefore, we can write $U(s' \mid s,a) = \Lambda^{-1}(s,a)P(s' \mid s,a) + (1 - \Lambda^{-1})G(s' \mid s,a)$ where we define $G(s' \mid s,a) := \frac{U(s'|s,a) - \Lambda^{-1}(s,a)P(s'|s,a)}{1 - \Lambda^{-1}(s,a)}$. Thus, the constraints on $G$ are that $G(\cdot \mid s,a) \ll P(\cdot \mid s,a)$ and $\|\frac{dG(s'|s,a)}{dP(s'|s,a)}\| \le \Lambda(s,a) + 1$. Setting $\tau(s,a) = \frac{1}{\Lambda(s,a)+1}$, we can apply the primal form of CVaR [3, 24] to obtain

$$\inf_{G \ll P: \|\frac{dG(\cdot|s,a)}{dP(\cdot|s,a)}\|_\infty \le \tau^{-1}(s,a)} \mathbb{E}_G[f(s')] = \text{CVaR}^-_{\tau(s,a)}[f(s') \mid s,a].$$

Therefore, the supremum in $\mathcal{T}_{\text{rob}}$ can be expressed as $\Lambda^{-1}(s,a)$ times the expectation under nominal $P$ and $(1 - \Lambda^{-1}(s,a))$ times the above CVaR expression, which finishes the proof of the $-$ case.

For the $+$ case, we can simply use $\sup$ instead of $\inf$ and upper CVaR instead of lower CVaR. $\quad\square$

## E.2 Identification of robust kernel and visitation

**Lemma 4.1.** *Suppose* $F^-(\beta^-_\tau(s,a) \mid s,a) = \tau$, *where* $\beta^-_\tau(s,a)$ *is the lower $\tau$-th quantile of* $F^-(\cdot \mid s,a)$. *Then,*

$$U^-(s' \mid s,a)/P(s' \mid s,a) = \Lambda^{-1}(s,a) + (1 - \Lambda^{-1})\tau(s,a)^{-1}\mathbb{I}[(V^-(s') - \beta^-_\tau(s,a)) \le 0]. \quad (4)$$

**Lemma E.1.** *Fix any* $v : \mathcal{S} \to \mathbb{R}$ *and define the pushforward* $F_v(y \mid s,a) = P(v(s') \le y \mid s,a)$. *Suppose* $F_v(\beta^\pm_{\tau,F_v(\cdot|s,a)}(s,a) \mid s,a) = \frac{1}{2} \pm (\frac{1}{2} - \tau)$, *where* $\beta^\pm_{\tau,F_v}$ *is the upper/lower $\tau$-quantile of* $F_v$. *Then,* $\sup_{U \in \mathcal{U}(P)} \mathbb{E}_U[v(s') \mid s,a] = \mathbb{E}_{s' \sim U^+_v(s,a)}[v(s')]$ *and* $\inf_{U \in \mathcal{U}(P)} \mathbb{E}_U[v(s') \mid s,a] = \mathbb{E}_{s' \sim U^-_v(s,a)}[v(s')]$, *where*

$$U^\pm_v(s' \mid s,a)/P(s' \mid s,a) = \Lambda^{-1}(s,a) + (1 - \Lambda^{-1})\tau(s,a)^{-1}\mathbb{I}[\pm(v(s') - \beta^\pm_{\tau,F_v(\cdot|s,a)}(s,a)) \ge 0].$$

*Proof.* We start with some intuitions. First, if the CDF of $v(s')$ is differentiable $\beta^+_\tau(s,a)$, then $\text{CVaR}^+_\tau(v(s') \mid s,a) = \mathbb{E}[v(s') \mid f(s') \ge \beta^+_\tau(s,a), s,a]$ and the result follows immediately from Lemma 3.1 by noticing that the form of $U^+$ exactly recovers the convex combination of expectation and CVaR. Alternatively, one can use the closed form solution of the primal CVaR as derived in [3] to obtain the result.

We now provide a formal proof. Fix any $s,a$ and let $\tau = \tau(s,a)$. Fix any function $v(s') \in \mathbb{R}$. We want to show that the worst-case $U^+ = \arg\max_{U \in \mathcal{U}(P)} \mathbb{E}_U[v(s') \mid s,a]$ has a closed form expression as shown in line 725. By the proof of Lemma 3.1 above, we can rewrite $U^+(s' \mid s,a) = \Lambda^{-1}(s,a)P(s' \mid s,a) + (1 - \Lambda^{-1}(s,a))G^+(s' \mid s,a)$, where $G^+ = \arg\max_{G \ll P:|dG(\cdot|s,a)/dP(\cdot|s,a)|_\infty \le \tau^{-1}(s,a)} \mathbb{E}_G[v(s')]$. Thus, it suffices to simplify $G^+$. To do so, we invoke the premise that the CDF of $v(s')$ is differentiable at $\beta^+_\tau$, i.e. $F_v(\beta^+_{\tau,F_v}(s,a) \mid s,a) = 1 - \tau$. This implies that the CVaR is exactly the conditional expectation of the $1 - \tau(s,a)$-fraction of best outcomes, i.e. $\text{CVaR}^+_\tau(v(s') \mid s,a) = \mathbb{E}[v(s') \mid v(s') \ge \beta^+_\tau(s,a), s,a]$, which in turn is equal to $\tau^{-1}\mathbb{E}[v(s')\mathbb{I}[v(s') \ge \beta^+_\tau(s,a)] \mid s,a]$. Thus, $G^+(s' \mid s,a) = \tau^{-1}P(s' \mid s,a)\mathbb{I}[v(s') \ge \beta^+_\tau(s,a)]$. This concludes the proof for the $+$ case. The proof for the $-$ case follows identical steps. $\quad\square$

## F Proofs for Robust FQE

We prove a more general result with approximate completeness, which shows that Theorem 3.4 is robust to approximate completeness.

**Assumption F.1** (Approximate Completeness). $\max_{q \in \mathcal{Q}} \min_{g \in \mathcal{Q}} \|g - \mathcal{T}^\pm_{\text{CVaR}}q\|_\nu \le \varepsilon_{\text{QComp}}$.

**Theorem F.2.** *Assume [Assumption F.1]. Under the same setup as [Theorem 3.4], we have*

$$\left\|\widehat{q}_K^\pm - Q^\pm\right\|_\mu \lesssim \frac{1}{(1-\gamma)^2}(\sqrt{C_\mu^\pm} \cdot (\varepsilon_n^\mathcal{Q} + \varepsilon_{\mathsf{QComp}}) + \mathrm{err}_{\mathsf{QR}}^2(n/2K, \delta/2K)),$$

*and*

$$\left|V_{d_1}^\pm - (1-\gamma)\mathbb{E}_{d_1}[\widehat{q}_K^\pm(s_1, \pi_t)]\right| \lesssim \gamma^K + \frac{1}{1-\gamma}(\sqrt{C_\mu^\pm} \cdot (\varepsilon_n^\mathcal{Q} + \varepsilon_{\mathsf{QComp}}) + \mathrm{err}_{\mathsf{QR}}^2(n/2K, \delta/2K)).$$

*Proof.* Let $U^\pm$ denote the worst-case kernel that satisfies $V_{d_1}^\pm = (1-\gamma)\mathbb{E}_{d_1}V_{U^\pm}^{\pi_t}(s_1)$. Then,

$$
\begin{aligned}
V_{d_1}^\pm - (1-\gamma)\mathbb{E}_{d_1}[\widehat{q}_K^\pm(s_1, \pi_t)] &= (1-\gamma)\mathbb{E}_{d_1}[V_{U^\pm}^{\pi_t}(s_1) - \widehat{q}_K(s_1, \pi_t)] \\
&= \mathbb{E}_{d_{U^\pm}^{\pi,\infty}}[\mathcal{T}_{U^\pm}^{\pi_t}\widehat{q}_K(s,a) - \widehat{q}_K(s,a)] && \text{(Lemma F.3)} \\
&\leq \frac{4}{1-\gamma}\max_{k=1,2,\dots}\left\|\widehat{q}_k - \mathcal{T}_{U^\pm}^{\pi_t}\widehat{q}_{k-1}\right\|_{d_{U^\pm}^{\pi_t,\infty}} + \gamma^{K/2}. && \text{(Lemma F.4)}
\end{aligned}
$$

Consider any $k = 1, 2, \dots$. By definition of $U^\pm$, we have

$$\left\|\widehat{q}_k - \mathcal{T}_{U^\pm}^{\pi_t}\widehat{q}_{k-1}\right\|_{d_{U^\pm}^{\pi_t,\infty}} = \left\|\widehat{q}_k - \mathcal{T}_{\beta_k^\star}^\pm\widehat{q}_{k-1}\right\|_{d^\pm,\infty}, \qquad\qquad \text{(by def of } U^\pm\text{)}$$

where $\beta_k^\star(s,a)$ is the true quantile of $\widehat{v}_{k-1}(s')$. Denote $q_k^\star := \mathcal{T}_{\mathsf{rob}}^\pm\widehat{q}_{k-1}$ and let $\beta_k^\star$ be the true upper/lower quantile of $\widehat{q}_{k-1}$. Recall the population loss function is

$$
\begin{aligned}
L_k(q, \beta) &:= \mathbb{E}\left[\left(y_k^\beta(s,a,s') - q(s,a)\right)^2\right] \\
y_k^\beta(s,a,s') &= r(s,a) + \gamma\Lambda^{-1}(s,a)\widehat{v}_{k-1}(s') \\
&\quad + \gamma(1 - \Lambda^{-1}(s,a))\left(\beta(s,a) + \tau^{-1}(s,a)(\widehat{v}_{k-1}(s') - \beta(s,a))_\pm\right).
\end{aligned}
$$

The empirical loss $\widehat{L}_k(q, \beta)$ is if $\mathbb{E}$ is replaced by $\mathbb{E}_n$. Note that $\widehat{q}_k = \arg\min_{q \in \mathcal{Q}}\widehat{L}_k(q, \widehat{\beta}_k)$.

**Nonparametric Least Squares with Model Misspecification.** We will directly invoke [72, Theorem 13.13], which gives a fast rate for misspecified least squares with general nonparametric classes. We now bound the misspecification. Recall that at the $k$-th iteration, our regression Bayes-optimal is $\mathbb{E}[y_k^{\widehat{\beta}_k}(s,a,s') \mid s, a] = \mathcal{T}_{\widehat{\beta}_k}\widehat{q}_{k-1}(s,a)$. By [Lemma H.3], we know this is close to $\mathcal{T}_{\beta_k^\star}\widehat{q}_{k-1}(s,a)$ with second order errors in $\beta$: for any $\mu$, we have

$$\left\|\mathcal{T}_{\widehat{\beta}_k}^\pm\widehat{q}_{k-1} - \mathcal{T}_{\beta_k^\star}^\pm\widehat{q}_{k-1}\right\|_{d_\mu^\pm,\infty} \lesssim \|\widehat{\beta}_k - \beta_k^\star\|_\infty^2.$$

Finally, by approximate completeness ([Assumption F.1]), there exists $g \in \mathcal{Q}$ such that $\|\mathcal{T}_{\beta_k^\star}\widehat{q}_{k-1}(s,a) - g\| \leq \varepsilon_{\mathsf{QComp}}$. Putting this together: for any $k$, there exists a $g \in \mathcal{Q}$ such that

$$
\begin{aligned}
\|g - \mathcal{T}_{\widehat{\beta}_k}\widehat{q}_{k-1}(s,a)\|_{d_\mu^\pm,\infty} &\leq \|g - \mathcal{T}_{\beta_k^\star}\widehat{q}_{k-1}(s,a)\|_{d_\mu^\pm,\infty} + \|\mathcal{T}_{\beta_k^\star}\widehat{q}_{k-1}(s,a) - \mathcal{T}_{\widehat{\beta}_k}\widehat{q}_{k-1}(s,a)\|_{d_\mu^\pm,\infty} \\
&\leq \sqrt{C_\mu^\pm} \cdot \varepsilon_{\mathsf{QComp}} + \|\widehat{\beta}_k - \beta_k^\star\|_\infty^2.
\end{aligned}
$$

Therefore, [72, Theorem 13.13] (and concentration of least squares) certifies that:

$$\left\|\widehat{q}_k - \mathcal{T}_{\widehat{\beta}_k}\widehat{q}_{k-1}\right\|_{d^\pm,\infty} \lesssim \sqrt{C_\mu^\pm} \cdot \left(\varepsilon_{\mathsf{QComp}} + \varepsilon_n\right) + \|\widehat{\beta}_k - \beta_k^\star\|_\infty^2.$$

Therefore, we have proven:

$$
\begin{aligned}
\left\|\widehat{q}_k - \mathcal{T}_{\beta_k^\star}^\pm\widehat{q}_{k-1}\right\|_{d_\mu^\pm,\infty} &\leq \left\|\widehat{q}_k - \mathcal{T}_{\widehat{\beta}_k}^\pm\widehat{q}_{k-1}\right\|_{d_\mu^\pm,\infty} + \left\|\mathcal{T}_{\widehat{\beta}_k}^\pm\widehat{q}_{k-1} - \mathcal{T}_{\beta_k^\star}^\pm\widehat{q}_{k-1}\right\|_{d_\mu^\pm,\infty} \\
&\lesssim \sqrt{C_\mu^\pm} \cdot \left(\varepsilon_{\mathsf{QComp}} + \varepsilon_n\right) + \|\widehat{\beta}_k - \beta_k^\star\|_\infty^2.
\end{aligned}
$$

This concludes the proof.

$\square$

**Lemma F.3** (Performance Difference). *For any $\pi$, transition kernel $P$, and function $f : \mathcal{S} \times \mathcal{A} \to \mathbb{R}$, we have*

$$V_P^\pi - \mathbb{E}_{s \sim d_1}[f(s, \pi)] = \frac{1}{1-\gamma} \mathbb{E}_{d_P^\pi, \infty}[\mathcal{T}_P^\pi f(s, a) - f(s, a)].$$

*Proof.* See Lemma C.1 of [15]. $\qquad\qquad\square$

**Lemma F.4** (Unrolling). *For any $\pi$, transition kernel $P$, and functions $f_0, f_1, \ldots, f_K : \mathcal{S} \times \mathcal{A} \to \mathbb{R}$ satisfying $f_0(s, a) = 0$, we have $\|f_K - \mathcal{T}_P^\pi f_K\|_{d_P^\pi, \infty} \leq \frac{4}{1-\gamma} \max_{k=1,2,\ldots} \|f_k - \mathcal{T}_P^\pi f_{k-1}\|_{d_P^\pi, \infty} + \gamma^{K/2}$.*

*Proof.* See Lemma C.2 of [15]. $\qquad\qquad\square$

# G   Proofs for Robust Minimax Algorithm

**Assumption G.1** (Approximate $W$-realizability and completeness). Assume the following hold for $\mathcal{W}$ and $\mathcal{F}$:
(A) Approximate realizability: $\min_{w \in \mathcal{W}} \|\mathcal{J}_{U^\pm}(w^\pm - w)\|_2 \leq \varepsilon_{\mathsf{WReal}}$;
(B) Approximate completeness: $\max_{w \in \mathcal{W}} \min_{f \in \mathcal{F}} \|f - \mathcal{J}'_{U^\pm}(w - w^\pm)\|_2 \leq \varepsilon_{\mathsf{WComp}}$.

We prove a more general result with approximate realizability and completeness, which implies Theorem 4.4 that is robust to misspecification in its assumptions.

**Theorem G.2.** *Under Assumption G.1 and the same setup as Theorem 4.4, we have*

$$\left\|\mathcal{J}'_{U^\pm}(\widehat{w} - w^\pm)\right\|_2 \lesssim \varepsilon_n^{\mathcal{W}} + \|\widetilde{\zeta}^\pm - \zeta^\pm\|_\infty + \sqrt{\frac{\log(1/\delta)}{n}} + \varepsilon_{\mathsf{WReal}} + \varepsilon_{\mathsf{WComp}}.$$

*Proof.* For this proof, we focus on the worst-case kernel $P^\star$ of the form $\frac{P^\star(s'|s,a)}{P(s'|s,a)} = \tau^{-1}(s, a)\mathbb{I}[\zeta^\star(s, a, s') \leq 0]$ where $\zeta^\star(s, a, s') = V^-(s') - \beta^-(s, a)$. This corresponds to the pure CVaR case of $\mathcal{T}_{\mathsf{rob}}^-$; the $\mathbb{E}$ part is identical to standard non-robust RL so we omit it. The best-case kernel $U^+$ can be handled similarly. Let $\widehat{P}(s' \mid s, a)$ denote our estimated robust kernel, which satisfies $\frac{\widehat{P}(s'|s,a)}{P(s'|s,a)} = \tau^{-1}(s, a)\mathbb{I}[\widehat{\zeta}(s, a, s') \leq 0]$, where $\widehat{\zeta}(s, a, s')$ is the given prior stage estimate of $\zeta^\star(s, a, s') = V^-(s') - \beta^-(s, a)$.

The key and only difference between our Algorithm 2 and the MIL algorithm ($\widehat{w}_{\mathsf{mil}}$) of [69] is that our next-state samples are importance weighted with $\xi^\pm(s, a, s')$, which is the density ratio of the estimated robust kernel $\widehat{P}(s' \mid s, a)$ and the nominal kernel $P(s' \mid s, a)$. Note also that $\xi^\pm(s, a, s') \leq \tau^{-1}(s, a) < \infty$, and hence $|\mathbb{E}_n[\zeta(s, a, s')f(s')] - \mathbb{E}_{s,a \sim \nu, s' \sim \widehat{P}(s,a)}[f(s')]| \lesssim \sqrt{\log(1/\delta)/n}$ w.p. $1 - \delta$. Therefore, up to $\mathcal{O}(\sqrt{\log(1/\delta)/n})$ errors, our Algorithm 2 can be viewed as MIL applied to the MDP with kernel $\widehat{P}$.

To invoke the result of [69, Theorem 6.1] (in MDP with kernel $\widehat{P}$), we need to show that its assumptions are met by bounding the model misspecification, *i.e.*, Eq. (6) and Appendix C of [69]. Note that these misspecifications are w.r.t. the MDP with kernel $\widehat{P}$, since this is the MDP in which we're applying Theorem 6.1 of [69]. Specifically, the two errors we need to bound are, (A) approximate realizability: $\varepsilon_A = \min_{w \in \mathcal{W}} \|\mathcal{J}'_{\widehat{P}}(w_{\widehat{P}} - w)\|_2$; and (B) approximate completeness: $\varepsilon_B = \max_{w \in \mathcal{W}} \min_{f \in \mathcal{F}} \|f - \mathcal{J}'_{\widehat{P}}(w - w_{\widehat{P}})\|_2$ where recall that $\mathcal{J}_P$ is the linear operator defined as $\mathcal{J}_P f(s, a) := \gamma \mathbb{E}_P[f(s', \pi_{\mathsf{t}}) \mid s, a] - f(s, a)$ and $\mathcal{J}'_P$ is the adjoint.

**Bounding misspecifications by $\|\widehat{\zeta} - \zeta^\star\|_\infty$.** Since $\zeta^\star(s, a, s')$ has a marginal CDF that's boundedly differentiable around 0 (*i.e.*, (ii) of Assumption 4.2), [37, Lemma 3] implies that $\zeta^\star(s, a, s')$ satisfies a 1-margin (Definition H.2). Hence, Lemma H.3 and the continuity of $\zeta^\star(s, a, s')$ implies that

$$\Pr\Big(\mathbb{I}[\widehat{\zeta}(s, a, s') \le 0] \neq \mathbb{I}[\zeta^\star(s, a, s') \le 0]\Big)$$

$$= \Pr\Big((\mathbb{I}[\widehat{\zeta}(s, a, s') \le 0] \neq \mathbb{I}[\zeta^\star(s, a, s') \le 0]), \zeta^\star(s, a, s') \neq 0\Big) \lesssim \|\widehat{\zeta} - \zeta^\star\|_\infty,$$

Thus, for any $v : \mathcal{S} \to \mathbb{R}$,

$$\mathbb{E}\big|(\mathbb{E}_{\widehat{P}} - \mathbb{E}_{P^\star})[v(s') \mid s, a]\big| \le \mathbb{E}[\tau^{-1}(s, a)(\mathbb{I}[\widehat{\zeta}(s, a, s') \le 0] \neq \mathbb{I}[\zeta^\star(s, a, s') \le 0]) \cdot |v(s')|]$$

$$\lesssim \|v\|_\infty \cdot \Pr\Big(\mathbb{I}[\widehat{\zeta}(s, a, s') \le 0] \neq \mathbb{I}[\zeta^\star(s, a, s') \le 0]\Big)$$

$$\lesssim \|v\|_\infty \|\widehat{\zeta} - \zeta^\star\|_\infty,$$

or equivalently

$$\mathbb{E}\|\widehat{P}(\cdot \mid s, a) - P^\star(\cdot \mid s, a)\|_{\mathsf{TV}} \lesssim \|\widehat{\zeta} - \zeta^\star\|_\infty. \tag{7}$$

Equipped with Eq. (7), we can now bound the following two types of errors: (i) $\langle f, (\mathcal{T}_{P^\star} - \mathcal{T}_{\widehat{P}})g\rangle$, and (ii) $\langle w_{\widehat{P}} - w_{P^\star}, h\rangle$, where $f, g : \mathcal{S} \times \mathcal{A} \to \mathbb{R}$ and $h : \mathcal{S} \to \mathbb{R}$, and $\mathcal{T}_P$ and $w_P$ are the Bellman operator and visitation density of target policy $\pi_\mathsf{t}$ in the MDP with kernel $P$.

For (i):

$$\big|\langle f, (\mathcal{J}_{P^\star} - \mathcal{J}_{\widehat{P}})g\rangle\big| = \big|\mathbb{E}[f(s, a)\big(\gamma(\mathbb{E}_{P^\star} - \mathbb{E}_{\widehat{P}})[g(s', \pi_\mathsf{t}) \mid s, a]\big)]\big|$$

$$\le \gamma\|f\|_\infty \mathbb{E}\big|(\mathbb{E}_{P^\star} - \mathbb{E}_{\widehat{P}})[g(s', \pi_\mathsf{t}) \mid s, a]\big|$$

$$\lesssim \gamma\|f\|_\infty \|g(\cdot, \pi_\mathsf{t})\|_\infty \|\widehat{\zeta} - \zeta^\star\|_\infty.$$

For (ii):

$$\langle w_{\widehat{P}} - w_{P^\star}, h\rangle = \mathbb{E}[(w_{\widehat{P}}(s) - w_{P^\star}(s))h(s)]$$

$$\le \|h\|_\infty \|d_{\widehat{P}} - d_{P^\star}\|_{\mathsf{TV}}$$

$$\le \|h\|_\infty \frac{\gamma}{1 - \gamma}\mathbb{E}_{d_{P^\star}}\|\widehat{P}(\cdot \mid s, a) - P^\star(\cdot \mid s, a)\|_{\mathsf{TV}} \qquad \text{(Eq. (9))}$$

$$\lesssim C\|h\|_\infty \frac{\gamma}{1 - \gamma}\mathbb{E}\|\widehat{P}(\cdot \mid s, a) - P^\star(\cdot \mid s, a)\|_{\mathsf{TV}} \qquad \text{(Assumption 4.2(i))}$$

$$\lesssim C\|h\|_\infty \frac{\gamma}{1 - \gamma}\|\widehat{\zeta} - \zeta^\star\|_\infty,$$

where $C = \|{}^{\mathrm{d}d^{P^\star}}\!/{}_{\mathrm{d}\nu}\|_\infty < \infty$.

For approximate realizability ($\varepsilon_A$): for any $w \in \mathcal{W}$, we have

$$\|\mathcal{J}'_{\widehat{P}}(w_{\widehat{P}} - w)\|_2$$

$$\le \|(\mathcal{J}_{\widehat{P}} - \mathcal{J}_{P^\star})'(w_{\widehat{P}} - w)\|_2 + \|\mathcal{J}'_{P^\star}(w_{\widehat{P}} - w_{P^\star})\|_2 + \|\mathcal{J}'_{P^\star}(w^\star - w)\|_2$$

$$= \langle w_{\widehat{P}} - w, (\mathcal{J}_{\widehat{P}} - \mathcal{J}_{P^\star})g_1\rangle + \langle w_{\widehat{P}} - w_{P^\star}, \mathcal{J}_{P^\star}g_2\rangle + \|\mathcal{J}'_{P^\star}(w^\star - w)\|_2$$

$$\lesssim \|\widehat{\zeta} - \zeta^\star\|_\infty + \|\mathcal{J}'_{P^\star}(w^\star - w)\|_2$$

where $g_1 = ((\mathcal{J}_{P^\star} - \mathcal{J}_{\widehat{P}})'(w_{\widehat{P}} - w))/\|(\mathcal{J}_{P^\star} - \mathcal{J}_{\widehat{P}})'(w_{\widehat{P}} - w)\|_2$, $g_2 = (\mathcal{J}'_{P^\star}(w_{\widehat{P}} - w_{P^\star}))/\|\mathcal{J}'_{P^\star}(w_{\widehat{P}} - w_{P^\star})\|_2$. The last inequality uses (i) and (ii) with the fact that $\|g_1\|_\infty < \infty$ and $\|g_2\|_\infty < \infty$ as the $w$ terms are bounded by our premise. Therefore, taking min over $w$ and using Assumption G.1, we have $\varepsilon_A \lesssim \|\widehat{\zeta} - \zeta^\star\|_\infty + \varepsilon_{\mathsf{WReal}}$.

For approximate completeness ($\varepsilon_B$): for any $w \in \mathcal{W}$ and $f \in \mathcal{F}$, we have

$$\|f - \mathcal{J}'_{\widehat{P}}(w - w_{\widehat{P}})\|_2$$

$$\le \|f - \mathcal{J}'_{P^\star}(w - w_{P^\star})\|_2 + \|(\mathcal{J}_{P^\star} - \mathcal{J}_{\widehat{P}})'(w - w_{P^\star})\|_2 + \|\mathcal{J}'_{P^\star}(w_{\widehat{P}} - w_{P^\star})\|_2$$

$$\lesssim \|f - \mathcal{J}'_{P^\star}(w - w_{P^\star})\|_2 + \|\widehat{\zeta} - \zeta^\star\|_\infty,$$

for the same reason as $\varepsilon_A$ as the error terms are the same. Thus, $\varepsilon_B \lesssim \|\widehat{\zeta} - \zeta^\star\|_\infty + \varepsilon_{\mathsf{WComp}}$.

In sum, we have shown that the misspecification is at most $\mathcal{O}(\|\widehat{\zeta} - \zeta^\star\|_\infty + \varepsilon_{\mathsf{WReal}} + \varepsilon_{\mathsf{WComp}})$. Therefore, [69, Theorem 6.1 and Appendix C] ensures that w.p. $1 - \delta$, our learned $\widehat{w}$ satisfies,

$$\left\| \mathcal{J}'_{\widehat{P}}(\widehat{w} - w_{\widehat{P}}) \right\|_2 \lesssim \varepsilon_n^{\mathcal{W}} + \|\widehat{\zeta} - \zeta^\star\|_\infty + \varepsilon_{\mathsf{WReal}} + \varepsilon_{\mathsf{WComp}} + \sqrt{\log(1/\delta)/n}.$$

**Concluding the proof.** The final step is to translate the above guarantee to $\|\mathcal{J}'_{P^\star}(\widehat{w} - w_{P^\star})\|_2$. The following shows that the switching cost is $\mathcal{O}(\|\widehat{\zeta} - \zeta^\star\|_\infty)$ as before:

$$\begin{aligned}
&\|\mathcal{J}'_{P^\star}(\widehat{w} - w_{P^\star})\|_2 \\
&\leq \|(\mathcal{J}_{P^\star} - \mathcal{J}_{\widehat{P}})'(\widehat{w} - w_{P^\star})\|_2 + \|\mathcal{J}'_{\widehat{P}}(\widehat{w} - w_{\widehat{P}})\|_2 + \|\mathcal{J}'_{\widehat{P}}(w_{\widehat{P}} - w_{P^\star})\|_2 \\
&\lesssim \varepsilon_n^{\mathcal{W}} + \|\widehat{\zeta} - \zeta^\star\|_\infty + \varepsilon_{\mathsf{WReal}} + \varepsilon_{\mathsf{WComp}} + \sqrt{\log(1/\delta)/n}.
\end{aligned}$$

This concludes the proof. $\qquad\square$

**Lemma G.3** (Visitation performance-difference). *Let $P, U : \mathcal{S} \to \mathbb{R}_+$ be non-negative measures, which should be thought of as transitions in a discounted Markov chain. Assume $U$ satisfies $\sum_{s'} U(s' \mid s) \leq 1$. Define $d_U = (1 - \gamma) \sum_{h=1}^\infty \gamma^{h-1} d_U^h$, where $d_U^h = \int_{s_1, s_2, \ldots, s_{h-1}} d_1(s_1) U(s_2 \mid s_1) \ldots U(s \mid s_{h-1}) \mathrm{d}s_{1:h-1}$. Assume the same for $P$.*

*Let $\mathcal{F} \subset \mathcal{S} \to \mathbb{R}$ be a function class that satisfies $f \in \mathcal{F} \implies g(s) = \mathbb{E}_{s' \sim P(s)}[f(s')] \in \mathcal{F}$, i.e., closed under projection with $P$. Then, define the integral (probability) metric $\|P - U\|_{\mathcal{F}} := \sup_{f \in \mathcal{F}} |(\mathbb{E}_P - \mathbb{E}_U)[f(s)]|$. Then we have,*

$$\|d_P - d_U\|_{\mathcal{F}} \leq \frac{\gamma}{1 - \gamma} \mathbb{E}_{d_U} \|P(\cdot \mid s) - U(\cdot \mid s)\|_{\mathcal{F}}. \tag{8}$$

*Proof.* Recall Bellman's flow, which is $d_P(s) = (1 - \gamma)d_1(s) + \gamma \mathbb{E}_{\widetilde{s} \sim d_P} P(s \mid \widetilde{s})$. Fix any $f \in \mathcal{F}$. The initial state distributions cancel, so we have,

$$\begin{aligned}
&|(\mathbb{E}_{d_P} - \mathbb{E}_{d_U})[f(s)]| \\
&= \left| \gamma \mathbb{E}_{\widetilde{s} \sim d_P} \mathbb{E}_{s \sim P(\cdot \mid \widetilde{s})}[f(s)] - \gamma \mathbb{E}_{\widetilde{s} \sim d_U} \mathbb{E}_{s \sim U(\cdot \mid \widetilde{s})}[f(s)] \right| \\
&\leq \left| \gamma \mathbb{E}_{\widetilde{s} \sim d_P} \mathbb{E}_{s \sim P(\cdot \mid \widetilde{s})}[f(s)] - \gamma \mathbb{E}_{\widetilde{s} \sim d_U} \mathbb{E}_{s \sim P(\cdot \mid \widetilde{s})}[f(s)] \right| \\
&\quad + \left| \gamma \mathbb{E}_{\widetilde{s} \sim d_U} \mathbb{E}_{s \sim P(\cdot \mid \widetilde{s})}[f(s)] - \gamma \mathbb{E}_{\widetilde{s} \sim d_U} \mathbb{E}_{s \sim U(\cdot \mid \widetilde{s})}[f(s)] \right| \\
&\leq \gamma \left| (\mathbb{E}_{\widetilde{s} \sim d_P} - \mathbb{E}_{\widetilde{s} \sim d_U})[\mathbb{E}_{s \sim P(\cdot \mid \widetilde{s})} f(s)] \right| + \gamma \mathbb{E}_{\widetilde{s} \sim d_U} \left| (\mathbb{E}_{s \sim P(\cdot \mid \widetilde{s})} - \mathbb{E}_{s \sim U(\cdot \mid \widetilde{s})})[f(s)] \right|.
\end{aligned}$$

Thus, taking supremum over $\mathcal{F}$, we have

$$\begin{aligned}
&\|d_P - d_U\|_{\mathcal{F}} \\
&\leq \gamma \sup_{f \in \mathcal{F}} \left| (\mathbb{E}_{\widetilde{s} \sim d_P} - \mathbb{E}_{\widetilde{s} \sim d_U})[\mathbb{E}_{s \sim P(\widetilde{s})} f(s)] \right| + \gamma \mathbb{E}_{\widetilde{s} \sim d_U} \sup_{f \in \mathcal{F}} \left| (\mathbb{E}_{s \sim P(\cdot \mid \widetilde{s})} - \mathbb{E}_{s \sim U(\cdot \mid \widetilde{s})})[f(s)] \right| \\
&= \gamma \|d_P - d_U\|_{\mathcal{F}} + \gamma \mathbb{E}_{\widetilde{s} \sim d_U} \|P(\cdot \mid \widetilde{s}) - U(\cdot \mid \widetilde{s})\|_{\mathcal{F}}. \qquad (\mathcal{F} \text{ closed under } P\text{-projection})
\end{aligned}$$

Rearranging terms finishes the proof. $\qquad\square$

If $\mathcal{F}$ is the class of functions with $\|f\|_\infty \leq 1$, then this recovers the TV distance, which gives,

$$\|d_P - d_U\|_{\mathsf{TV}} \leq \frac{\gamma}{1 - \gamma} \mathbb{E}_{d_U} \|P(\cdot \mid s) - U(\cdot \mid s)\|_{\mathsf{TV}}. \tag{9}$$

This generalizes Lemma E.3 of [1] to infinite horizon.

# H    Proofs and Additional Details for the Orthogonal Estimator

## H.1    Intuition for Theorem 5.3

We provide some intuition for the results in Theorem 5.3. Consider the $V^-$ bound and let us decouple the indicator $\mathbb{I}[v(s') - \beta(s,a) \leq 0]$ that appears implicitly in the $(v^-(s') - \beta^-(s,a))_-$ notation of Theorem 5.1. We augment the set of nuisances with $\zeta(s,a,s') = v^-(s') - \beta^-(s,a)$ such that $(v^-(s') - \beta^-(s,a))_- = (v^-(s') - \beta^-(s,a))\mathbb{I}[\zeta(s,a,s') \leq 0]$. We state the following lemma (which we elaborate upon in Lemmas H.4 and H.5 in the Appendix):

**Lemma H.1** (Double sharpness with correct $\zeta^\star$)**.** *Let $\mathbb{E}[\psi(s,a,s';q,w,\beta,\zeta^\star)]$ be the expectation of the (R)EIF with an arbitrary nuisance set $\eta = (w,q,\beta)$, but where the indicator $\mathbb{I}[v^-(s') \leq \beta^-(s,a)]$ has been replaced with the correct indicator $\mathbb{I}[\zeta^\star(s,a,s') \leq 0]$. Then:*

$$V_{d_1}^- = \mathbb{E}[\psi(s,a,s';q,w^\star,\beta^\star,\zeta^\star)] = \mathbb{E}[\psi(s,a,s';q^\star,w,\beta^\star,\zeta^\star)]$$

This lemma implies that if $\beta^- = (\beta^*)^-$ and $\zeta = \zeta^*$, then the estimator $\widehat{V}_{d_1}^-$ has a property known as "double-robustness" [43] or "double-sharpness" [24] in $q$ and $w$, meaning the bias vanishes when either $q$ or $w$ is consistent. Moreover, the convergence rate would be $O_p(r_{n,2}^w r_{n,2}^q)$. This condition holds provided that $\beta$ and $\zeta$ are correctly specified. However, estimation errors in $\beta$ introduce an additional $O_p\left((r_{n,\infty}^\beta)^2\right)$ term, reflecting that $\beta$ is first-order optimal for the CVaR component. Additionally, discrepancies between $\zeta$ and $\zeta^*$ contribute an extra $O_p\left((r_{n,\infty}^q)^2\right)$ to the error. While this discussion gives some insight into how we achieve the results in Theorem 5.3, we provide a a rigorous analysis in the next section.

## H.2    Preliminaries

For this proof, our focus will be on $\widehat{V}_{d_1}^-$. The argument for $\widehat{V}_{d_1}^+$ is analogous, following a symmetric approach. To improve the clarity of our exposition, we will omit the $-$ and $\tau$ indices, assuming their presence is clear from the context.

For simplicity, we assume that $n$ is a multiple of $K$ such that $n = Kn_K$, where $n_K$ is the size of a fold. We let $\mathbb{E}_n, \mathbb{E}_k$ denote the empirical averages over the entire sample and the $k^{\text{th}}$ fold, respectively. Recall that we use $\widehat{\eta} = (\widehat{w}, \widehat{q}, \widehat{\beta})$ and $\eta^* = (w^*, q^*, \beta^*)$ to denote the estimated and oracle nuisances, respectively.

We further suppress the dependency on $s, a$ in $\Lambda$ and $\tau$ and we write the $\rho$ term in Theorem 5.1 as

$$\rho(s,a,s';v,\beta) = (1-\lambda)v(s') + \lambda\big(\beta(s,a) + \tau^{-1}(v(s') - \beta(s,a))_-\big). \tag{10}$$

We justify this by noting that the analysis holds regardless of whether $\lambda$ and $\tau$ depend on $s, a$. Sometimes, it will be useful to decouple the indicator $\mathbb{I}[v(s') - \beta(s,a) \leq 0]$ implicit in the definition of $\rho$. In this case, we augment the set of nuisances with $\zeta(s,a,s') = v(s') - \beta(s,a)$ and write $\rho$ as

$$\rho(s,a,s';v,\beta,\zeta) = (1-\lambda)v(s') + \lambda\big(\beta(s,a) + \tau^{-1}(v(s') - \beta(s,a))\mathbb{I}[\zeta(s,a,s') \leq 0]\big). \tag{11}$$

Similarly define $\psi(\cdot; w, q, \beta, \zeta)$ with the $\rho(\cdot; v, \beta, \zeta)$.

## H.3    Auxiliary Lemmas

**Definition H.2** (Margin Condition)**.** A function $f : \mathcal{X} \to \mathbb{R}$ of some random variable $X$ is said to satisfy the margin condition with sharpness $\alpha \in [0, \infty]$ (or more succinctly, an $\alpha$-margin) if there exist a fixed constant $c > 0$ such that

$$\forall t > 0 : P(0 < |f(X)| \leq t) \leq ct^\alpha.$$

If $f(X)$ is either zero or bounded away from zero almost surely, then $f$ satisfies an infinite margin, *i.e.*, $\alpha = \infty$ [37, Lemma 2]. If $f(X)$ is continuously distributed in a neighborhood around 0, *i.e.*,

its CDF is boundedly differentiable on $(-\varepsilon, 0) \cup (0, \varepsilon)$ for some $\varepsilon > 0$, then $f$ has a 1-margin [37, Lemma 3].

**Lemma H.3** (Margin Guarantees). *For any $f : \mathcal{X} \to \mathbb{R}$ satisfying $\alpha$-margin (Definition H.2), $p \in [1, \infty]$, and any $g : \mathcal{X} \to \mathbb{R}$, the following statements hold for some constant $C > 0$:*

$$\mathbb{E}[(\mathbb{I}[g(X) \leq 0] - \mathbb{I}[f(X) \leq 0])f(X)] \leq C\|f - g\|_p^{\frac{p(1+\alpha)}{p+\alpha}}, \tag{12}$$

$$P[\mathbb{I}[g(X) \leq 0] \neq \mathbb{I}[f(X) \leq 0], f(X) \neq 0] \leq C\|f - g\|_p^{\frac{p\alpha}{p+\alpha}}, \tag{13}$$

*where $\|\cdot\|_p$ is the $L^p$ norm and we set $\infty t/\infty = t$ in the exponents.*

The proof of Eq. (12) for any $p \in [1, \infty]$ and of Eq. (13) for $p = \infty$ is given in [4, Lemmas 5.1 and 5.2]. The proof of Eq. (13) for $p < \infty$ is given in [37, Lemma 5].

**Lemma H.4** (Sharpness with correct $q^\star$ and $\beta^\star$). $\frac{1}{n}\sum_{(s,a,s')\sim\mathcal{D}}\psi(s, a, s'; w, q, \beta)$ *is an unbiased estimator of $V_{d_1}^\star$ when $q = q^\star, \beta = \beta^\star$, i.e.,*

$$(1 - \gamma)\mathbb{E}_{d_1}v^\star(s_1) = \mathbb{E}[\psi(s, a, s'; w, q^\star, \beta\star)].$$

*Proof.* Since $q^\star$ and $\beta^\star$ are correct, the robust Bellman equation holds, and so for every $s, a$,

$$\mathbb{E}\big[(1 - \lambda)v^\star(s') + \lambda(\beta^\star(s, a) + \tau^{-1}(v^\star(s') - \beta^\star(s, a))_-) \mid s, a\big] = 0.$$

Thus, multiplying by any $w$ does not change the fact that the debiasing term in $\psi$ has expectation zero. Since we have $v^\star$, the first term in $\psi$ is exactly the estimand, which concludes the proof. $\square$

**Lemma H.5** (Sharpness with correct $w^*$ and $\zeta^*$). $\frac{1}{n}\sum_{(s,a,s')\sim\mathcal{D}}\psi(s, a, s'; w, q, \beta, \zeta)$ *is an unbiased estimator of $V_{d_1}^\star$ when $w = w^\star, \zeta = \zeta^\star$, i.e.,*

$$(1 - \gamma)\mathbb{E}_{d_1}v^\star(s_1) = \mathbb{E}[\psi(s, a, s'; q, w^\star, \beta, \zeta^\star)]$$

*Proof.* Let $P^\star$ denote the robust transition kernel and let $d^\star$ denote the robust visitation measure under $\pi$, which satisfies: for all functions $f$,

$$\mathbb{E}_{d^\star}[f(s, a)] = (1 - \gamma)\mathbb{E}_{d_1}f(s, \pi) + \gamma\mathbb{E}_{\widetilde{s}, \widetilde{a}\sim d^\star, s\sim P^\star(s,a)}[f(s, \pi)].$$

Since $\zeta^\star$ is correct, for any $v, s, a$, we have

$$\mathbb{E}_{s'\sim P(s,a)}\big[(1 - \lambda)v(s') + \lambda\big(\beta(s, a) + \tau^{-1}(v(s') - \beta(s, a))\mathbb{I}[\zeta^\star(s, a, s') \leq 0]\big)\big]$$
$$= \mathbb{E}_{s'\sim P(s,a)}\big[(1 - \lambda)v(s') + \lambda\tau^{-1}v(s')\mathbb{I}[\zeta^\star(s, a, s') \leq 0]\big] \tag{$\bigstar$}$$
$$= \mathbb{E}_{s'\sim P^\star(s,a)}[v(s')], \tag{Lemma 4.1}$$

where in $\bigstar$ we used $\mathbb{E}_{s'\sim P(s,a)}\big[\beta(s, a)\big(1 - \tau^{-1}\mathbb{I}[\zeta^\star(s, a, s') \leq 0]\big)\big] = \beta(s, a)\big(1 - \tau^{-1}\tau\big) = 0$. That is, for all function $f$, we have

$$(1 - \gamma)\mathbb{E}_{d_1}v(s_1) + \mathbb{E}[w^\star(s, a)(r(s, a) + \gamma\rho(s, a, s'; v, \beta, \zeta^\star) - q(s, a))]$$
$$= (1 - \gamma)\mathbb{E}_{d_1}v(s_1) + \mathbb{E}_{s,a\sim d^\star}[r(s, a) + \gamma\rho(s, a, s'; v, \beta, \zeta^\star) - q(s, a)]$$
$$= \mathbb{E}_{s,a\sim d^\star}[r(s, a)] + (1 - \gamma)\mathbb{E}_{d_1}v(s_1) + \mathbb{E}_{s,a\sim d^\star}\big[\gamma\mathbb{E}_{s'\sim P^\star(s,a)}[v(s')] - q(s, a)\big]$$
$$= \mathbb{E}_{s,a\sim d^\star}[r(s, a)] \qquad\qquad \text{(robust Bellman flow)}$$
$$= (1 - \gamma)\mathbb{E}_{d_1}v^\star(s_1).$$

This concludes the proof. $\square$

## H.4  Proof of Rates

The estimation error is given by:

$$|\widehat{V}_{d_1} - V_{d_1}^*| = \left|\frac{1}{K}\sum_{k=1}^K \mathbb{E}_k[\psi(s, a, s'; \widehat{\eta}^{[k]})] - V_{d_1}^*\right| \leq \frac{1}{K}\sum_{k=1}^K \left|\mathbb{E}_k[\psi(s, a, s'; \widehat{\eta}^{[k]})] - V_{d_1}^*\right|$$

We wish need to bound $\left| \mathbb{E}_k[\psi(s, a, s'; \widehat{\eta}^{[k]})] - V_{d_1}^* \right|$. We have that:

$$\left| \mathbb{E}_k[\psi(s, a, s'; \widehat{\eta}^{[k]})] - V_{d_1}^* \right| \leq \left| \mathbb{E}_k[\psi(s, a, s'; \widehat{\eta}^{[k]})] - \mathbb{E}[\psi(s, a, s'; \widehat{\eta}^{[k]})] \right| + \left| \mathbb{E}[\psi(s, a, s'; \widehat{\eta}^{[k]})] - V_{d_1}^* \right|$$

The first term is $O_p(n^{-1/2})$ by the CLT. We are now interested in bounding the second term:

$$\varepsilon(\widehat{\eta}) := \left| \mathbb{E}[\psi(s, a, s'; \widehat{\eta})] - V_{d_1}^* \right|. \tag{14}$$

where we dropped the $[k]$ indicator without loss of generality. We further decompose $\varepsilon(\widehat{\eta})$ into two error terms, $\varepsilon_A$ and $\varepsilon_B$, as follows:

$$\varepsilon(\widehat{\eta}) = \left| \mathbb{E}\Big[ \psi(s, a, s'; \widehat{q}, \widehat{w}, \widehat{\beta}) \Big] - \mathbb{E}\Big[ \psi(s, a, s'; \widehat{q}, w^\star, \widehat{\beta}, \zeta^\star) \Big] \right| \qquad \text{(Lemma H.5)}$$

$$\leq \left| \mathbb{E}\Big[ \psi(s, a, s'; \widehat{q}, \widehat{w}, \widehat{\beta}) \Big] - \mathbb{E}\Big[ \psi(s, a, s'; \widehat{q}, \widehat{w}, \widehat{\beta}, \zeta^\star) \Big] \right| \qquad (\varepsilon^A)$$

$$+ \left| \mathbb{E}\Big[ \psi(s, a, s'; \widehat{q}, \widehat{w}, \widehat{\beta}, \zeta^\star) \Big] - \mathbb{E}\Big[ \psi(s, a, s'; \widehat{q}, w^\star, \widehat{\beta}, \zeta^\star) \Big] \right|. \qquad (\varepsilon^B)$$

**Bounding $\varepsilon^A$: Error from the incorrect indicator $\zeta$.**

$$\varepsilon_A = \gamma\lambda\tau^{-1}\mathbb{E}\widehat{w}(s, a)\Big(\widehat{v}(s') - \widehat{\beta}(s, a)\Big)\Big(\mathbb{I}\left[\widehat{v}(s') - \widehat{\beta}(s, a) \leq 0\right] - \mathbb{I}\left[v^\star(s') - \beta^\star(s, a) \leq 0\right]\Big)$$

$$\leq C\gamma\lambda\tau^{-1}\mathbb{E}\Big(\widehat{v}(s') - \widehat{\beta}(s, a)\Big)\Big(\mathbb{I}\left[\widehat{v}(s') - \widehat{\beta}(s, a) \leq 0\right] - \mathbb{I}\left[v^\star(s') - \beta^\star(s, a) \leq 0\right]\Big)$$
$$\text{(Assumption 4.2)}$$

$$\lesssim \mathbb{E}\Big(\widehat{v}(s') - \widehat{\beta}(s, a)\Big)\Big(\mathbb{I}\left[\widehat{v}(s') - \widehat{\beta}(s, a) \leq 0\right] - \mathbb{I}\left[v^\star(s') - \beta^\star(s, a) \leq 0\right]\Big)$$

We break these terms down as follows:

$$\mathbb{E}\Big(\widehat{v}(s') - \widehat{\beta}(s, a)\Big)\Big(\mathbb{I}\left[\widehat{v}(s') - \widehat{\beta}(s, a) \leq 0\right] - \mathbb{I}\left[v^\star(s') - \beta^\star(s, a) \leq 0\right]\Big)$$

$$= \mathbb{E}(v^\star(s') - \beta^\star(s, a))\Big(\mathbb{I}\left[\widehat{v}(s') - \widehat{\beta}(s, a) \leq 0\right] - \mathbb{I}\left[v^\star(s') - \beta^\star(s, a) \leq 0\right]\Big) \qquad (\varepsilon_1^A)$$

$$+ \mathbb{E}\Big(\widehat{v}(s') - \widehat{\beta}(s, a) - v^\star(s') + \beta^\star(s, a)\Big)\Big(\mathbb{I}\left[\widehat{v}(s') - \widehat{\beta}(s, a) \leq 0\right] - \mathbb{I}\left[v^\star(s') - \beta^\star(s, a) \leq 0\right]\Big).$$
$$(\varepsilon_2^A)$$

We first bound $\varepsilon_1^A$. Assumption 4.2 implies

$$P(0 < |v^\star(s') - \beta^\star(s, a)| \leq t) \leq c'' t, \ \forall t \in [0, c'), \quad P(|v^\star(s') - \beta^\star(s, a)| = 0) = 0,$$

where $c' < 1$ is the min of 1 and the given neighborhood of zero and $c'' \geq 1$ is the max of 1 and the bound on the density in that neighborhood. This implies a margin condition with $\alpha = 1$ and $c = c''/c'$.

We can instantiate the first part of Lemma H.3 with $f(X) = v^\star(s') - \beta^\star(s, a)$, $g(X) = \widehat{v}(s') - \widehat{\beta}(s, a)$ and obtain

$$\varepsilon_1^A \lesssim \left\| v^\star(s') - \beta^\star(s, a) - \widehat{v}(s') + \widehat{\beta}(s, a) \right\|_p^{\frac{2p}{p+1}}$$

$$\leq \|\widehat{v}(s') - v^\star(s')\|_p^{\frac{2p}{p+1}} + \left\| \widehat{\beta}(s, a) - \beta^\star(s, a) \right\|_p^{\frac{2p}{p+1}}.$$

To bound $\varepsilon_2^A$, first write

$$\left| \mathbb{E}\Big(\widehat{v}(s') - \widehat{\beta}(s, a) - v^\star(s') + \beta^\star(s, a)\Big)\Big(\mathbb{I}\left[\widehat{v}(s') - \widehat{\beta}(s, a) \leq 0\right] - \mathbb{I}\left[v^\star(s') - \beta^\star(s, a) \leq 0\right]\Big) \right|$$

$$\leq \left\| \widehat{v}(s') - \widehat{\beta}(s, a) - v^\star(s') + \beta^\star(s, a) \right\|_p$$

$$\cdot \mathbb{P}\Big( \mathbb{I}\left[\widehat{v}(s') - \widehat{\beta}(s, a) \leq 0\right] \neq \mathbb{I}\left[v^\star(s') - \beta^\star(s, a) \leq 0\right] \Big)^{(p-1)/p}. \qquad \text{(Holder's inequality)}$$

We can bound $\mathbb{P}\left(\mathbb{I}\left[\widehat{v}(s') - \widehat{\beta}(s,a) \le 0\right] \ne \mathbb{I}\left[v^\star(s') - \beta^\star(s,a) \le 0\right]\right)$ using the second part of [Lemma H.3](#) such that

$$
\begin{aligned}
\varepsilon_2^A &\lesssim \left\|\widehat{v}(s') - \widehat{\beta}(s,a) - v^\star(s') + \beta^\star(s,a)\right\|_p \left\|\widehat{v}(s') - \widehat{\beta}(s,a) - v^\star(s') + \beta^\star(s,a)\right\|^{\frac{p-1}{p+1}} \\
&= \left\|\widehat{v}(s') - \widehat{\beta}(s,a) - v^\star(s') + \beta^\star(s,a)\right\|_p^{\frac{2p}{p+1}} \\
&\le \|\widehat{v}(s') - v^\star(s')\|_p^{\frac{2p}{p+1}} + \left\|\widehat{\beta}(s,a) - \beta^\star(s,a)\right\|_p^{\frac{2p}{p+1}}.
\end{aligned}
$$

Putting the $\varepsilon_1^A$ and $\varepsilon_2^A$ together, we have

$$
\begin{aligned}
\varepsilon_A &\lesssim \|\widehat{v}(s') - v^\star(s')\|_p^{\frac{2p}{p+1}} + \left\|\widehat{\beta}(s,a) - \beta^\star(s,a)\right\|_p^{\frac{2p}{p+1}} && \text{(when } p \in [1,\infty)) \\
&\lesssim \|\widehat{v}(s') - v^\star(s')\|_\infty^2 + \left\|\widehat{\beta}(s,a) - \beta^\star(s,a)\right\|_\infty^2. && \text{(when } p = \infty)
\end{aligned}
$$

**Bounding $\varepsilon^B$: Error with correct indicator but wrong nuisances.** Now we focus on bounding $\varepsilon^B$.

$$
\begin{aligned}
\varepsilon_B &= \mathbb{E}\left[\psi(s,a,s';\widehat{q},\widehat{w},\widehat{\beta},\zeta^\star)\right] - \mathbb{E}\left[\psi(s,a,s';\widehat{q},w^\star,\widehat{\beta},\zeta^\star)\right] \\
&= \mathbb{E}(\widehat{w}(s,a) - w^\star(s,a))\left(r(s,a) + \gamma\rho(s,a,s';\widehat{v},\widehat{\beta},\zeta^\star) - \widehat{q}(s,a)\right) \\
&= \mathbb{E}(\widehat{w}(s,a) - w^\star(s,a))\left(r(s,a) + \gamma\rho(s,a,s';\widehat{v},\widehat{\beta},\zeta^\star) - \widehat{q}(s,a)\right) \\
&\quad - \mathbb{E}(\widehat{w}(s,a) - w^\star(s,a))(r(s,a) + \gamma\rho(s,a,s';v^\star,\beta^\star) - q^\star(s,a)) && \text{([Lemma H.4](#))} \\
&= \mathbb{E}(\widehat{w}(s,a) - w^\star(s,a))\left(\widehat{q}(s,a) - q^\star(s,a) + \gamma(\rho(s,a,s';\widehat{v},\widehat{\beta},\zeta^\star) - \rho(s,a,s';v^\star,\beta^\star))\right).
\end{aligned}
$$

In the [Lemma H.4](#) step, we used

$$
0 = (1-\gamma)\mathbb{E}_{d_1}v^\star(s_1) - \mathbb{E}[\psi(s,a,s';q^\star,\widehat{w},\beta^\star)] = (1-\gamma)\mathbb{E}_{d_1}v^\star(s_1) - \mathbb{E}[\psi(s,a,s';q^\star,w^\star,\beta^\star)].
$$

Finally, note that

$$
\begin{aligned}
&\rho(s,a,s';\widehat{v},\widehat{\beta},\zeta^\star) - \rho(s,a,s';v^\star,\beta^\star) \\
&= (1-\lambda)(\widehat{v}(s') - v^\star(s')) + \lambda\tau^{-1}(\widehat{v}(s') - v^\star(s'))\mathbb{I}\left[\zeta^\star(s,a,s') \le 0\right] \\
&\quad + \lambda(\widehat{\beta}(s,a) - \beta^\star(s,a))\left(1 - \tau^{-1}\mathbb{I}\left[\zeta^\star(s,a,s') \le 0\right]\right).
\end{aligned}
$$

Due to continuity of the CDF of $v^\star(s')$ at $\beta^\star(s,a)$ for all $s,a$, we have $\Pr(\zeta^\star(s',s,a) \le 0 \mid s,a) = \tau$ and so the last term vanishes. Thus, we're left with a quantity that is at most $\lesssim (\widehat{v}(s') - v^\star(s'))$. Therefore,

$$
\begin{aligned}
\varepsilon_B &\lesssim \mathbb{E}(\widehat{w}(s,a) - w^\star(s,a))(\mathcal{J}_{U\pm}(\widehat{q}(s,a) - q^\star(s,a))) \\
&\le \|\mathcal{J}'_{U\pm}(\widehat{w} - w^\star)\|_2 \|\widehat{q} - q^\star\|_2. && \text{(Holder's inequality)}
\end{aligned}
$$

Putting everything together, we obtain the desired rates:

$$
\begin{aligned}
|\widehat{V}_{d_1} - V_{d_1}^*| &\lesssim O_p(n^{-1/2}) + \|\mathcal{J}'_{U\pm}(\widehat{w} - w^\star)\|_2 \|\widehat{q} - q^\star\|_2 + \|\widehat{v} - v^\star\|_p^{\frac{2p}{p+1}} + \left\|\widehat{\beta} - \beta^\star\right\|_p^{\frac{2p}{p+1}} \\
&= O_p(n^{-1/2}) + O_p\left(r_n^w r_n^q + (r_{n,p}^q)^{\frac{2p}{p+1}} + (r_{n,p}^\beta)^{\frac{2p}{p+1}}\right) && \text{(when } p \in [1,\infty)) \\
&\lesssim O_p(n^{-1/2}) + \|\mathcal{J}'_{U\pm}(\widehat{w} - w^\star)\|_2 \|\widehat{q} - q^\star\|_2 + \|\widehat{v} - v^\star\|_\infty^2 + \left\|\widehat{\beta} - \beta^\star\right\|_\infty^2 \\
&= O_p(n^{-1/2}) + O_p\left(r_n^w r_n^q + (r_{n,\infty}^q)^2 + (r_{n,\infty}^\beta)^2\right). && \text{(when } p = \infty)
\end{aligned}
$$

## H.5 Proof of Normality & Efficiency

In this part of the theorem, we let:

$$\widetilde{V}_{d_1} = \frac{1}{K} \sum_{k=1}^{K} \mathbb{E}_k[\psi(s, a, s'; \eta^*)]$$

Then, we can write the following equality:

$$\sqrt{n}(\widehat{V}_{d_1} - V_{d_1}^*) = \sqrt{n}(\widehat{V}_{d_1} - \widetilde{V}_{d_1}) + \underbrace{\sqrt{n}(\widetilde{V}_{d_1} - V_{d_1}^*)}_{\xrightarrow{d} \mathcal{N}(0, \Sigma)}$$

The second term converges in distribution to $\mathcal{N}(0, \Sigma)$ from the CLT and the fact that $\psi$ is the efficient influence function. Thus, it remains to show that the first term is $o_p(1)$. We decompose the first term as follows:

$$\sqrt{n}(\widehat{V}_{d_1} - \widetilde{V}_{d_1}) = \sqrt{n}\frac{1}{K} \sum_{k=1}^{n} \Big( \mathbb{E}[\psi(s, a, s'; \widehat{\eta}^{[k]})] - \mathbb{E}[\psi(s, a, s'; \eta^*)] \Big) \tag{15}$$

$$+ \sqrt{n}\frac{1}{K} \sum_{k=1}^{n} \underbrace{(\mathbb{E}_k - \mathbb{E})[\psi(s, a, s'; \widehat{\eta}^{[k]}) - \psi(s, a, s'; \eta^*)]}_{\varepsilon_k} \tag{16}$$

In Eq. (15), we have that $|\mathbb{E}[\psi(s, a, s'; \widehat{\eta}^{[k]})] - \mathbb{E}[\psi(s, a, s'; \eta^*)]|$ is bounded as in Eq. (Rates). Given the theorem's assumption about the nuisance rates, this term is $o_p(n^{-1/2})$ and Eq. (15) is $o_p(1)$. We now seek to control the $\varepsilon_k$ term in Eq. (16). Letting $\mathcal{D}_k$ represent the samples in the $k^{\text{th}}$ fold, we leverage sample splitting to show that the mean of $\varepsilon_k \mid \mathcal{D}_k$ is 0:

$$\mathbb{E}[\varepsilon_k \mid \mathcal{D}_k] = \mathbb{E}[\mathbb{E}_k[\psi(s, a, s'; \widehat{\eta}^{[k]}) - \psi(s, a, s'; \eta^*)] - \mathbb{E}[\psi(s, a, s'; \widehat{\eta}^{[k]}) - \psi(s, a, s'; \eta^*)] \mid \mathcal{D}_k]$$
$$= 0$$

where we consider $\widehat{\eta}^{[k]}$ fixed with respect to the second expectation. The result follows from the fact that $\widehat{\eta}^{[k]}$ does not depend on $\mathcal{D}_k$. Then, we can invoke Chebyshev's inequality to obtain the following bound:

$$P\left( \frac{\varepsilon_k}{\text{Var}[\varepsilon_k \mid \mathcal{D}_k]^{1/2}} \geq \epsilon \,\middle|\, \mathcal{D}_k \right) \leq \frac{1}{\epsilon^2}, \ \forall \epsilon > 0$$

Thus, we have shown that $\varepsilon_k \mid \mathcal{D}_k = O_p(\text{Var}[\varepsilon_k \mid \mathcal{D}_k]^{1/2}) = O_p(n^{-1/2}\mathbb{E}[(\psi(s, a, s'; \widehat{\eta}^{[k]}) - \psi(s, a, s'; \eta^*))^2 \mid \mathcal{D}_k]^{1/2})$. Here, we used the fact that $n_K = n/K$ (the size of $\mathcal{D}_k$) and that $K$ is a fixed integer that doesn't grow with $n$. Moreover, $\varepsilon_k$ has 0 conditional mean.

For the remainder of the analysis, we leave the conditioning on $\mathcal{D}_k$ implicit for simplicity. To bound $\mathbb{E}[(\psi(s, a, s'; \widehat{\eta}^{[k]}) - \psi(s, a, s'; \eta^*))^2 \mid \mathcal{D}_k]^{1/2} = \|\psi(s, a, s'; \widehat{\eta}^{[k]}) - \psi(s, a, s'; \eta^*)\|_2$, we use similar notation and techniques as in Appendix H.4:

$$\|\psi(s, a, s'; \widehat{\eta}^{[k]}) - \psi(s, a, s'; \eta^*)\|_2 \leq \|\psi(s, a, s'; \widehat{q}, \widehat{w}, \widehat{\beta}) - \psi(s, a, s'; \widehat{q}, \widehat{w}, \widehat{\beta}, \zeta^*)\|_2 \tag{$\sigma_1$}$$

$$+ \|\psi(s, a, s'; \widehat{q}, \widehat{w}, \widehat{\beta}, \zeta^*) - \psi(s, a, s'; q^*, w^*, \beta^*, \zeta^*)\|_2 \tag{$\sigma_2$}$$

where we invoked Cauchy-Schwarz for the $L_2$ norm. We bound $\sigma_2$ as follows:

$$\sigma_2 \leq \|\psi(s, a, s'; \widehat{q}, \widehat{w}, \widehat{\beta}) - \psi(s, a, s'; q^*, \widehat{w}, \widehat{\beta}, \zeta^*)\|_2 \tag{$\sigma_{2a}$}$$

$$+ \|\psi(s, a, s'; q^*, \widehat{w}, \widehat{\beta}, \zeta^*) - \psi(s, a, s'; q^*, \widehat{w}, \beta^*, \zeta^*)\|_2 \tag{$\sigma_{2b}$}$$

$$+ \|\psi(s, a, s'; q^*, \widehat{w}, \beta^*, \zeta^*) - \psi(s, a, s'; q^*, w^*, \beta^*, \zeta^*)\|_2 \tag{$\sigma_{2c}$}$$

$$\leq \|\widehat{v} - v^*\|_2 + \gamma(1 - \lambda)\|\widehat{w}\|_2\|\widehat{v} - v^*\|_2 + \gamma\lambda\tau^{-1}\|\widehat{w}\|_2\|\widehat{v} - v^*\|_2 + \|\widehat{w}\|_2\|\widehat{q} - q^*\|_2 \tag{$\sigma_{2a}$}$$

$$+ \gamma\lambda\|\widehat{w}\|_2\|\widehat{\beta} - \beta^*\|_2 + \gamma\lambda\tau^{-1}\|\widehat{w}\|_2\|\widehat{\beta} - \beta^*\|_2 \tag{$\sigma_{2b}$}$$

$$+ \|\widehat{w} - w^*\|_2 \left( \|r\|_2 + \gamma(1 - \lambda)\|v^*\|_2 + \gamma\lambda\|\beta^*\|_2 + \gamma\lambda\tau^{-1}\|v^* - \beta^*\|_2 \right) \tag{$\sigma_{2c}$}$$

Given our rate assumptions, our boundedness assumptions for $\widehat{w}$, the implicit boundedness of $q^*, v^*, w^*, \beta^*$, as well as the ordering of the $L_2$ and $L_\infty$ norms, $\sigma_2$ is $o_p(1)$. We now bound the $\sigma_1$ term:

$$\sigma_2 = \gamma\lambda\tau^{-1}\left\|\widehat{w}(s,a)(\widehat{v}(s') - \widehat{\beta}(s,a))(\mathbb{I}[\widehat{v}(s') \le \widehat{\beta}(s,a)] - \mathbb{I}[v^*(s') \le \beta^*(s,a)])\right\|_2$$

There are two cases in which the difference of indicators is non-zero:

$$\begin{cases} \widehat{v}(s') \le \widehat{\beta}(s,a) \text{ and } v^*(s') > \beta^*(s,a) \Rightarrow \mathbb{I}[\widehat{v}(s') \le \widehat{\beta}(s,a)] - \mathbb{I}[v^*(s') \le \beta^*(s,a)] = 1 \\ \widehat{v}(s') > \widehat{\beta}(s,a) \text{ and } v^*(s') \le \beta^*(s,a) \Rightarrow \mathbb{I}[\widehat{v}(s') \le \widehat{\beta}(s,a)] - \mathbb{I}[v^*(s') \le \beta^*(s,a)] = -1 \end{cases}$$

In the first case, $\widehat{v}(s') - \widehat{\beta}(s,a) \le 0, \beta^*(s,a) - v^*(s') < 0$ and thus

$$|(\widehat{v}(s') - \widehat{\beta}(s,a))(\mathbb{I}[\widehat{v}(s') \le \widehat{\beta}(s,a)] - \mathbb{I}[v^*(s') \le \beta^*(s,a)])| \le |\widehat{v}(s') - \widehat{\beta}(s,a) + \beta^*(s,a) - v^*(s')|.$$

In the second case, $\widehat{v}(s') - \widehat{\beta}(s,a) > 0, \beta^*(s,a) - v^*(s') \le 0$ and

$$|(\widehat{v}(s') - \widehat{\beta}(s,a))(\mathbb{I}[\widehat{v}(s') \le \widehat{\beta}(s,a)] - \mathbb{I}[v^*(s') \le \beta^*(s,a)])| \le |\widehat{v}(s') - \widehat{\beta}(s,a) + \beta^*(s,a) - v^*(s')|.$$

Going back to $\sigma_1$, we have:

$$\sigma_2 \le \gamma\lambda\tau^{-1}\|\widehat{w}\|_2\|\widehat{v}(s') - \widehat{\beta}(s,a) + \beta^*(s,a) - v^*(s'))\|_2$$
$$\le \gamma\lambda\tau^{-1}\|\widehat{w}\|_2(\|\widehat{v} - v^*\|_2 + \|\widehat{\beta} - \beta^*\|_2)$$

By out theorem's assumptions, this term is also $o_p(1)$. Putting $\sigma_1$ and $\sigma_2$ together, we have that $\|\psi(s,a,s';\widehat{\eta}^{[k]}) - \psi(s,a,s';\eta^*)\|_2$ is $o_p(1)$ and $\varepsilon_k \mid \mathcal{D}_k$ is $o_p(n^{-1/2})$. By the bounded convergence theorem, this implies that $\varepsilon_k$ is also $o_p(n^{-1/2})$. Then, the term in 16 is $o_p(1)$, which further means that $\sqrt{n}(\widehat{V}_{d_1} - \widetilde{V}_{d_1}) = o_p(1)$. Our proof is now complete.

# I  Derivation of the Efficient Influence Function

We use the $\varepsilon$-contamination approach of [31] to derive an influence function (IF) for our estimand $V_{d_1}^-$. The proof for $V_{d_1}^+$ follows symmetrically. We note that since our tangent space is the whole space as it factorizes in the trivial way (as in [39, Page 54]), the IF we derive is actually the efficient influence function (EIF).

Let $P(s,a,s')$ denote the data distribution. Consider the $\varepsilon$-contamination $P_\varepsilon(s,a,s') = (1 - \varepsilon)P(s,a,s') + \varepsilon\delta(\bar{s},\bar{a},\bar{s}')$, where $\delta(\bar{z})$ is the dirac delta at $\bar{z}$, $i.e.$, $\delta(\bar{z})$ has infinite mass at $\bar{z}$ and 0 mass elsewhere. Let $V_\varepsilon^-$ denote the robust value function under the transition kernel $P_\varepsilon(s' \mid s,a)$. Omitting the $\varepsilon$ subscript means $\varepsilon = 0$. The IF of $V_{d_1}^-$ is then given by

$$\frac{\mathrm{d}}{\mathrm{d}\varepsilon}(1 - \gamma)\mathbb{E}_{d_1}V_\varepsilon^-(s_1)|_{\varepsilon=0}.$$

We dedicate the rest of this section towards this goal, which will be obtained in Theorem I.5.

**Lemma I.1.**

$$\frac{\mathrm{d}}{\mathrm{d}\varepsilon}P_\varepsilon(s' \mid s,a)|_{\varepsilon=0} = \frac{\delta(\bar{s},\bar{a})}{P(s,a)}(\delta(\bar{s}') - P(s' \mid s,a)).$$

*Proof.* Use the fact $P_\varepsilon(s' \mid s,a) = \frac{P_\varepsilon(s,a,s')}{P_\varepsilon(s,a)} = \frac{(1-\varepsilon)P(s,a,s')+\varepsilon\delta(\bar{s},\bar{a},\bar{s}')}{(1-\varepsilon)P(s,a)+\varepsilon\delta(\bar{s},\bar{a})}$ and take derivative.  □

**Lemma I.2** (IF of conditional expectation). *For any $s,a$ and $f_\varepsilon$,*

$$\frac{\mathrm{d}}{\mathrm{d}\varepsilon}\mathbb{E}_{P_\varepsilon}[f_\varepsilon(s') \mid s,a]|_{\varepsilon=0} = \frac{\delta(\bar{s},\bar{a})}{P(s,a)}(f(\bar{s}') - \mathbb{E}_P[f(s') \mid s,a]) + \mathbb{E}_P\left[\frac{\mathrm{d}}{\mathrm{d}\varepsilon}f_\varepsilon(s')|_{\varepsilon=0} \mid s,a\right],$$

*where $f = f_0$.*

*Proof.*

$$\frac{\mathrm{d}}{\mathrm{d}\varepsilon}\mathbb{E}_{P_\varepsilon}[f_\varepsilon(s') \mid s,a]|_{\varepsilon=0} = \sum_{s'} f(s')\frac{\mathrm{d}}{\mathrm{d}\varepsilon}P_\varepsilon(s' \mid s,a)|_{\varepsilon=0} + \sum_{s'}\frac{\mathrm{d}}{\mathrm{d}\varepsilon}f_\varepsilon(s')|_{\varepsilon=0}P(s' \mid s,a)$$

$$= \frac{\delta(\bar{s},\bar{a})}{P(s,a)}(f_0(\bar{s}') - \mathbb{E}_P[f_0(s') \mid s,a]) + \mathbb{E}_P\left[\frac{\mathrm{d}}{\mathrm{d}\varepsilon}f_\varepsilon(s')|_{\varepsilon=0} \mid s,a\right],$$

□

**Lemma I.3** (IF of conditional CVaR). *For any $\tau, s, a$ and $f_\varepsilon$,*

$$\frac{\mathrm{d}}{\mathrm{d}\varepsilon}\mathrm{CVaR}_{\tau,P_\varepsilon}[f_\varepsilon(s') \mid s,a]|_{\varepsilon=0} = \frac{\delta(\bar{s},\bar{a})}{P(s,a)}\big(\beta_\tau(s,a) + \tau^{-1}(f(\bar{s}') - \beta_\tau(s,a))_- - \mathrm{CVaR}_\tau(f(s') \mid s,a)\big)$$

$$+ \mathbb{E}_P\left[\tau^{-1}\mathbb{I}\left[f(s') \le \beta_\tau(s,a)\right]\frac{\mathrm{d}}{\mathrm{d}\varepsilon}f_\varepsilon(s')|_{\varepsilon=0} \mid s,a\right],$$

*where $f = f_0$ and $\beta_\tau(s,a)$ be the $(1-\tau)$-th quantile of $f(s'), s' \sim P(s,a)$.*

*Proof.*

$$\frac{\mathrm{d}}{\mathrm{d}\varepsilon}\mathrm{CVaR}_{P_\varepsilon}[f_\varepsilon(s') \mid s,a]|_{\varepsilon=0} \tag{17}$$

$$= \frac{\mathrm{d}}{\mathrm{d}\varepsilon}\min_b \mathbb{E}_{P_\varepsilon}\left[b + \tau^{-1}(f_\varepsilon(s') - b)_- \mid s,a\right]|_{\varepsilon=0} \tag{18}$$

$$= \frac{\mathrm{d}}{\mathrm{d}\varepsilon}\mathbb{E}_{P_\varepsilon}\left[\beta_\tau(s,a) + \tau^{-1}(f_\varepsilon(s') - \beta_\tau(s,a))_- \mid s,a\right]|_{\varepsilon=0}, \tag{19}$$

where the last equality is due to Danskin's theorem and the fact that $\beta_\tau(s,a)$ is the maximizer of the CVaR dual form at $\varepsilon = 0$. Continuing, let $g_\varepsilon(s';s,a) := \beta_\tau(s,a) + \tau^{-1}(f_\varepsilon(s') - \beta_\tau(s,a))_-$, so

$$\frac{\mathrm{d}}{\mathrm{d}\varepsilon}\mathbb{E}_{P_\varepsilon}[g_\varepsilon(s';s,a) \mid s,a]$$

$$= \frac{\delta(\bar{s},\bar{a})}{P(s,a)}(g(\bar{s}';s,a) - \mathbb{E}_P[g(s',s,a) \mid s,a]) + \mathbb{E}_P\left[\frac{\mathrm{d}}{\mathrm{d}\varepsilon}g_\varepsilon(s';s,a)|_{\varepsilon=0} \mid s,a\right] \quad \text{(Lemma I.2)}$$

$$= \frac{\delta(\bar{s},\bar{a})}{P(s,a)}(g(\bar{s}';s,a) - \mathrm{CVaR}_\tau(f(s') \mid s,a)) + \mathbb{E}_P\left[\tau^{-1}\mathbb{I}\left[f(s') \le \beta_\tau(s,a)\right]\frac{\mathrm{d}}{\mathrm{d}\varepsilon}f_\varepsilon(s')|_{\varepsilon=0} \mid s,a\right].$$

This concludes the proof. □

We now prove the key "one-step forward" lemma.

**Lemma I.4** (One-Step Forward). *For any state distribution $\nu(s)$, we have*

$$\mathbb{E}_{s\sim\nu}\left[\frac{\mathrm{d}}{\mathrm{d}\varepsilon}V_\varepsilon^-(s)|_{\varepsilon=0}\right]$$

$$= \frac{\nu(\bar{s})\pi(\bar{a} \mid \bar{s})}{P(\bar{s},\bar{a})}\big(r(\bar{s},\bar{a}) + \gamma\big((1-\lambda)V^-(\bar{s}') + \lambda\big(\beta_\tau(\bar{s},\bar{a}) + \tau^{-1}(V^-(\bar{s}') - \beta_\tau(\bar{s},\bar{a}))_-\big)\big)$$

$$- Q^-(\bar{s},\bar{a})\big)$$

$$+ \gamma\mathbb{E}_{s\sim\nu}\left[\mathbb{E}_{\pi,P}\left[\big((1-\lambda) + \lambda\tau^{-1}\mathbb{I}\left[V^-(s') \le \beta_\tau(s,a)\right]\big)\frac{\mathrm{d}}{\mathrm{d}\varepsilon}V_\varepsilon^-(s')|_{\varepsilon=0} \mid s\right]\right].$$

*Proof.* For any $s_1$, we have

$$\frac{\mathrm{d}}{\mathrm{d}\varepsilon}V_\varepsilon^-(s_1)$$

$$= \frac{\mathrm{d}}{\mathrm{d}\varepsilon}\mathbb{E}_{a_1\sim\pi(s_1)}\big[r(s_1,a_1) + \gamma((1-\lambda)\mathbb{E}_{P_\varepsilon}\big[V_\varepsilon^-(s_2) \mid s_1,a_1\big] + \lambda\operatorname{CVaR}_{\tau,P_\varepsilon}\big[V_\varepsilon^-(s_2) \mid s_1,a_1\big]\big]_{\varepsilon=0}$$

$$= \gamma\mathbb{E}_{a_1\sim\pi(s_1)}\bigg[(1-\lambda)\frac{\mathrm{d}}{\mathrm{d}\varepsilon}\mathbb{E}_{\tau,P_\varepsilon}\big[V_\varepsilon^-(s_2) \mid s_1,a_1\big]|_{\varepsilon=0} + \frac{\mathrm{d}}{\mathrm{d}\varepsilon}\operatorname{CVaR}_{\tau,P_\varepsilon}\big[V_\varepsilon^-(s_2) \mid s_1,a_1\big]|_{\varepsilon=0}\bigg]$$

$$= \gamma(1-\lambda)\mathbb{E}_{a_1\sim\pi(s_1)}\bigg[\frac{\delta(\bar{s},\bar{a})}{P(s_1,a_1)}\big(V^-(\bar{s}') - \mathbb{E}_P\big[V^-(s_2) \mid s_1,a_1\big]\big)\bigg]$$

$$+ \gamma(1-\lambda)\mathbb{E}_{a_1\sim\pi(s_1)}\mathbb{E}_P\bigg[\frac{\mathrm{d}}{\mathrm{d}\varepsilon}V_\varepsilon^-(s_2)|_{\varepsilon=0} \mid s_1,a_1\bigg]$$

$$+ \gamma\lambda\mathbb{E}_{a_1\sim\pi(s_1)}\bigg[\frac{\delta(\bar{s},\bar{a})}{P(s_1,a_1)}\big(\beta_\tau(s_1,a_1) + \tau^{-1}(V^-(\bar{s}') - \beta_\tau(s_1,a_1))_- - \operatorname{CVaR}_\tau(V^-(s_2) \mid s_1,a_1)\big)\bigg]$$

$$+ \gamma\lambda\mathbb{E}_{a_1\sim\pi(s_1)}\mathbb{E}_P\bigg[\tau^{-1}\mathbb{I}\big[V^-(s_2) \le \beta_\tau(s_1,a_1)\big]\frac{\mathrm{d}}{\mathrm{d}\varepsilon}V_{\pi,P_\varepsilon}^-(s_2)\bigg].$$

Taking expectation over $s_1 \sim \nu$, we have

$$\mathbb{E}_{s\sim\nu}\bigg[\frac{\mathrm{d}}{\mathrm{d}\varepsilon}V_\varepsilon^-(s)|_{\varepsilon=0}\bigg] = \gamma\frac{\nu(\bar{s})\pi(\bar{a} \mid \bar{s})}{P(\bar{s},\bar{a})}\bigg((1-\lambda)V^-(\bar{s}') + \lambda\big(\beta_\tau(\bar{s},\bar{a}) + \tau^{-1}(V^-(\bar{s}') - \beta_\tau(\bar{s},\bar{a}))_-\big)$$

$$- \big((1-\lambda)\mathbb{E}\big[V^-(s') \mid \bar{s},\bar{a}\big] + \lambda\operatorname{CVaR}_\tau(V^-(s') \mid \bar{s},\bar{a})\big)\bigg)$$

$$+ \gamma\mathbb{E}_{s\sim\nu}\bigg[\mathbb{E}_{\pi,P}\big[\big((1-\lambda) + \lambda\tau^{-1}\mathbb{I}\big[V^-(s') \le \beta_\tau(s,a)\big]\big)\frac{\mathrm{d}}{\mathrm{d}\varepsilon}V_\varepsilon^-(s')|_{\varepsilon=0} \mid s\big]\bigg].$$

Finally recall that $V^-$ satisfies the Bellman equation, so

$$(1-\lambda)\mathbb{E}\big[V^-(s') \mid \bar{s},\bar{a}\big] + \lambda\operatorname{CVaR}_\tau(V^-(s') \mid \bar{s},\bar{a}) = Q^-(\bar{s},\bar{a}) - r(\bar{s},\bar{a}).$$

This concludes the proof. $\qquad\square$

Equipped with our main one-step lemma, we can now unroll it an infinite number of steps to derive the IF of our estimand.

**Theorem I.5** (IF of Estimand). *Let us denote*

$$g(\bar{s},\bar{a},\bar{s}') := r(\bar{s},\bar{a}) + \gamma\big((1-\lambda)V^-(\bar{s}') + \lambda\big(\beta_\tau(\bar{s},\bar{a}) + \tau^{-1}(V^-(\bar{s}') - \beta_\tau(\bar{s},\bar{a}))_-\big)\big).$$

*Then, we have*

$$\mathbb{E}_{d_1}\bigg[\frac{\mathrm{d}}{\mathrm{d}\varepsilon}V_\varepsilon^-(s_1)|_{\varepsilon=0}\bigg] = \frac{d_{rob}^{\pi,\infty}(\bar{s},\bar{a})}{P(\bar{s},\bar{a})}g(\bar{s},\bar{a},\bar{s}').$$

*Proof.* Let $d_h$ denote the $h$-th step visitation in the robust MDP, with transition $P_{\mathrm{rob}}$ satisfying $\frac{P_{\mathrm{rob}}(s'|s,a)}{P(s'|s,a)} = (1-\lambda) + \lambda\tau^{-1}\mathbb{I}\big[V^-(s') \le \beta_\tau(s,a)\big]$. Then notice that the final term of Lemma I.4 is exactly $\mathbb{E}_{s\sim\nu}\big[\mathbb{E}_{\pi,P_{\mathrm{rob}}}\big[\frac{\mathrm{d}}{\mathrm{d}\varepsilon}V_\varepsilon^-(s')|_{\varepsilon=0} \mid s\big]\big]$. Therefore,

$$\mathbb{E}_{d_1}\bigg[\frac{\mathrm{d}}{\mathrm{d}\varepsilon}V_\varepsilon^-(s_1)|_{\varepsilon=0}\bigg]$$

$$= \frac{d_1(\bar{s})\pi(\bar{a} \mid \bar{s})}{P(\bar{s},\bar{a})}g(\bar{s},\bar{a},\bar{s}') + \gamma\mathbb{E}_{s_2\sim d_2}\bigg[\frac{\mathrm{d}}{\mathrm{d}\varepsilon}V_\varepsilon^-(s_2)|_{\varepsilon=0}\bigg]$$

$$= \frac{d_1(\bar{s})\pi(\bar{a} \mid \bar{s})}{P(\bar{s},\bar{a})}g(\bar{s},\bar{a},\bar{s}') + \gamma\frac{d_2(\bar{s})\pi(\bar{a} \mid \bar{s})}{P(\bar{s},\bar{a})}g(\bar{s},\bar{a},\bar{s}') + \gamma^2\mathbb{E}_{s_3\sim d_3}\bigg[\frac{\mathrm{d}}{\mathrm{d}\varepsilon}V_\varepsilon^-(s_3)|_{\varepsilon=0}\bigg].$$

Iterating the process, we have

$$\mathbb{E}_{d_1}\bigg[\frac{\mathrm{d}}{\mathrm{d}\varepsilon}V_\varepsilon^-(s_1)|_{\varepsilon=0}\bigg] = \sum_{h=1}^\infty \gamma^{h-1}\frac{d_h(\bar{s})\pi(\bar{a} \mid \bar{s})}{P(\bar{s},\bar{a})}g(\bar{s},\bar{a},\bar{s}') = \frac{d_{\mathrm{rob}}^{\pi,\infty}(\bar{s},\bar{a})}{P(\bar{s},\bar{a})}g(\bar{s},\bar{a},\bar{s}'),$$

as desired. $\qquad\square$

Finally, we can conclude that the IF in Theorem I.5 is in fact the efficient IF (EIF) because it is in the tangent space, as the tangent space is contains all functions [39].

# J   Additional Validity Guarantees for Orthogonal Estimator

Our orthogonal estimator has additional desirable properties such as *validity* when some nuisances are misspecified. Specifically, the bounds returned by our orthogonal estimator will be asymptotically valid, though possibly loose, when some nuisances are inconsistent, *i.e.*, do not converge to the their true values. Below, we detail conditions under which we achieve validity. To be concise, we focus on the $-$ case as the $+$ case is symmetric.

**Validity with correct $Q^\pm$.**   If $\widehat{Q} = Q^\pm$, we obtain valid bounds even if $w, \beta$ are inconsistent.

**Lemma J.1.** *For any $w, \beta$, we have $\mathbb{E}[\psi(s, a, s'; Q^-, \beta, w)] \leq V_{d_1}^-$ with equality when $\beta = \beta_\tau^-$.*

**Validity with $Q = \mathcal{T}_\beta^\pm Q$.**   Even if $\widehat{Q}$ is misspecified, we still have a valid bound if it solves a Bellman-type equation of the dual CVaR form. For a $\beta : \mathcal{S} \times \mathcal{A} \to \mathbb{R}$, define:

$$\mathcal{T}_\beta^\pm f(s, a) := r(s, a) + \gamma \Lambda^{-1}(s, a) \mathbb{E}[f(s', \pi_t) \mid s, a]$$
$$+ \gamma (1 - \Lambda^{-1}(s, a)) \mathbb{E}[\beta(s, a) + \tau^{-1}(s, a)(f(s', \pi_t) - \beta(s, a))_\pm \mid s, a].$$

**Lemma J.2.** *Fix any $w, \beta$. If $Q_\beta^\pm = \mathcal{T}_\beta^\pm Q_\beta^\pm$, then $\mathbb{E}[\psi(s, a, s'; Q_\beta^-, \beta, w)] \leq V_{d_1}^-$.*

*Remark* J.3.  Lemmas J.1 and J.2 are dual to each other: in Lemma J.1, the plug-in is consistent while the debiasing correction errs in the valid direction (*i.e.*, $\geq 0$ for $+$ and $\leq 0$ for $-$). In Lemma J.2, the plug-in is valid while the debiasing correction has expectation zero.

## J.1   Proofs for validity

**Lemma J.1.** *For any $w, \beta$, we have $\mathbb{E}[\psi(s, a, s'; Q^-, \beta, w)] \leq V_{d_1}^-$ with equality when $\beta = \beta_\tau^-$.*

*Proof.*

$$\mathbb{E}[\psi(s, a, s'; Q^-, \beta, w)] \leq (1 - \gamma) \mathbb{E}_{d_1}[V_\beta^-(s_1)] + \mathbb{E}[w(s, a)(Q^-(s, a) - \mathcal{T}_{\mathrm{CVaR}}^- Q^-(s, a))]$$
$$= V_{d_1}^- + 0 = V_{d_1}^-,$$

where the inequality comes from the fact that $\beta$ is sub-optimal for $\mathbb{E}[\beta(s, a) + \tau^{-1}(V^-(s') - \beta(s, a))_-]$. The same proof applies for $Q^+$. $\qquad\qquad\square$

We now prove Lemma J.2. First, we show that the $\mathcal{T}_\beta$ perspective gives rise to a dual definition of $Q^\pm$ (dual to Eq. (2)).

**Lemma J.4.**

$$Q^+(s, a) = \arg\min_{\beta: Q_\beta = \mathcal{T}_\beta^+ Q_\beta} Q_\beta(s, a), \quad Q^-(s, a) = \arg\max_{\beta: Q_\beta = \mathcal{T}_\beta^- Q_\beta} Q_\beta(s, a).$$

*Proof.* Unroll $Q^-(s, a) = r(s, a) + \gamma \inf_{U \in \mathcal{U}(P)} \mathbb{E}_U[r(s', a') + \gamma \inf_{U \in \mathcal{U}(P)} \mathbb{E}_U[\dots]]$, replacing each $\inf_{U \in \mathcal{U}(P)}$ with the convex combination of $\mathbb{E}$ and CVaR from Lemma 3.1. Then, write each CVaR using the dual form, *i.e.*, $\max_\beta \{\beta(s, a) + \tau^{-1}(s, a) \mathbb{E}[(\cdots - \beta(s, a))_+]\}$. By $s, a$-rectangularity, the scalar $\max_\beta$ separates per $s, a$, so we can pull all the maxes out front as a $\max$ over $\beta(s, a)$ functions. Note that not all $\beta(s, a)$ functions have a well-defined infinite sum in this manner, as $\mathcal{T}_\beta$ is not always a contraction. The condition $Q_\beta = \mathcal{T}_\beta^- Q_\beta$ exactly characterizes when this unrolling is well-defined. Thus, $Q^-$ is exactly the minimum $Q_\beta$ whenever this procedure of unrolling with $\beta$ is well-defined. This concludes the proof. $\qquad\qquad\square$

**Lemma J.2.** *Fix any $w, \beta$. If $Q_\beta^\pm = \mathcal{T}_\beta^\pm Q_\beta^\pm$, then $\mathbb{E}[\psi(s, a, s'; Q_\beta^-, \beta, w)] \leq V_{d_1}^-$.*

*Proof.*

$$\mathbb{E}[\psi(s, a, s'; Q_\beta^-, \beta, w)] = (1 - \gamma)\mathbb{E}_{d_1}[V_\beta^-(s_1)] + 0 \leq V_{d_1}^-.$$

The first equality is because the correction term is $\mathcal{T}_\beta^- Q_\beta^- - Q_\beta^-$, which is zero since $Q_\beta^-$ is a fixed point. The inequality is due to Lemma J.4. □

# K Additional Details for Main Experiment

## K.1 Environment

We consider a simple MDP with a one-dimensional state space $\mathcal{S} = [0, 5]$, a binary action space $\mathcal{A} = \{0, 1\}$, reward function

$$r(s, a) = \frac{26 - s^2 - \mathbb{I}[a = 1]}{26},$$

which we note takes values in the range $[0, 1]$, and with transitions given by

$$P(\cdot \mid s, a = 0) = \text{UnifClip}[s - 0.2, \ s + 1]$$
$$P(\cdot \mid s, a = 1) = \text{UnifClip}[0.2s - 0.02, \ s + 0.5],$$

where $\text{UnifClip}[a, b]$ denotes a uniform distribution between $\max(a, 0)$ and $\min(b, 5)$. In addition, the environment always starts in initial state $s_0 = 2$. Essentially, this is a simple control environment, where high rewards are obtained by maintaining state as close to zero as possible, the action $a = 1$ is a control action that (in expectation) moves the state closer to zero, and which occurs a small reward cost, and the action $a = 0$ is a passive action that allows the state to freely drift (with an overall drift away from zero).

## K.2 Target Policy

We focus on estimating the worst-case policy value $V_{d_1}^-$ for the simple threshold-based target policy $\pi_t$ which takes action $a = 1$ when $s \geq 2$, and $a = 0$ whenever $s < 2$.

## K.3 Logging Policy and Data Sampling Procedure

We sample data using an evaluation policy $\pi_b$ which is an $\epsilon$-smoothed threshold policy similar to $\pi_t$. Specifically, $\pi_b$ takes action $a = 1$ when $s \geq 1.5$ with probability 0.95, and takes action $a = 0$ when $s < 1.5$ with probability 0.95. We obtain a dataset $\{s_i, a_i, s_i', r_i\}$ by first rolling out with $\pi_b$ for 1000 burn-in time steps, and then sampling the tuple $(s, a, s', r)$ every 10 time steps. For each replication of our experiment, we sample 10,000 tuples in total.

## K.4 Calculation of True Worst-Case Policy Values

A major challenge in studying robust policy value estimation is that, even with ground truth knowledge of the MDP and/or access to a simulator, it may be intractable to estimate the robust policy values $V_{d_1}^\pm$. Fortunately, the above environment has the desirable property that we can analytically compute the best/worst-case transition distributions allowed by our sensitivity model, since no matter what policy $\pi_t$ the agent is acting with, it always strictly prefers transitions to smaller states. In detail, suppose that for some state, action pair $(s, a)$ we have $P(\cdot \mid s, a) = \text{Unif}[x, y]$, for some $0 \leq x \leq y \leq 5$. Then, letting $\alpha = 1/(1 + \Lambda(s, a))$, it is easy to verify that the worst case transition kernel is given by

$$U^-(\cdot \mid s, a) = (1 - \Lambda^{-1}(s, a))\text{Unif}[y - \alpha(y - x), y] + \Lambda^{-1}(s, a)\text{Unif}[x, y].$$

That is, the worst case transition kernel is given by a mixture of two uniform distributions. Therefore, we can easily simulate rollouts with the best/worst case transition kernels, and accurately estimate the robust policy values. This allows us to validate our methodology in this synthetic environment. Specifically, for each $\Lambda(s, a)$ we experiment with, we can compute the corresponding ground truth $V_{d_1}^-$ up to arbitrary precision via Monte Carlo sampling, by rolling out trajectories with $\pi_t$ in the adversarial MDP according to the above worst-case transition kernel.

Note as well that if one wanted to estimate the best-case policy value, analogous reasoning would give us

$$U^+(\cdot \mid s, a) = (1 - \Lambda^{-1}(s, a))\text{Unif}[x, x + \alpha(y - x)] + \Lambda^{-1}(s, a)\text{Unif}[x, y].$$

However, in our experiments we only concern ourselves with worst-case policy value estimation.

### K.5 Nuisance Estimation

We instantiate slight variations of Algorithms 1 and 2 using neural nets for the classes $\mathcal{Q}$, $\mathcal{B}$, and $\mathcal{W}$ used for fitting $Q^-$, $\beta^-$, and $w^-$ respectively, and linear sieves for the corresponding critic class $\mathcal{Q}$ that we perform maximization over for the minimax estimation of $w^-$. Specifically, we grow the linear sieve for the critic class in a data-driven way, as follows: at each step $k$ of the respective algorithm, we compute the best response $q_k \in \mathcal{Q}$ to the previous iterate solution $w_k \in \mathcal{W}$ by optimizing over a neural net class, and then we append this best-response function to the set of functions in our linear sieve for the corresponding critic class. Full exact nuisance estimation details necessary for reproducibility will be available in our code release.

### K.6 Estimators

We estimate the worst-case policy value using three different estimators:

- **Q**: Direct estimator given by:

$$\widehat{V}_{d_1}^- = \widehat{Q}^-(s_1, \pi_t(s_1)),$$

  where $s_1$ is the deterministic initial state.

- **W**: Importance sampling-style estimator using $\hat{w}^-$, which is given by:

$$\widehat{V}_{d_1}^- = \frac{1}{n} \sum_{i=1}^n \widehat{w}^-(s_i, a_i)\widehat{\xi}_i r_i,$$

  where

$$\widehat{\xi}_i = \Lambda^{-1} + (1 - \Lambda^{-1})(1 + \Lambda)\mathbb{I}\left[\widehat{V}^-(s_i') \leq \widehat{\beta}^-(s_i, a_i)\right].$$

- **Orth**: Our orthogonal estimator using EIF, given by

$$\widehat{V}_{d_1}^- = \frac{1}{n} \sum_{i=1}^n \psi(s_i, a_i, s_i'; \widehat{Q}^-, \widehat{\beta}^-, \widehat{w}^-).$$

Note as well that we used a simpler data splitting procedure rather than the cross-fitting procedure described in Algorithm 3. Specifically, we used the first 10,000 tuples for estimating nuisances, and the second 10,000 tuples for the final estimators. This was done for the sake of computational ease in running experiments with many replications, and was performed in the same way for all methods.

In addition, for extra robustness, in each experiment replication we ran the nuisance estimation pipeline 5 times (on the same fixed sampled dataset), and took the 80th percentile policy value estimates, since the estimators tend to under-estimate the true policy value by design, with greater under-estimation when the nuisance estimates are less well optimized.

# L  Empirical Investigation on Medical Application

Here, we describe an additional empirical investigation of our methodology on medical data. Specifically, we consider the problem of sepsis management using RL. For all parts of the investigation described below, fully complete details can be obtained from our code release.

## L.1  Motivation of Investigation

Training RL models in simulated environments derived from real-world data is an exciting avenue for leveraging AI towards critical medical use cases. However, doing this obviously has the downside that, unless one undergoes the very risky process of training an RL agent online via real medical interventions, one has to resort to training within simulators, and then has to account for the inevitable "sim-to-real" gap. Therefore, our robust OPE methodology provides an interesting approach for estimating worst-case performance of RL models under potential changes in dynamics when moving to real application.

## L.2  RL Environment

Our RL environment is based on the OpenAI Gym sepsis simulator environment of [44]. This RL environment allows for simulation of dynamic sepsis management, which was created by training a blackbox ML model to mimic observed transition dynamics from the real-world electronic health record-based MIMIC-III dataset [36]. This existing sepsis simulator is an episodic environment that continues until the agent either recovers or dies. It has a 46-dimensional state space containing various vital measurements, a discrete action space containing 24 possible actions (where an action is essentially the Cartesian product of some independent base actions). The reward function in this original simulator gives zero reward whenever an episode has not terminated, a +15 reward at termination when if the patient survives, or a -15 reward at termination if the patient dies. Please see [44] and the code release linked therein for additional details.

We built an RL environment for our investigation by creating a simple wrapper around this existing sepsis simulator, in order to make it fit our setup. In particular, we made the following key changes:

1. We made the environment infinite-horizon, by automatically looping to a new random starting state for a fresh patient whenever the episode in the base simulator terminates

2. We normalized the reward function so that it lies in range $[0, 1]$, where:
   (a) $r(s, a) = 0$ if patient dies
   (b) $r(s, a) = 1$ if patient recovers and is discharged
   (c) $r(s, a) = 0.5$ if treatment has not terminated for current patient

In addition, for this environment, we perform all experiments with $\gamma = 0.95$.

## L.3  Policies for Investigation

We constructed RL policies for our empirical investigation by training some deep RL models using the sepsis simulator environment.

In the case of the behavioral policy $\pi_b$ used to generate the observational offline data, we trained this policy by running Proximal Policy Optimization (PPO: [63]) over a relatively large (16,000) number timesteps, in order to emulate a reasonably good "current best practices" model for creating observational data.

In the case of the target policy $\pi_t$ to be evaluated, we trained this policy using Deep Q Learning (DQL: [53]), over a relatively small (1,600) number of timesteps, in order to emulate a potentially risky new candidate model.

| Λ | Median Policy Value Estimate | | |
|---|---|---|---|
| | **Q** | **W** | **Orth** |
| 1 | $.546 \pm .003$ | $.386 \pm .087$ | $.532 \pm .008$ |
| 2 | $.454 \pm .040$ | $.534 \pm .141$ | $.515 \pm .036$ |
| 4 | $.381 \pm .077$ | $.287 \pm .106$ | $.338 \pm .086$ |

Table 2: Median policy value estimate for sepsis management investigation, for each estimator and value of $\Lambda$ over 5 runs of each estimator from random initial seeds. The $\pm$ values are given by half the difference between 80th and 20th percentiles.

## L.4    Creating an Offline Dataset

Using our behavioral policy $\pi_b$ which we created as above, we generated a fixed offline dataset consisting of 20,000 observed tuples of state, action, reward, and next state. Unlike with our main empirical investigation in the main paper, we did not perform any "thinning" on these sampled tuples to make them more independent, so that the observed transitions are sequentially correlated as with real-world medical data.

## L.5    Nuisance Estimation

We perform nuisance estimation almost identically as in our main empirical investigation, with the only change being a slight change to our neural network architectures to better handle the large discrete action space. Specifically, instead of training neural networks that take state as input and produce $|\mathcal{A}|$ outputs (one per action), we train neural networks that take both state and action as inputs, using a learnt low-dimensional encoding of the actions, and produce a single output. Please see our code release for details.

## L.6    Estimators

We consider the same three estimators (**Q**, **W**, and **Orth**) as in our main empirical investigation. As in that investigation, we use these to estimate the worst-case policy value for the given $\Lambda(s, a)$. In addition, as in the main experiments, we consider these estimators for various fixed $\Lambda(s, a)$ that do not depend on $s$ or $a$. In this case, we consider $\Lambda \in \{1, 2, 4\}$, as these reflect a reasonable range of possible confounding strength for real application.

## L.7    Results

Below, in Table 2 we show the estimated policy value for all three estimators for each fixed $\Lambda \in \{1, 2, 4\}$. Here, we present the median policy value estimate over 5 runs of our estimators from random starting seeds after removing outliers.[2] In addition, we present a $\pm$ spread given by half the difference between the 80th and 20th percentiles.

Although for this investigation we cannot analytically compute the ground truth "true" adversarial policy values to evaluate against when $\Lambda > 1$, we can still analyze the trends of these estimators and compare them to those observed in our main synthetic experiment, and we can also compare their accuracy when $\Lambda = 1$.

First, in the case of $\Lambda = 1$, we computed the true policy value of $\pi_t$ to be within the range $0.532 \pm 0.002$ with 95% confidence. This is almost exactly equal to the median **Orth** estimator, but far outside the spread of outputs of the **Q** estimator. That is, although the **Q** estimator has somewhat lower variance in outputs over multiple runs for $\Lambda = 1$ compared with **Orth**, it appears to be far more biased.

Next, looking more broadly across all values of $\Lambda$, as in our main experiment, the **Q** and **Orth** estimators generally result in similar estimates to each other, and the **W** estimators are very variable.

---

[2]Specifically, we exclude policy value estimates that lie outside the possible range of $[0, 1]$, which occasionally occur due to bad optimization from the starting seed.

This may reflect the relative difficulty of estimating the $w^-$ nuisance function compared with $Q^-$ and $\beta^-$; although both **Orth** and **W** are affected by this difficulty, the **Orth** estimator has a theoretical robustness to the errors of these nuisance functions that the **W** estimator does not, as outlined in our theory.

We also observe that when $\Lambda = 1$ the **Q** estimator is significantly more stable than **Orth**, but when $\Lambda > 1$ the stability of **Orth** is either comparable to or superior to **Q**. In order to understand this, we first note that unlike in our main experiments, here the repetitions are re-runs of the estimators with the same offline sepsis dataset, so these $\pm$ spreads reflect potential computational errors rather than statistical errors. Given this, this pattern of errors could be explained by the fact that when $\Lambda = 1$ the **Q** estimation is extremely simple, reducing to standard FQI, whereas when $\Lambda > 1$ it requires a more complex robust FQI estimation with simultaneous estimation of $\beta^-$. That is, the difference in computational difficulty of estimating **Orth** versus **Q** may be smaller for $\Lambda > 1$.

Overall, although it is hard to definitively compare the accuracy of these estimators for $\Lambda > 1$ given a fundamental lack of ground truth, given both a similar pattern of results as in our synthetic experiments, as well as the far greater accuracy of **Orth** when $\Lambda = 1$, it seems reasonably to believe based on these results that our proposed **Orth** estimator may be more reliable than the existing robust FQI approach of the **Q** estimator.

Finally, we consider the implication of our results for the problem of learning sepsis management policies from simulators. Our **Orth** estimator suggests that there is relatively little sensitivity of this environment to deviations allowed by $\Lambda = 2$, but very significant deviation allowed by $\Lambda = 4$. Indeed, given the reward structure described above, the worst-case results under $\Lambda = 4$ imply an extremely high mortality rate. Whether worst-case deviations of this magnitude are reasonable or not is unclear, and this is something that requires further investigation for future work on RL for sepsis management.

