# OpenReview forum: "Efficient and Sharp Off-Policy Evaluation in Robust Markov Decision Processes"
_NeurIPS.cc/2024/Conference — NeurIPS 2024 poster_

### Official Review · Reviewer_kuGv · 2024-06-30

**Soundness:** 3
**Presentation:** 2
**Contribution:** 2
**Rating:** 6
**Confidence:** 3

**Summary:**

This paper proposes a statistical theory for off-policy evaluation given observational data from an unknown transition kernel of an original MDP, possibly under policies the same as or different from the target policy. This paper focuses on the situation of environment shift, i.e., the transition kernel of the MDPs at test time can shift from the original MDP to a certain extent. These transition kernels are assumed to be center around the observational transition kernel. This paper proposes new algorithms and sharp estimators for robust estimation of policy value. The proposed estimation method has several desired properties.. First, it is semiparametrically efficient, with a $\sqrt{n}$-rate of estimation even when certain nuisance functions are estimated at slow parametric rates. The estimator is also asymptotically normal which enables easy statistical inference. Even when the nuisance functions are estimated with bias, the estimator is still constant. Numerical simulations verify the efficiency and sharpness of the proposed estimator.

**Strengths:**

**Significance**:

**1.** This paper focuses on an important and challenging theoretical problem of policy evaluation under transition kernel shift. This is a very realistic and probably more common scenario compared to the classical OPE setting where only the policies shift.

**2** This paper presents a credible statistical theory for policy evaluation under transition kernel shift and the existence of nuisance functions. The statistical analysis shows that strong statistical guarantee can be achieved for the proposed estimator.

**Quality**: this paper has a good quality.

**1.** The paper has a solid theoretical analysis, with well specified definitions and theorems.

**Weaknesses:**

**1.** This paper is purely theoretical and thus it is very hard to evaluate the practicability of the proposed estimation framework. The numerical simulation also only covers a very simplified synthetic environment. Thus it is unclear how useful or computationally efficient the proposed estimation framework really is.

**2.** I feel this paper is missing a discussion of the line of work on OPE under POMDP. It seems that the OPE on POMDP is naturally connected with the topic of this paper, though I am not very familiar with that line of work.

**Minor**:

**1.** The notation of this paper is a bit heavy. For example, the plus-minus sign confused me for a while when I read about its meaning (“read twice”) at the bottom of page 3. But this is not a big issue and is sort of understandable given the complex framework of this paper.

**2.** This paper is a bit hard to read for readers unfamiliar with this topic. No example or demonstration of any kind is given.

**Questions:**

Please see weakness.

**Limitations:**

Yes.

---

> ### Author Rebuttal · Authors · 2024-08-05
>
> Dear reviewer kuGv, thanks for acknowledging the importance of our problem and the strengths of our theoretical results. Please find our responses to your questions below:
>
> **W1: "This paper is purely theoretical and thus it is very hard to evaluate the practicability of the proposed estimation framework. The numerical simulation also only covers a very simplified synthetic environment. Thus it is unclear how useful or computationally efficient the proposed estimation framework really is."**
>
> **Author's Response:** Thank you for the feedback. We would like to offer the following clarifications about usefulness and computational efficiency of our framework:
> 1. While our paper is indeed theoretical, it builds upon the well-established marginal sensitivity model (MSM) framework, which has been practically validated in numerous prior studies (e.g. Kallus and Zhou 2021 "Minimax-Optimal Policy Learning under Unobserved Confounding", Bruns-Smith and Zhou 2023 "Robust Fitted-Q-Evaluation and Iteration under Sequentially Exogenous Unobserved Confounders", and Dorn and Guo 2021 "Sharp Sensitivity Analysis for Inverse Propensity Weighting via Quantile Balancing", and many more). Thus, we can be assured that the MSM framework is useful to the broader causal inference community.
> 2. Our experiments, though synthetic, serve as a crucial proof of concept. We deliberately designed a synthetic environment where the ground truth $V^-$ is known, allowing us to compute the true error and show that our method converges the fastest.
> 3. Regarding computational efficiency, our method is compatible with neural nets as demonstrated in our experiments. Hence, our method offers comparable computational efficiency to standard OPE methods such as doubly robust estimation. This scalability ensures that our approach remains practical for larger-scale applications.
>
>
> **W2: "I feel this paper is missing a discussion of the line of work on OPE under POMDP. It seems that the OPE on POMDP is naturally connected with the topic of this paper, though I am not very familiar with that line of work."**
>
> **Author's Response:** Thanks for bringing up OPE for POMDPs. Indeed, there are several relevant lines of work in this area, each making *unverifiable* assumptions to identify the confounded policy value:
> 1. Namkoong et al. (2020) assume confounding occurs at only one step;
> 2. Shi et al. (2022) require access to auxiliary mediators;
> 3. Bennett and Kallus (2023) utilize bridge functions from proximal causal inference.
>
> Our approach builds on the well-established Marginal Sensitivity Model (MSM) from Tan (2006) to *bound* policy value in the absence of assumptions that point-identify the value given observations. Our key contribution is providing statistically optimal (semiparametrically efficient) estimators for the sequential "RL" setting, whereas prior estimators for the MSM were limited to single-step scenarios. Some of this was discussed in our related works section (Sec 1.1). We will add more discussion to the camera-ready.
>
> **W3: "The notation of this paper is a bit heavy. For example, the plus-minus sign confused me for a while when I read about its meaning (“read twice”) at the bottom of page 3. But this is not a big issue and is sort of understandable given the complex framework of this paper."**
>
> **Author's Response:** Thanks for the feedback about plus-minus. To improve readability for the camera-ready, we will focus on the minus case only in the main text and dedicate an appendix section to the symmetric plus case.
>
> **W4: "This paper is a bit hard to read for readers unfamiliar with this topic. No example or demonstration of any kind is given."**
>
> **Author's Response:** We concur that the topic is quite technical and many concepts & notations from stats and RL are required to achieve the mathematical rigor consistent with past works addressing similar topics. To help with clarity, we will dedicate another appendix section to discuss examples, both further elaborating on our experiments, and how the MSM can be useful in real applications (e.g. as in Bruns-Smith and Zhou 2023).
>
>
> Thanks again for your helpful feedback. We hope we have addressed your concerns and please let us know if you have any additional questions.

---

> > ### Comment · Reviewer_kuGv · 2024-08-12
> >
> > Thank you for your feedback. I will take all the responses into consideration during the discussion and decision session. In the paper revision, please make sure to (i) adding the above discussion in the paper revision to make the paper clearer and easier to read; (ii) adding results on real-world dataset, such as the health dataset mentioned in the rebuttal and perhaps other appropriate datasets.
> >
> > Best,
> > reviewer

---

### Official Review · Reviewer_ghkS · 2024-07-09

**Soundness:** 3
**Presentation:** 1
**Contribution:** 2
**Rating:** 6
**Confidence:** 4

**Summary:**

This paper studies the problem of evaluating a policy under best- and worst-case perturbations to a Markov decision process (MDP), given transition observations from the original MDP. The first contribution is the robust Q learing algorithm for CVaR problem, the second contribution is the estimation of robust visitation distribution in the function approximation setting and their complexity bounds.

**Strengths:**

The problem studies in this paper is of significant practical importance, which is to evaluate the worst/best performance over an uncertainty set. The authors designed algorithms and derived their complexity bounds, which are shown to be optimal.

**Weaknesses:**

- The authors seem to be using offline and off-policy interchangably, which is misleading and confusing.
- One important challenge in offline reinforcement learning is due to partial coverage and distribution shift. However,  this paper does not explicitly define the single-policy concentrability coefficient that measures the gap between the sampling distribution $\nu$ and the target policy $\pi$, and therefore did not consider this key challenge in the offline setting.
- The notations are unnecessarily complicated, and many are used without being defined. For example, in algorithm 1 line 6, the hat on top of the expectation, and the subscript of the expectation is not clear.
- There are many technical assumptions that remain unjustified. Please justify each of the assumptions, and provide examples when those assumptions can be satisfied, and how to verify those assumptions in practice.
- The perturbation model in eq. (1) is unjustified. It could happen easily that $\Lambda(s,a)$ is zero or infinite for some $(s,a)$. The authors shall explain why this perturbation model is of interest, and why this model is superior over existing uncertainty set models, e.g., those defined by KL divergence, TV.

**Questions:**

- There are several other important papers on robust offline reinforcement learning that are missing in this paper, e.g.,
Double Pessimism is Provably Efficient for Distributionally Robust Offline Reinforcement Learning: Generic Algorithm and Robust Partial Coverage
Jose Blanchet, Miao Lu, Tong Zhang, Han Zhong
Distributionally Robust Model-Based Offline Reinforcement Learning with Near-Optimal Sample Complexity
Laixi Shi, Yuejie Chi
Comparison with existing robust offline RL is not clear in this paper.
- The writing of this paper needs to be improved. Is Section 3 and Algorithm 1 only about CVaR?

---

> ### Author Rebuttal · Authors · 2024-08-05
>
> Dear reviewer ghkS, thanks for acknowledging the strengths of the theoretical results. Please find our responses to your questions below:
>
> **W1: "The authors seem to be using offline and off-policy interchangably, which is misleading and confusing."**
>
> **Author's Response:** We agree this may be confusing and will use "off-policy" consistently in revising since we focus only on evaluation.
>
> **W2: "One important challenge in offline reinforcement learning is due to partial coverage and distribution shift. However, this paper does not explicitly define the single-policy concentrability coefficient that measures the gap between the sampling distribution $\nu$ and the target policy $\pi$, and therefore did not consider this key challenge in the offline setting."**
>
> **Author's Response:** The concentrability coefficient is defined in Line 208, and it makes an appearance in our guarantee for robust FQE. Note that our concentrability coefficient differs from the one from standard OPE since the target policy's distribution is under the *adversarial MDP*. We will make this more apparent in the revision. We will also clarify how concentrability makes an appearance in the (efficient) variance our estimator achieves (namely, $\Sigma$ on Line 304). In particular, $\Sigma=\mathbb E[(\frac{\mathrm{d}d^{\pm,\infty}(s)}{\mathrm{d}\nu(s)})^2(\frac{\pi_{\mathrm t}(a\mid s)}{\nu(a\mid s)})^2(r(s,a)+\gamma \rho^\pm(s,a,s';v^\pm,\beta^\pm)-q^\pm(s,a))^2]$. That is, the amount of distribution shift (in the adversarial MDP) makes an explicit appearance in the (minimal) variance of robust evaluation.
>
> **W3: "The notations are unnecessarily complicated, and many are used without being defined. For example, in algorithm 1 line 6, the hat on top of the expectation, and the subscript of the expectation is not clear."**
>
> **Author's Response:** Thanks for catching, we use the \hat on top of $\mathbb{E}$ to denote empirical expectation (in this case, over the second split of $\mathcal{D}_i$). For the camera-ready, we will try to simplify and reduce notation (e.g. we will focus the main text on the - case, rather than presenting both -/+ cases together). However, we do want to concede that the nature of orthogonal ML and semiparametric statistics is inherently very technical and many concepts / notations are needed to achieve the mathematical rigor consistent with past works addressing similar topics.
>
> **W4: "There are many technical assumptions that remain unjustified. Please justify each of the assumptions, and provide examples when those assumptions can be satisfied, and how to verify those assumptions in practice."**
>
> **Author's Response:** We justify and discuss each of the assumptions right before / after they are stated. For example, Assumption 3.2 is discussed in lines 193-195; Assumption 3.3 in 199-200; Assumption 4.3 in Lines 254-255. We concede that these justifications are rather brief but we do note that our assumptions are *standard and known* in the OPE literature (Munos and Czepesvari 2008, Uehara et al. 2021). To improve readability, we will add more discussions about these assumptions for the camera-ready.
>
> **W5: "The perturbation model in eq. (1) is unjustified. It could happen easily that $\Lambda(s,a)$ is zero or infinite for some $(s,a)$. The authors shall explain why this perturbation model is of interest, and why this model is superior over existing uncertainty set models, e.g., those defined by KL divergence, TV."**
>
> **Author's Response:** Our perturbation model is a natural generalization of the celebrated Marginal Sensitivity Model (MSM; Tan 2006 [56]) from causal inference, which has been validated by many follow-up papers (e.g. Kallus and Zhou 2021 "Minimax-Optimal Policy Learning under Unobserved Confounding", Bruns-Smith and Zhou 2023 "Robust Fitted-Q-Evaluation and Iteration under Sequentially Exogenous Unobserved Confounders", and Dorn and Guo 2021 "Sharp Sensitivity Analysis for Inverse Propensity Weighting via Quantile Balancing", and many more). For example, the MSM importantly captures settings where the behavior policy and transition kernels may be perturbed by *unobserved confounders*, which is not captured by any those defined by KL or TV. Since the uncertainty set is a modeling choice there is no "superiority" of one over another, but the MSM does nicely capture possible unobserved selection effects. We elaborate on this point in Lines 84-93. Line 135 specifies $\Lambda(s,a)\in[1,\infty)$.
>
> **Q1: "There are several other important papers on robust offline reinforcement learning that are missing in this paper"**
>
> **Author's Response:** Thanks for pointing out these two papers. Both papers are for the task of robust offline policy *learning* (OPL, i.e. learning a robust policy), whereas we focus on the task of robust off-policy evaluation (OPE, i.e. evaluating a policy's performance under distribution shift). Both are important problems with their own challenges. In practice, even if one is using robust policy learning, robust OPE is still of paramount interest, as it allows policy makers to reliably assess the actual performance of the learned policy under environment shifts or confounding. We will certainly add more discussion about robust policy learning and cite the papers you mentioned.
>
> **Q2: "Is Section 3 and Algorithm 1 only about CVaR?"**
>
> **Author's Response:** Yes, they are for robust Bellman equations with $\lambda \mathbb{E}+(1-\lambda)\text{CVaR}$. Indeed, one of our key insights is that the confounding-robust uncertainty set of Eq (1) can be equivalently expressed by the above robust Bellman equation, leading to tractable solutions in Sec 3 and Alg 1. We highlight that our approach can be easily extended to deal with other convex risk measures such as entropic risk or mean-variance (Markowitz), which we leave for future work.
>
> Thanks again for your helpful feedback. We hope we have addressed your concerns and please let us know if you have any additional questions.

---

> > ### Comment · Reviewer_ghkS · 2024-08-11
> >
> > Thank you for the detailed response.
> > W1 addressed
> > W2 Addressed
> > W3 not clear from the rebuttal
> > W4 the reviewer would like to see why those assumptions are reasonable assumptions and how can they be satisfied in practice.
> > W5 why it cannot be zero or infinity?
> > Q1 and Q2 are addressed.

---

> ### Author Response · Authors · 2024-08-12
> **Response to Reviewer ghkS**
>
> Dear Reviewer ghkS,
>
> Thanks for specifying your questions and for taking the time to consider our response. We will now answer them in order:
>
> **W3 not clear from the rebuttal.**
> To clarify, the $\hat{\mathbb{E}}_D$ notation (where $D$ is a dataset of points $x_1, \dots, x_n$) is defined as follows. For any function $f(x)$, we define $\hat{\mathbb{E}}_D[f(x)]=\frac{1}{n}\sum_if(x_i)$. In other words, $\hat{\mathbb{E}}_D$ is the empirical expectation for the dataset $D$. In the revision, we will directly use the $\frac{1}{n}\sum_if(x_i)$ in Alg 1, Line 6 to avoid using this notation.
>
> **W4 the reviewer would like to see why those assumptions are reasonable assumptions and how can they be satisfied in practice.**
> We are happy to provide clarifications on why these assumptions are reasonable and when they are satisfied. We now discuss each assumption in the paper.
>
> **Assumption 3.2.** This assumption posits that the quantile regression step succeeds in Alg 1. Quantile regression is a well-established problem with many classical algorithms for doing so: for example, Meinshausen and Ridgeway showed that random forests can learn conditional quantiles and this satisfies Assumption 3.2 where the $err_{QR}$ converges to zero as the number of datapoints grows to infinity (Theorem 1 of Meinshausen and Ridgeway 2006). More recently, there are also effective deep RL approaches that successfully use neural networks to learn quantiles, such as QR-DQN (Dabney et al 2018a) and IQN (Dabney et al 2018b). These approaches have obtained state-of-the-art results in Atari and can also be used to satisfy this assumption.
>
> **Assumption 3.3.** The Bellman completeness (BC) assumption posits that the $Q$ function class is closed under the (robust) Bellman operator. BC is a standard assumption for proving convergence in offline RL (Xie et al. 2021) and has also appeared in prior robust MDP works (Panaganti et al 2022; Bruns-Smith and Zhou 2023). In fact, BC is necessary for TD-style algorithms to succeed; without it, TD can diverge or converge to bad fixed points, e.g. Tsitsiklis and Van Roy, 1996 showed such a counterexample. Since our Algorithm 1 is based on TD, it is quite natural for our results to also rely on BC.
>
> **Assumption 4.2.** This is a mild regularity assumption that simply requires (i) the outputs of our function class to have bounded outputs and (ii) the thresholding function $V^\pm(s')-\beta^\pm(s,a)$ to be continuously distributed around the point $0$. We note that our algorithms do not crucially rely on (ii); if the distribution is discrete, we can use the discrete form for CVaR.
>
> **Assumption 4.3.** This assumption is a realizability assumption for importance-weighted estimators (i.e. Algorithm 2). The first part (i) simply requires that our density function class $\mathcal{W}$ to realize the density ratio and can be satisfied as we increase our function class's expressivity. The second part (ii) posits that our adversary function class $\mathcal{F}$ is rich enough to capture the error terms that take the form $\mathcal{J}'(w-w^\pm)$ for $w\in\mathcal{W}$. We note that both (i) and (ii) are *monotone* in the function class size and can be satisfied by making functions more expressive (i.e. increasing size of neural network).
>
> Regarding Assumption 4.3, we remark that our algorithms and theory are in fact robust to violations. In particular, we proved in Appendix I that the error from Algorithm 2 will depend gracefully on the approximation errors to Assumption 4.3 (as we formalized in Assumption I.1). In fact, this robustness also holds for the BC assumption (Assumption 3.3), i.e. our bounds will gracefully degrade with the robust Bellman error: $\epsilon_{be} = \min_{q\in\mathcal{Q}}\max_{q'\in\mathcal{Q}}\|\|q-\mathcal{T}^\pm_{rob}q'\|\|_2$. We will add the robustness-to-BC result in our revision. To conclude, our theory and algorithms are robust even when these assumptions fail to hold exactly.
>
> **W5 why it ($\Lambda$) cannot be zero or infinity?**
> Recall that $\Lambda$ is a parameter chosen by the user to capture the magnitude of perturbations $U$, which ought to satisfy $\Lambda^{-1}(s,a) \leq \frac{dU(s'\mid s,a)}{dP(s'\mid s,a)} \leq \Lambda(s,a)$. Since the definition implies $\Lambda^{-1}(s,a)\leq\Lambda(s,a)$, we have that $\Lambda(s,a)\geq 1$ so it cannot be $0$ and must be at least $1$. The infinity case is possible: if $\Lambda(s,a)=\infty$, then $U(s'\mid s,a)$ is allowed to be arbitrarily different from $P(s'\mid s,a)$, and our robust evaluation would evaluate the best- or worst-paths, i.e. what is the maximum or minimum possible cumulative reward. This setting was also studied by Du et al. 2023.
>
> Thank you again so much for the detailed feedback and for taking our responses into consideration. Please let us know if you have more questions and we will do our best to answer them promptly. Thank you again for your valuable efforts in reviewing our paper.

---

> ### Author Response · Authors · 2024-08-12
> **More Citations**
>
> **Citations**
>
> [1] Du, Yihan, Siwei Wang, and Longbo Huang. "Provably efficient risk-sensitive reinforcement learning: Iterated cvar and worst path." ICLR 2023.
>
> [2] Meinshausen, Nicolai, and Greg Ridgeway. "Quantile regression forests." Journal of machine learning research 7.6 (2006).
>
> [3] Dabney, Will, et al. "Distributional reinforcement learning with quantile regression." Proceedings of the AAAI conference on artificial intelligence. Vol. 32. No. 1. 2018a.
>
> [4] Dabney, Will, et al. "Implicit quantile networks for distributional reinforcement learning." International conference on machine learning. PMLR, 2018b.
>
> [5] Panaganti, Kishan, et al. "Robust reinforcement learning using offline data." Advances in neural information processing systems 35 (2022): 32211-32224.
>
> [6] Tsitsiklis, John, and Benjamin Van Roy. "Analysis of temporal-diffference learning with function approximation." Advances in neural information processing systems 9 (1996).
>
> [7] David Bruns-Smith and Angela Zhou. Robust fitted-q-evaluation and iteration under sequentially exogenous unobserved confounders. arXiv preprint arXiv:2302.00662, 2023.
>
> [8] Xie, Tengyang, et al. "Bellman-consistent pessimism for offline reinforcement learning." Advances in neural information processing systems 34 (2021): 6683-6694.

---

> > ### Author Response · Authors · 2024-08-13
> >
> > Dear reviewer, the open discussion period is nearing the end. We hope our responses addressed your concerns. If you feel that they have we would greatly appreciate if you update your score accordingly. If you still have questions let us know. We worked hard to be thorough while clear in our response and address all questions raised. Thank you again!

---

> > > ### Comment · Reviewer_ghkS · 2024-08-13
> > >
> > > Thank you for the response! My concerns are addressed, and therefore, I raised the score to 6.

---

### Official Review · Reviewer_QdT5 · 2024-07-13

**Soundness:** 3
**Presentation:** 3
**Contribution:** 3
**Rating:** 6
**Confidence:** 3

**Summary:**

This paper studies off-policy evaluation problems in robust Markov decision processes, where the environment dynamics may change over time. The authors consider a perturbation model that modifies the transition probability up to a given multiplicative factor. They first propose two new algorithms for learning the Q-value function $Q$ and the density ratio function $w$. Then, they design a new policy value estimator based on the estimated nuisance parameters. The authors provide a theoretical analysis of the new algorithms and estimators, along with numerical experimental results to verify their effectiveness.

**Strengths:**

a. The studied problem of off-policy evaluation in robust MDPs is interesting and meaningful.

b. The theoretical analysis is solid. The proposed $Q$-function estimation algorithm achieves a $\widetilde{\mathcal{O}}(n^{-1/2})$ rate for parametric classes. More importantly, the proposed orthogonal estimator in Section 5 has a second-order dependence on the estimation error of nuisance parameters. As a result, it only requires the rate of nuisance parameter estimation to be faster than $n^{-1/4}$ to achieve an $n^{-1/2}$ rate.

c. The numerical results align with the theoretical analysis; the orthogonal estimator consistently achieves the best performance compared to the $Q$ estimator and $w$ estimator.

**Weaknesses:**

a. This paper considers a perturbation model where the ratio between the modified transition kernel and the nominal kernel is bounded by a sensitivity parameter. In Algorithms 1 and 2, the learner needs access to the parameter $\Lambda(s,a)$. However, it is unclear how to estimate $\Lambda(s,a)$ in real-world scenarios. Additionally, the authors claim that the perturbation model is more general and captures confounded settings. More detailed analysis and explanations are needed to support this claim.

b. Algorithm 2 requires solving a minimax optimization problem, which is usually very time-consuming in practice. In the experiments, the authors use an iterative method as an approximation. Is there any theoretical analysis for that implementation?

c. Additional experiments using real-world datasets would help to better demonstrate the practical applicability of the proposed algorithms.

Minor:
Redundant content in line 194

**Questions:**

See weakness part.

---

> ### Author Rebuttal · Authors · 2024-08-07
>
> Dear reviewer QdT5, thank you for your review and positive feedback on our work. We appreciate your recognition of the problem's significance, our theoretical contributions, and the alignment of our numerical results with the theory.
>
> **Motivation for perturbation model**
>
> If the cause of the perturbation is unobserved confounding, a common approach to set $\Lambda(s,a)$ is to conduct an ablation study by withholding one feature of the state at a time. In a clinical setting, for example, smoking status or BMI could be withheld one by one, and the ratio in (1) calculated. A practitioner could then set $\Lambda(s,a)$ by assuming that an unobserved confounder could induce a shift in the transition kernel up to, say, twice the maximum $\Lambda(s,a)$ observed in the ablation study (see [1]). Another approach is to consider multiple $\Lambda(s,a)$ values until a specific condition (e.g., the lower bound of the value being less than $\alpha$, or less than 0 if $r \in [-1, 1]$). For instance, [2] studied the effect of smoking on lung cancer and found that unobserved confounding had to be nine times larger in smokers ($\Lambda = 9$) than in non-smokers to negate the measured negative effect of smoking. Since other features in the data had smaller $\Lambda$ values (ranging from 1 to 3), the conclusion was that there is at least some negative effect of smoking despite the observational dataset being affected by unobserved confounding. We will add this discussion to the final draft.
>
> Our perturbation model has a one-to-one correspondence to the MSM model in the confounded setting, which places a constraint on the ratio of confounded and unconfounded policies (rather than the transition kernels). Specifically, our perturbation model implies the policy MSM and therefore it subsumes the confounded setting. However, the reverse is only true for $A=2$. For $A>2$, a bound on the policy ratio imples only a loose bound on the ratio of transition probabilities, thus yielding non-sharp bounds for $A>2$. This discussion is included in the related literature section.
>
> **Minimax optimization problem**
>
> A couple of points regarding the hardness of the minimax optimization problem and proposed solution:
>
> 1. In general these kinds of minimax problems given by conditional moment conditions can be efficiently solved. For example, [4] show that an iterative procedure of solving a sequence of T least squares regression problems and cost-sensitive classification problems can guarantee a $log(T)/T$-approximate solution (which works via a reduction to no-regret learning, similar to the original theoretical analysis of boosting.)
>
> 2. The solution we used is somewhat similar to that mentioned above, although differs a little. It can be seen as a special case of [5] motivated by the observation of [6] that such OWGMM estimators are equivalent to minimax estimators. The sieve-based procedure of [5] has strong theoretical guarantees of consistency and semiparametric efficiency when the sieve is grown sufficiently fast (as the number of data points $n$ increases), and we augment this kind of estimator by adding extra functions to the sieve via a similar no-regret style procedure as in [4], by iteratively computing the OWGMM estimator and finding the worst-case adversary function. This augmentation does nothing to subtract from its existing theoretical guarantees, but we find is empirically very helpful. Furthermore, since we compute the worst-case adversary at each iteration, it gives a computational certificate of how close to min-max optimality each iterate is, which can make the procedure trustworthy in practice. In terms of computation, this algorithm just involves minimizing a convex loss function a fixed number of times, and so is computationally efficient.
>
> We will provide some discussion of this.
>
>
> **Additional experiments**
>
> For the final version of the manuscript, we will include results using real-world healthcare data. Specifically, we will demonstrate how our method extends to complex, real=world scenarios via a case study using MIMIC-III data for off-policy evaluation of learned policies for the management of sepsis in the ICU with fluids and vasopressors (see [3] for example).
>
> [1] Hsu, J.Y. and Small, D.S., 2013. Calibrating sensitivity analyses to observed covariates in observational studies. Biometrics, 69(4), pp.803-811.
>
> [2] Cornfield, J., Haenszel, W., Hammond, E.C., Lilienfeld, A.M., Shimkin, M.B. and Wynder, E.L., 1959. Smoking and lung cancer: recent evidence and a discussion of some questions. Journal of the National Cancer institute, 22(1), pp.173-203.
>
> [3] Killian, T.W., Zhang, H., Subramanian, J., Fatemi, M. and Ghassemi, M., 2020. An empirical study of representation learning for reinforcement learning in healthcare. arXiv preprint arXiv:2011.11235.
>
> [4] Dikkala, N., Lewis, G., Mackey, L., & Syrgkanis, V. (2020). Minimax estimation of conditional moment models. Advances in Neural Information Processing Systems, 33, 12248-12262.
>
> [5] Hansen, L. P. (1982). Large sample properties of generalized method of moments estimators. Econometrica: Journal of the econometric society, 1029-1054.
>
> [6] Bennett, A., & Kallus, N. (2023). The variational method of moments. Journal of the Royal Statistical Society Series B: Statistical Methodology, 85(3), 810-841.

---

> > ### Comment · Reviewer_QdT5 · 2024-08-14
> >
> > Thank you for the detailed response. I will maintain my positive score. However, I also suggest that the authors should make an effort to simplify the notation and enhance the readability of the paper.

---

### Official Review · Reviewer_xbQo · 2024-07-22

**Soundness:** 2
**Presentation:** 1
**Contribution:** 3
**Rating:** 4
**Confidence:** 2

**Summary:**

This paper studies statistically-efficient robust/optimistic (e.g., worst- or best-case within the uncertainty set $\mathcal{U}$) off-policy evaluation with bounded Radon-Nikodym uncertainty sets centered around known nominal models. It proposes novel algorithms for computing robust/optimistic Q functions, and provides theoretical analysis for them. It further claims that the sharp and efficient estimators are optimal in the local-minimax sense, and provides preliminary experimental results to support the claim.

**Strengths:**

* The paper seems to include a wide collection of new theoretical results, with assumptions clearly stated and proofs attached.
* The problem itself is simple, fundamental, and stated in an comprehensible way.

**Weaknesses:**

* The writing of this paper needs to be improved, at least for non-experts in this specific area.
    * Frankly speaking, I'm having a lot of trouble following the paper, or even understand the problems and the results. Though I have to admit that I major in CS and don't have a strong statistics background.
    * First and foremost, it doesn't seem like the $\pm$ superscript is doing any good here, except confusing the reader for the first few moments. If I were to write this paper, I would just sell the story of "robust OPE", and say a sentence or two on how to re-interpret it as a optimistic evaluation scheme (I do think negating rewards is easier to understand).
    * Some sentences are broken and not fully comprehensible (e.g., lines 193~195), some notations are left undefined before their first use (e.g., the $\frac{dU}{dP}$ notation, which I regard as Radon-Nikodym derivative), and some paragraphs are duplicated with subtle differences (e.g., Lemma 4.1 and G.1).
    * The literature review part (Section 1.1 and Appendix B) seems to be very incomplete, and some paragraphs seem irrelevant (e.g., lines 84~104). Since this is a paper on robust OPE, a more comprehensive review on existing works regarding robust MDPs and OPE methods is definitely needed, especially a comparison on the settings and the bounds.
    * As far as I'm concerned, the flow of the whole paper needs major revision. For example, it's unclear why we need to estimate $w^{\pm}$ in Section 4, until in the estimator for $V^{\pm}\_{d\_1}$ is proposed in Section 5. Another issue is that I don't fully get the point why we need a few more complex procedures to estimate $V^{\pm}\_{d\_1}$ after we've got the estimates for $Q^{\pm}$. Besides, the key results are not explained in plain English (e.g., given the sentences between lines 211 and 217, I still don't see clearly what Theorem 3.4 means. What is the typical order of $\varepsilon^{\mathcal{Q}}_n$? What about the $\mathrm{err}$ term?)
    * Simply put, Sections 3~5 are not well-motivated, the results are not intuitively interpreted, and their connections need to be further specified.
    * Personally, I expect to first see a full overview of the proposed algorithm and the "ultimate" end-to-end sample complexity result (which is now hidden in the paragraph "Putting everything together" on lines 316~322) to get the big picture, before diving into all the technical details introducing the subroutines.
* Since I'm not an expert in this area, I don't fully understand the technical details. (***To AC:*** Please do take other reviewers' opinions for technical soundness. I've tried hard but failed to fully check the proofs.)
    * Specifically, I don't follow the proofs in Sections E.5 (for the second part of Theorem 5.3), G.1 (for Lemma 3.1) and G.2 (for Lemma 4.1). The latter two don't seem like legit proofs in my opinion.

**Questions:**

* Why do you use bounded Radon-Nikodym derivative for defining ambiguity sets, as I haven't seen people doing this? Does it have any connections with other "distances", e.g. total variation, KL-divergence, or Wasserstein distance?

**Limitations:**

There are a few sentences discussing limitations of the setting, but a summary is missing.

---

> ### Author Rebuttal · Authors · 2024-08-07
>
> **Weakness: Paper is difficult to read**
>
> Thank you for your detailed feedback on how the quality of writing in the paper could be improved. On some level, this is a very difficult research topic to present in a way that is easily accessible without at least some background in stats and/or causal inference, because of the topics that it considers (i.e. related to semiparametric efficiency theory, orthogonal ML, etc.) It is typical that papers on such topics, including many of the papers that we cite related to causal inference, double/debiased machine learning, or semiparametric efficiency theory, are similarly technically dense. Also, our work is cross-disciplanary between these research areas and RL too, so there is the additional challenge of making it accessible to both audiences, while also being technically rigorous enough for both.
>
> However, it is not our intention for the paper to be inaccessible, and we will work hard on improving its readability and accessibility for the final camera ready version, following the helpful advice of you and other reviewers. In particular, regarding your specific suggestions on improving readability:
>
> - **Use of $\pm$:** Our use of the $\pm$ notation is as a succinct way to denote that there are two different quantities being defined. For example, for the operator $\mathcal{T}^\pm$ defined in Lemma 3.1 in terms of $\text{CVaR}^\pm$, we are succinctly defining two different operators: $\mathcal{T}^+$, which is defined in terms of $\text{CVaR}^+$, and $\mathcal{T}^-$, which is defined in terms of $\text{CVaR}^-$. This was done because, ultimately, we are presenting estimators for both the worst-case and best-case policy values under the sensitivity model. However, we understand that this notation can be somewhat confusing and cumbersome, and so we propose the following change to the final camera ready version, which would eliminate all $\pm$ notation:
> 1. Instead of discussing both the best and worst cases (+/-) simultaneously, we will provide results only for worst case (-) in the main text.
> 2. We will add a brief discussion about how the best-case version can be reduced to the worst-case version for a transformed MDP. (This reduction works by replacing all rewards $r(s,a)$ with $1-r(s,a)$, and then replacing the final estimator $\psi$ with $1-\psi$.)
>
> - **Definition of notations / writing quality:** Thank you for pointing out these issues, we will do a thorough edit of the paper to fix all such writing issues and ensure that all non-standard notations are defined. Regarding the Radon-Nikodym derivative, we would argue that this is standard probability theory notation, but we will clarify.
>
> - **Literature Review:** The part of the literature review that you highlighted is actually specifically about unobserved confounding in both sequential / non-sequential decision making, not about robust-MDPs (which, other than the works we cite and discuss, do **not** allow for unobserved confounding). Given limited space we focussed our literature review on the areas most related to our paper (focussing on settings that allow for unobserved confounding), and relegated our discussion of the literature on robust MDPs more generally to Appendix B ("other related works"). However, we will add some clearer pointers to this discussion in the related work section in the main paper.
>
> - **Ordering of Content in Paper:** We acknowledge the feedback here, and can understand how the order of the paper may seem unintuitive for readers unfamiliar with the related literature. We believe the current order is inuitive for readers familiar with similar orthogonal policy value estimators in non-robust settings, which are defined in terms of $Q$ and $W$, however it could be made more accessible. Therefore, we propose to present these existing orthogonal estimators for non-robust MDPs before Sections 3-5, explaining how they are defined in terms of $Q$/$W$. In addition, at this location we will foreshadow the Seciton 5 result that our estimator generalizes this estimator, using robust versions of the $Q$/$W$ functions, which are the $Q$/$W$ functions for the worst-case MDP. After this motivation, sections 3/4 will naturally follow, as they describe how to identify and estimate the robust $Q$/$W$ functions that we just introduced/defined. Similarly, section 5 will naturally follow after these as it puts together how these robust $Q$/$W$ functions are combined in the more general estimator that was previously foreshadowed. In order to make space for this, we will move some of the more technical content that is less important for the overall narrative to the appendix.
>
> **Weakness: Issues in Proofs**
>
> While we are open to feedback on how the presentation of our proofs could be improved, we have gone to great lengths to ensure that these results are technically sound. Could you please provide specific details about your claim that "the latter two don't seem like legit proofs in my opinion" about our proofs of Lemma 3.1 / Lemma 4.1 during the discussion period? This is a vague statement and we are not sure how to respond without more specific detail about what you find problematic with these proofs, but we are happy to defend the technical details of our proofs.
>
>
> **Q1: Why do you use bounded Radon-Nikodym derivative for defining ambiguity sets?**
>
> We use this definition to allow for full generality to state spaces that could be either continuous or discrete, or some combination of the two. For discrete state spaces, this Radon-Nikkodym (RN) derivative is just the ratio of transition probabilities, whereas for continuous state spaces, this RN derivative is just the ratio of probability densities. In either case, as discussed in the paper, this definition comes naturally from generalizing the standard marginal sensitivity model used in causal inference in the single decision (i.e. non-RL, or horizon=1) setting. We will clarify this further in the final paper.

---

> > ### Comment · Reviewer_xbQo · 2024-08-11
> > **Thanks for the responses!**
> >
> > I appreciate the authors' efforts to settle my questions and concerns, especially their open attitude towards adjusting the presentation and flow of the paper.
> >
> > When I say I cannot follow the proofs, I'm not implying that the proofs are technically wrong or suspicious. Rather, I just cannot get through certain steps or sentences. For a few examples:
> > * The equation on line 657 seems to happen only with high probability, and the following sentence on line 658 doesn't make any sense to me (What is $n_K$? Why does it matter here?)
> > * In Appendix G.1, I don't know how to relate the equation between lines 717 and 718 to the statement of Lemma 3.1 (specifically, the form of $\mathcal{T}_{\mathrm{rob}}$. Besides, there is a typo in the equation (nothing after $\mathrm{sup}$).
> > * In Appendix G.2, two proof sketches are provided, but it's better to present only one in a clear and rigorous way.
> >
> > I understand that I'm probably not the target audience of this paper (lol), and this is a tough (and kind of ignorant) decision, but I decide to keep my rating for now.

---

> ### Author Response · Authors · 2024-08-12
>
> Thank you very much for specifying your questions about the technical appendix. We greatly appreciate the detailed review – this helps us improve our paper. We will answer them in order:
>
> * "The equation on line 657 seems to happen only with high probability"
>
> You are correct: this occurs with high probability, which is why we used the big-O in probability ($O_p$) notation in this expression. This in fact is simply a restatement of the display equation just above this line, arising from Chebychev's inequality. Recall that $a_n=O_p(b_n)$ iff for any $\eta>0$ we have $P(a_n/b_n\geq \epsilon)\leq\eta$ for some $\epsilon>0$. This is precisely what we get from the display by letting $\epsilon=1/\sqrt{\eta}$ with $a_n=\varepsilon_k$ and $b_n=\mathrm{Var}[\epsilon_k\mid \mathcal D_k)]^{1/2}$ and the probability $P$ being conditional on $\mathcal D_k$, $P(\cdot\mid\mathcal D_k)$. We will add a sentence here to make it extra clear.
>
> * "the following sentence on line 658 doesn't make any sense to me (What is $n_K$ Why does it matter here?)"
>
> Here we are simply defining $n_K$ in-line as $n/K$, being the size of the dataset $\mathcal D_k$, to make extra clear why the asymptotics are the same whether we take $n\to\infty$ or the size of the fold (wrt which we are taking Chebychev) to infinity. This is to clarify that the conditional big-O in probability ($O_p(\cdot\mid \mathcal D_k)$) indeed makes logical sense here. Again, we will add a sentence here to make it extra clear.
>
> * "In Appendix G.1, I don't know how to relate the equation between lines 717 and 718 to the statement of Lemma 3.1 (specifically, the form of $\mathcal{T}_{rob}$). Besides, there is a typo in the equation (nothing after $\sup$)."
>
> Firstly, thank you for catching the typo: the missing expression after the $\sup$ is $\mathbb E_{G}[f(s')]$. That this completes the proof follows by the definition of $G$ in line 715. In particular, inverting the definition, each feasible $U$ in the supremum in the definition of $\mathcal{T} _ {rob}$ can be equivalently expressed as $U = \Lambda^{-1}(s,a)P + (1-\Lambda^{-1}(s,a))G$ where $G$ satisfies the constraints in line 717. Thus, the supremum in $\mathcal{T} _ {rob}$ can be expressed as $\Lambda^{-1}(s,a)$ times the expectation under the nominal $P$, and $(1-\Lambda^{-1}(s,a))$ times the sup over $G$, i.e. $\sup_{U: \Lambda^{-1}(s,a)\leq \frac{dU(s'\mid s,a)}{dP(s'\mid s,a)}\leq \Lambda(s,a)}\mathbb{E} _ U[v(s')\mid s,a] = \Lambda^{-1}(s,a)\mathbb{E} _ P[v(s')\mid s,a] + (1-\Lambda^{-1}(s,a))\sup_{G\ll P: \|dG(\cdot\mid s,a)/dP(\cdot\mid s,a)\| _ \infty\leq \tau^{-1}(s,a)}\mathbb{E} _ G[v(s')]$.
> Finally, line 717 observes that the latter term is equivalent to CVaR, which is proved in Section 3.4 of [3]. Thank you for bringing up this question. Explaining the completion of the proof certainly merits another line or two -- although it follows from the previous lines, it can be made easier to follow.
>
> * "In Appendix G.2, two proof sketches are provided, but it's better to present only one in a clear and rigorous way."
>
> Thank you for the feedback. We hoped to provide intuition to the reader by providing two ways to think about this, but we do understand that a clear complete proof may be the most instructive. Here is a complete proof we can add here in the appendix:
>
> Fix any $s,a$ and let $\tau=\tau(s,a)$. Fix any function $v(s')\in\mathbb{R}$. We want to show that the worst-case $U^+ = \arg\max _ {U\in\mathcal{U}(P)} \mathbb{E} _ U[v(s')\mid s,a]$ has a closed form expression as shown in line 725. By the proof of Lemma 3.1 above, we can rewrite $U^+(s'\mid s,a) = \Lambda^{-1}(s,a)P(s'\mid s,a) + (1-\Lambda^{-1}(s,a)) G^+(s'\mid s,a)$, where $G^+ = \arg\max _ {G\ll P: \|dG(\cdot\mid s,a)/dP(\cdot\mid s,a)\| _ \infty\leq \tau^{-1}(s,a)}\mathbb{E} _ G[v(s')]$. Thus, it suffices to simplify $G^+$. To do so, we invoke the premise that the CDF of $v(s')$ is differentiable at $\beta^+ _ \tau$, i.e. $F _ v(\beta^+ _ {\tau,F _ v}(s,a)\mid s,a)=1-\tau$. This implies that the CVaR is exactly the conditional expectation of the $1-\tau(s,a)$-fraction of best outcomes, i.e. $\text{CVaR}^+ _ \tau(v(s')\mid s,a) = \mathbb{E}[v(s')\mid v(s')\geq\beta^+ _ \tau(s,a),s,a]$, which in turn is equal to $\tau^{-1}\mathbb{E}[v(s')\mathbb{I}[v(s')\geq\beta^+_\tau(s,a)]\mid s,a]$.
> Thus, $G^+(s'\mid s,a) = \tau^{-1}P(s'\mid s,a) \mathbb{I}[v(s')\geq\beta^+ _ \tau(s,a)]$. This concludes the proof for the $+$ case. The proof for the $-$ case is symmetric and follows identical steps.
>
>
> Thank you again so much for the detailed feedback about the technical appendix.
>
> **Citations**
>
> [1] Chernozhukov, Victor, et al. "Double/debiased machine learning for treatment and structural parameters." (2018): C1-C68.
>
> [2] Kallus, Nathan, and Masatoshi Uehara. "Double reinforcement learning for efficient off-policy evaluation in markov decision processes." Journal of Machine Learning Research 21.167 (2020): 1-63.

---

> > ### Author Response · Authors · 2024-08-13
> >
> > Dear reviewer, the open discussion period is nearing the end. We hope our responses addressed your concerns. If you feel that they have we would greatly appreciate if you update your score accordingly. If you still have questions let us know. We worked hard to be thorough while clear in our response and address all questions raised. Thank you again.

---

### Decision · Program_Chairs · 2024-09-25

**Decision:**

Accept (poster)

**Comment:**

Reviewers agreed that this paper made solid contributions to off-policy evaluation in robust Markov decision processes by proposing two algorithms with theoretical guarantees along with numerical results to verify the effectiveness of the proposed algorithms.